



# High resolution global grids of revised Priestley-Taylor and Hargreaves-Samani coefficients for assessing ASCE-standardized reference crop evapotranspiration and solar radiation

Vassilis G. Aschonitis[1], Dimitris Papamichail[2], Kleoniki Demertzi[2], Nicolo Colombani[3], Micol Mastrocicco[4], Andrea Ghirardini[1], Giuseppe Castaldelli[1], Elisa-Anna Fano[1]

[1]Department of Life Sciences and Biotechnology, University of Ferrara, Ferrara, Italy
[2]Department of Hydraulics, Soil Science and Agricultural Engineering, Aristotle University of Thessaloniki, Thessaloniki, Greece
[3]Department of Earth Sciences, Sapienza University of Rome, Rome, Italy
[4]Department of Environmental, Biological and Pharmaceutical Sciences and Technologies, Second University of Naples, Caserta, Italy

*Correspondence to*: Vassilis G. Aschonitis (schvls@unife.it)

**Abstract.** The objective of the study is to provide high resolution global grids of revised annual coefficients for the Priestley-Taylor (P-T) and Hargreaves-Samani (H-S) evapotranspiration methods after calibration based on ASCE-standardized Penman-Monteith method (ASCE method includes two reference crops: short clipped grass and tall alfalfa). The analysis also includes the derivation of global grids of revised annual coefficients for solar radiation $R_s$ estimations using the respective $R_s$ formula of H-S. The analysis was based on global gridded climatic data of the period 1950-2000. The method for deriving annual coefficients of P-T and H-S methods was based on partial weighted averages (p.w.a.) of their mean monthly values, which eliminate the effect of monthly coefficients that occur during periods where $ET_o$ and $R_s$ fall below a specific threshold. Five resolution global grids (30 arc-sec, 2.5, 5, 10 arc-min and 0.5 deg) of annual coefficients for each method were developed. The new coefficients were validated based on data from 140 stations located at various climatic zones of USA and Australia with expanded observations up to 2016. Nine statistical criteria including Taylor diagrams were used in the validation procedure. The validation procedure for $ET_o$ estimations of short reference crop showed that the P-T and H-S methods with the new revised coefficients outperformed in comparison to the typical methods reducing the $ET_o$ *RMSE* of estimated values by 39% and 36%, respectively. The estimations of $R_s$ using the H-S formula with revised coefficients reduced the *RMSE* by 30% in comparison to the typical H-S radiation formula (the given results are based on the finer resolution grid). All the statistical criteria indicated better performance of the revised coefficients of all resolutions versus the typical coefficients used in the original methods. Finally, a raster database was built consisting of: a) global maps of revised annual coefficients for the $ET_o$ methods of P-T and H-S for both reference crops and the $R_s$ H-S formula, b) global maps which indicate the optimum locations for using the original P-T and H-S methods and their expected error based on reference values. The provision of the database aims to improve $ET_o$ and $R_s$ estimations which are used in hydrologic/climatic applications when climatic data are limited. The datasets produced in this study are archived in



PANGAEA database (https://doi.pangaea.de/10.1594/PANGAEA.868808, https://doi.org/10.1594/PANGAEA.868808) and in ESRN-database (http://www.esrn-database.org or http://esrn-database.weebly.com/).

## 1 Introduction

The reference crop evapotranspiration $ET_o$ is defined as the maximum value of water losses by evaporation and transpiration above a reference crop (e.g. grass), which can be achieved under no water restrictions, and it is one of the most important parameters for water balance estimations and irrigation planning of crops (Allen et al., 1998). Several methods have been proposed for $ET_o$ estimations (Itenfisu et al., 2003; Allen et al., 2005; Wang and Dickinson, 2012; McMahon et al., 2013) with the most popular the FAO-56 Penman-Monteith (Allen et al., 1998), the Priestley-Taylor (Priestley and Taylor, 1972) and the Hargreaves-Samani (Hargreaves and Samani, 1982; 1985) methods. The FAO-56 has been updated to the ASCE-standardized method (Allen et al., 2005), which reflects the current state-of-the-art, providing $ET_o$ estimations for two reference crops (a short and a tall reference crop, which correspond to clipped grass and alfalfa, respectively). The ASCE-standardized method has been proposed by the ASCE-EWRI Task Committee as the most precise method and requires a wide range of climatic parameters, which in many cases are not available. The problem of data availability can be confronted by other methods such as the Priestley-Taylor and Hargreaves-Samani that require less information for their determination and they are considered as the most precise among the simplified methods with reduced parameters (Xu and Singh, 2002; Sumner and Jacobs, 2005).

The Priestley-Taylor (P-T) method requires solar radiation and temperature data. The P-T formula includes an empirical factor known as advection coefficient $a_{pt}$, which is usually set equal to 1.26 (Priestley and Taylor, 1972) and generally ranges between 1.08 and 1.34 for agricultural lands (Tateishi and Ahn, 1996; Xu et al., 2013). Other studies for various climatic conditions have shown that $a_{pt}$ presents significant spatial and seasonal variability (Castellvi et al., 2001; Xu and Singh, 2002; Moges et al., 2003; Pereira, 2004; Aschonitis et al., 2015). Weiß and Menzel (2008) used the value 1.26 for wet and the value 1.75 for dry climatic conditions, as suggested by Maidment (1992). The value $a_{pt}$=1.26 has been verified experimentally for bare irrigated soil (Eichinger et al., 1996). Theoretical simulations for the case of the reference crop in saturated soil have also verified the $a_{pt}$=1.26 for the case of non or restricted advection effects (Lhomme, 1997; McMahon et al., 2013). Low values of the advection coefficient (~1.14) have been reported by Singh and Irmak (2011) for Nebraska (USA), while high values ranging between 1.82-2.14 have been reported for cold-dry lands of Iran (Tabari and Talaee, 2011). Values of the $a_{pt}$ coefficient <1 have been reported for forested steep areas (Shuttleworth and Calder, 1979; Giles et al. 1984; Flint and Childs, 1991). Aschonitis et al. (2015) analyzed the monthly variation of $a_{pt}$ for the Italian territories and observed through regression analysis that more than 90% of the spatial variability of the seasonal $a_{pt}$ is explained by the spatial variability of vapor pressure deficit $DE$ (positive correlation). The rate of $a_{pt}$ variation per unit $DE$ was found significantly different between seasons and it was negatively correlated to net solar radiation and/or temperature. The general





trends of $a_{pt}$ led to the conclusion that colder-drier conditions due to low net radiation and high vapor pressure deficit tend to increase its values.

The Hargreaves-Samani method requires only temperature data including four empirical factors (or three depending on the formula). Internal part of the equation empirically describes the incident solar radiation $R_s$. A basic problem of the

5 Hargreaves-Samani method is that it tends to underestimate $ET_o$ under high wind conditions ($u_2 > 3$ m s$^{-1}$) and to overestimate $ET_o$ under conditions of high relative humidity (Allen et al., 1998). Analysis of the deviation of the Hargreaves-Samani method and recalibration for various climates have been recently performed by various authors (Tabari, 2010; Tabari and Talaee, 2011; Aschonitis et al., 2012; Mohawesh and Talozi, 2012; Rahimikhoob et al., 2012; Ravazzani et al., 2012; Bachour et al., 2013; Long et al., 2013; Mendicino and Senatore, 2013; Ngongondo et al., 2013; Berti et al., 2014; Heydari

and Heydari, 2014), which indicates a global interest for simplistic methods mainly driven by the lack of data.

The analysis of $ET_o$ at global scale is of special interest since it provides a general view about the spatio-temporal variation of this parameter, while together with rainfall provide significant information about the aridity of terrestrial systems. A basic limitation of global analysis is the lack of homogeneously distributed meteorological stations around the globe and especially in mountainous regions. The last years, climatic models, advanced interpolation and other methods have

15 succeeded to generate datasets of various climatic parameters (Hijmans et al., 2005; Sheffield et al., 2006; Osborn and Jones, 2014; Brinckmann et al., 2016) facilitating any attempt to develop $ET_o$ maps. Significant works of global $ET_o$ estimations have been performed from various scientists. Mintz and Walker (1993) used the Thorthwaite (1948) and provided isoline maps of $ET_o$. Tateishi and Ahn (1996) used the Priestley-Taylor method and provided $ET_o$ maps at 0.5 degrees resolution. Droogers and Allen (2002) used FAO-56 Penman-Monteith providing $ET_o$ maps at 10 min resolution and a modified

Hargreaves-Samani method which considers rainfall. Weiß and Menzel (2008) compared four different methods (Priestley-Taylor, Kimberly Penman, FAO-56 Penman-Monteith and Hargreaves-Samani) and provided $ET_o$ maps at 0.5 degrees resolution. Zomer et al. (2008) used Hargreaves-Samani method and succeed in developing the highest resolution (30 arc-sec) available $ET_o$ maps.

The objectives of the study are a) to develop high resolution (up to 30 arc-sec ~1 km) mean monthly maps of reference

crop evapotranspiration $ET_o$ for the period 1950-2000 at global scale using the most precise method ASCE-standardized method for both reference crops (short clipped grass and tall alfalfa), b) to identify the optimum locations for the application of the typical H-S and P-T formulas based on their proximity to the results of ASCE for short reference crop, c) to propose a new method for the derivation of readjusted mean annual coefficients of P-T and H-S methods based on ASCE for both reference crops (for H-S the analysis is made in two steps to recalibrate the internal radiation formula), d) to validate the

results of the re-adjusted P-T and H-S coefficients using data from meteorological stations from different locations with different climatic conditions. The final purpose of the aforementioned procedures is to develop a global database with revised coefficients for P-T and H-S methods. The availability of the database can support estimations of reference evapotranspiration and solar radiation in future research efforts for locations where climate data are limited.



## 2 Data and methods

### 2.1 Global climatic data

The analysis presented in this study was based on global climatic data obtained from the following databases:

- The database of Hijmans et al. (2005) provides mean monthly values for the parameters of precipitation, maximum,
minimum and mean temperature at 30 arc-sec spatial resolution. The data are provided as grids of mean monthly values of the period 1950-2000 (http://www.worldclim.org/). The database also includes a revised version of the GTOPO30 DEM based on SRTM DEM at 30 arc-sec spatial resolution, which was used for the estimation of atmospheric pressure. The DEM was also used as a base to calculate the distance from the coastlines in raster format at 30 arc-sec spatial resolution based on the Euclidean distance.

- The database of Sheffield et al. (2006) provides mean monthly values of parameters such as wind speed at the height of 10 m above the ground surface, incoming short and long-wave solar radiation, specific humidity, precipitation and temperature for the period 1948-2006 at 0.5 degrees spatial resolution. The data are available in the form of netcdf files of monthly values of each year for the period 1948-2006 (http://hydrology.princeton.edu/data.pgf.php).

- The database of Peel et al. (2007) provides the revised global Köppen-Geiger climate map. The data are provided in
raster form with 0.1 degrees spatial resolution. The climate map was developed using the GHCN version 2.0 dataset (Peterson and Vose, 1997) which includes precipitation data from 12396 stations and temperature data from 4844 stations data for the periods 1909-1991 and 1923-1993, respectively. The Köppen-Geiger map was used to verify the climatic type variability of the meteorological stations used in the validation dataset.

In this study, the $ET_o$ is estimated combining the databases of Hijmans et al. (2005) and Sheffield et al. (2006), as
follows: a) mean monthly values of maximum, minimum, mean temperature and precipitation of the highest spatial resolution (30 arc-sec ~ 1 km) were obtained from Hijmans et al. (2005), while b) wind speed, specific humidity and incident solar radiation were obtained from Sheffield et al. (2006) database. The specific humidity was converted to actual vapor pressure using the equation given by Peixoto and Oort (1996). The raster grids of climatic parameters obtained by Sheffield et al. (2006) were resampled to 30 arc-sec spatial resolution using the bilinear resampling scheme in order to produce grids
of respective spatial resolution equal to the database of Hijmans et al. (2005). All the 30 arc-sec raster derivatives of this study were used to produce rasters of the same parameters in four additional coarser resolutions (2.5 arc-min, 5 arc-min, 10 arc-min and 30 arc-min~0.5 deg.) in order to cover the range of the minimum and maximum resolution of the initial climatic data. The analysis, which is going to be presented in this study, is mainly based on the finer 30 arc-sec resolution, while additional results are given for the coarser resolution of 0.5 deg. for comparative purposes.

All the calculations presented in the next sections were performed in ArcGIS 9.3 ESRI environment at WGS84 ellipsoid coordinate system. For area coverage calculations or for estimations of mean global values of various parameters, coordinate system conversions were performed from WGS84 to projected Cylindrical Equal Area system (Antarctica is not included in



the maps, thus any % globe coverage calculations and derivation of mean global values of various parameters are referred to the rest terrestrial surface).

## 2.2 Methods

### 2.2.1 The ASCE standardized reference evapotranspiration method

The estimation of $ET_o$ using the ASCE method is performed by the following equation (Allen et al. 2005):

$$ET_o = \frac{0.408\Delta\left(R_n - G\right) + \dfrac{\gamma u_2 (e_s - e_a)C_n}{\left(T_{\mathrm{mean}} + 273.16\right)}}{\Delta + \gamma(1 + C_d u_2)} \qquad (1)$$

where $ET_o$: is the reference crop evapotranspiration (mm d$^{-1}$), $R_n$: the net solar radiation at the crop surface (MJ m$^{-2}$ d$^{-1}$), $u_2$: the wind speed at 2 m height above the soil surface (m s$^{-1}$), $T_{\mathrm{mean}}$: the mean daily air temperature ($^o$C), $G$: the soil heat flux density at the soil surface (MJ m$^{-2}$ d$^{-1}$), $e_s$: the saturation vapor pressure (kPa), $e_a$: the actual vapor pressure (kPa), $\Delta$: the

slope of the saturation vapor pressure-temperature curve (kPa $^o$C$^{-1}$), $\gamma$: the psychrometric constant (kPa $^o$C$^{-1}$), $C_n$ and $C_d$: are constants, which vary according to the time step and the reference crop type and describe the bulk surface resistance and aerodynamic roughness. The short reference crop (ASCE-short) corresponds to clipped grass of 12 cm height and surface resistance of 70 s m$^{-1}$ where the constants $C_n$ and $C_d$ have the values 900 and 0.34, respectively. The tall reference crop (ASCE-tall) corresponds to full cover alfalfa of 50 cm height and surface resistance of 45 s m$^{-1}$, where the constants $C_n$ and

$C_d$ have the values 1600 and 0.38, respectively (Allen et al., 2005). The use of Eq.1 at daily or monthly step for short reference crop is equivalent to FAO-56 method (Allen et al., 1998).

### 2.2.2 The Priestley-Taylor method

The calculation of Priestley-Taylor (P-T) method is performed by the following equation (Priestley and Taylor, 1972):

$$ET_o = a_{pt} \frac{\Delta}{\lambda\left(\Delta + \gamma\right)}\left(R_n - G\right) \qquad (2)$$

where $ET_o$: is the potential evapotranspiration (mm d$^{-1}$), $R_n$: the net solar radiation (MJ m$^{-2}$ d$^{-1}$), $G$: the soil heat flux density at the soil surface (MJ m$^{-2}$ d$^{-1}$), $\Delta$: the slope of the saturation vapor pressure-temperature curve (kPa $^o$C$^{-1}$), $\gamma$: the psychrometric constant (kPa $^o$C$^{-1}$), $\lambda$: the latent heat of vaporization (MJ kg$^{-1}$) and $a_{pt}$: the P-T advection coefficient. The value of $\lambda$ was considered equal to 2.45 MJ kg$^{-1}$ (Allen et al., 1998) (this value is also constant in Eq.1 and appears as

$1/\lambda$=0.408). Eq.1 strictly refers to the reference crop evapotranspiration (i.e. short or tall crop), whereas Eq.2 has been used for the calculation of evapotranspiration under non-limiting water conditions of short reference crop, bare soil or open water surface and for this reason is also called potential evapotranspiration which is a more general term. Eq.2 is applied in this



study as a reference crop evapotranspiration method and for this reason is compared with Eq.1 for short reference crop using the typical mean global value 1.26 for the factor $a_{pt}$.

### 2.2.3 The Hargreaves-Samani method

The Hargreaves-Samani (H-S) method (Hargreaves and Samani, 1982;1985) for $ET_o$ includes an internal function, which estimates the incoming shortwave solar radiation $R_s$ (MJ m$^{-2}$ day$^{-1}$), as follows:

$$R_s = K_{RS} \cdot R_a \cdot (TD)^{0.5} \tag{3}$$

where $K_{RS}$: is the adjustment coefficient for the H-S radiation formula (°C$^{-0.5}$), $R_a$: the extraterrestrial radiation (MJ m$^{-2}$ day$^{-1}$) and $TD$: the temperature difference between maximum and minimum daily temperature (°C). According to Allen et al. (1998), the empirical $K_{RS}$ coefficient differs for 'interior' or 'coastal' regions: a) $K_{RS}$=0.16 for "interior" locations, where land mass dominates and air masses are not strongly influenced by a large water body and b) $K_{RS}$ =0.19 for "coastal" locations, situated on or adjacent to the coast of a large land mass and where air masses are influenced by a nearby water body. For general use of Eq.3, a mean global value of $K_{RS}$=0.17 has been adopted in this study. $R_a$ and $R_s$ divided by $\lambda$ change units from MJ m$^{-2}$ day$^{-1}$ to mm day$^{-1}$ as it is required in the next equation of $ET_o$. The formula for estimating the $ET_o$ by H-S method is given by the following equation (Hargreaves and Samani, 1982; 1985):

$$ET_o = 0.0135 \left( T_{mean} + 17.8 \right) \frac{R_s}{\lambda} = 0.0135 \left( T_{mean} + 17.8 \right) K_{RS} \cdot \frac{R_a}{\lambda} \cdot (TD)^{0.5} \tag{4a}$$

Considering Eq.4a, the $K_{RS}$ and the exponent 0.5 are adjustment factors of radiation formula (Eq.3) while the 0.0135 and 17.8 are adjustment factors of the $ET_o$ formula leading to a total of four empirical coefficients. Using the mean global value of $K_{RS}$=0.17, Eq.4a is simplified according to the following (Allen et al., 1998):

$$ET_o = c_{hs2} \cdot \left( T_{mean} + 17.8 \right) \cdot \frac{R_a}{\lambda} \cdot (TD)^{0.5} = 0.0023 \left( T_{mean} + 17.8 \right) \cdot \frac{R_a}{\lambda} \cdot (TD)^{0.5} \tag{4b}$$

where in both Eqs.4a and 4b, the $ET_o$: is the potential evapotranspiration (mm d$^{-1}$), $R_a$: the extraterrestrial radiation (MJ m$^{-2}$ d$^{-1}$), $\lambda$: the latent heat of vaporization (MJ kg$^{-1}$) and $T_{mean}$: the mean daily temperature (°C).

The Eq.4b is applied in this study as a reference crop evapotranspiration method and for this reason is compared with Eq.1 for short reference crop. For the case of $T_{mean}$<-17.8 °C, the term of ($T_{mean}$+17.8) was set to zero which is necessary for a global application (Weiß and Menzel, 2008).

In further steps of analysis the coefficient 0.0135 (Eq.4a) is symbolized as $c_{hs1}$ while the coefficient 0.0023 is symbolized as $c_{hs2}$ which is equal to $c_{hs2} = c_{hs1} \cdot K_{RS}$.

A modification of Eq.4a, which includes the monthly precipitation, has been presented by Droogers and Allen (2002) (H-S$_{DRAL}$ model) and it is given by the following equation:

$$ET_o = 0.0013 \cdot 0.408 R_a \left( T_{mean} + 17.0 \right) \cdot (TD - 0.0123P)^{0.76} \tag{4c}$$



where $P$: is precipitation in (mm month$^{-1}$). Eq.4c is used in the analysis in order to compare the predictive power of constant annual coefficients versus formulas that readjust the monthly $ET_o$ using additional easily accessible data.

In order to reduce the errors of the aforementioned methods in the high latitudes and altitudes (polar and alpine environments) where negative temperatures exist, a filter was used in all methods to set mean monthly $ET_o$=0 when mean monthly $T_{max}$ is ≤0 (conditions of extreme frost).

### 2.2.4 Steps of analysis

*Step 1: Comparative analysis between the typical $ET_o$ formulas of ASCE, P-T and H-S and error analysis of H-S radiation formula*

The first step of the analysis includes the estimation of mean monthly and mean annual $ET_o$ using the ASCE method (Eq.1) for the two reference crops (short and tall), the typical P-T method (Eq.2) with $a_{pt}$=1.26 and the typical H-S method according to Eq.4b with $c_{hs2}$=0.0023. The difference between the $ET_o$ methods will be captured using as a base the mean annual and the mean monthly $ET_o$ values of ASCE-short.

In the case of mean annual $ET_o$, the analysis is based on the % of mean annual difference ($MAD\%$) of each method M versus the mean annual $ET_o$ of ASCE-short which is given by:

$$MAD\%_{(M)} = 100\left[YET_{o(M)} - YET_{os}\right] / \left(YET_{os}\right) \tag{5}$$

where $YET_{os}$: is the mean annual $ET_o$ of method ASCE-short, $YET_{o(M)}$: is the mean annual $ET_o$ of M method (as M is used ASCE-tall either they typical P-T or the typical H-S). The $MAD\%$ for ASCE-tall was estimated in order to assess the effects of reference crop type at different climatic environments on the annual estimations of $ET_o$. The $MAD\%$ of P-T and H-S methods was used to investigate the strength of the two methods to approximate the annual $ET_o$ of ASCE-short. Positive values of $MAD\%$ indicate overestimation of the mean annual $ET_o$ values using the M method in comparison to ASCE-short method while negative values indicate underestimation, respectively. Furthermore, the difference between the absolute $MAD\%$ values ($DMAD$) of the typical P-T (with $a_{pt}$=1.26) and H-S (with $c_{hs2}$=0.0023) methods was estimated in order to assess which of the two methods is more appropriate to be used locally based on its proximity to ASCE-short method. The $DMAD$ was estimated as follows:

$$DMAD=|MAD\%_{(H-S)}|-|MAD\%_{(P-T)}| \tag{6}$$

where positive $DMAD$ values indicate better performance of P-T while negative values indicate better performance of H-S method in a region. Regions that showed $DMAD$ values between -1 and +1 were considered transition zones where both methods were assumed to show approximately the same proximity to the annual ASCE-short estimations.

In the case of mean monthly $ET_o$, the coefficient of determination $R^2$ and the root mean square difference $RMSD$ (equivalent to $RMSE$) (Droogers and Allen, 2002) were used to compare the mean monthly values of ASCE-tall, typical P-T (with $a_{pt}$=1.26) and H-S (with $c_{hs2}$=0.0023) methods with the respective values of the ASCE-short method.



The procedures of *MAD*%, $R^2$ and *RMSD* were also used to analyze the mean annual and mean monthly estimations of $R_s$ by the typical solar radiation formula of H-S (Eq.3 with $K_{RS}$=0.17) versus the given $R_s$ values of Sheffield et al. (2006).

*Step 2: Readjustment of annual P-T and H-S coefficients for both reference crops*

For the case of P-T, the readjustment of the mean monthly $a_{pt}$ coefficient was performed directly for each location by solving for $a_{pt}$ after equating Eq.1 with Eq.2 of each month. A filter was used in order to set $a_{pt}$ equal to 0 when Eq.1 or/and Eq.2 without $a_{pt}$ are equal to 0. In this case, the $a_{pt}$ changes its physical meaning in order to indicate that mean monthly $ET_o$ approximates to 0. Doing the above procedures for both short and tall reference crop, twelve images of mean monthly readjusted $a_{pt}$ coefficients were produced for each reference crop.

For the case of H-S method, the readjustment of the coefficients is performed in two stages. In the first stage, the readjustment is performed in the radiation formula (Eq.3) only for the $K_{RS}$ coefficient while the exponent 0.5 (square root) of the *DT* remains as it is. The mean monthly $K_{RS}$ is estimated using the values of solar radiation $R_s$ given by Sheffield et al. (2006). In the second stage, the readjustment is performed in the evapotranspiration formula (Eq.4b) for the coefficient of $c_{hs2}$ using as a base the ASCE method for both reference crops while the coefficients of 17.8 and 0.5 remain the same. In this

way the readjusted values of $c_{hs2}$ and $K_{RS}$ can also provide readjusted values of the $c_{hs1}$ since $c_{hs1} = c_{hs2}/K_{RS}$. A similar filter to set $c_{hs2}$=0 as in the case of $a_{pt}$ was used, when Eq.1 or/and Eq.4b without $c_{hs2}$ are equal to 0. Following the above procedures, twelve images of mean monthly readjusted $c_{hs2}$ coefficients for each reference crop (short and tall) and twelve $K_{RS}$ images were produced.

The new mean monthly $a_{pt}$, $c_{hs2}$ and $K_{RS}$ coefficients were used to build respective mean annual coefficients. The

robustness of mean annual coefficients is strongly related to their ability to capture better the values of the dependent variable (i.e. $ET_o$ and $R_s$) especially in the months which present its larger values (i.e. summer/hot months). For this reason, weighted annual averages of mean monthly $a_{pt}$, $c_{hs2}$ and $K_{RS}$ coefficients were estimated. Under cold conditions, the monthly coefficients may present unrealistic values that significantly affect the weighted averages. In order to solve this problem, threshold values for the mean monthly dependent variables (i.e. $ET_o$ and $R_s$) were set before their participation in the

weighted average estimations. Preliminary analysis for the readjustment of $a_{pt}$ and $c_{hs2}$ coefficients (based on the values of ASCE-short) showed that when the mean monthly $ET_o$ values of ASCE-short, H-S and P-T were below 45 mm month$^{-1}$ (~1.5 mm d$^{-1}$), then unrealistic mean monthly values of $a_{pt}$ and $c_{hs2}$ coefficients started to appear (as unrealistic values were considered those, which were at least one order of magnitude larger or smaller from the typical values of $a_{pt}$=1.26 and $c_{hs2}$=0.0023). Taking into account the above, the following procedure was followed in order to obtain a partial weighted

annual average (after excluding months with $ET_o$≤45 mm month$^{-1}$) of mean monthly $a_{pt}$ and $c_{hs2}$ coefficients for short reference crop (based on ASCE-short method):

$$\text{when } ET_{os\ i} > 45 \text{ mm month}^{-1} \quad \text{then} \quad Fr_i=1 \text{ else } =0 \qquad (7a)$$

and



$$\text{when } ET_{oi\,(M)} > 45 \text{ mm month}^{-1} \quad \text{then} \quad Fm_i=1 \text{ else } =0 \qquad (7b)$$

$$ET_{os\,adj.i} = ET_{os\,i} \cdot Fr_i \cdot Fm_i \qquad (7c)$$

$$YET_{os\,adj.} = \sum_{i=1}^{12}\left(ET_{os\,adj.i}\right) \qquad (7d)$$

$$C = \sum_{i=1}^{12}\left(\frac{ET_{os\,adj.i}}{YET_{os\,adj.}}\cdot C_i\right) \qquad (7e)$$

where $ET_{os\,i}$: is the mean monthly value of $ET_o$ (mm month$^{-1}$) obtained from the ASCE-short method, $ET_{oi\,(M)}$: is the mean monthly value of $ET_o$ (mm month$^{-1}$) obtained from the M method (M is either P-T or H-S), $Fr_i$: is the filter function of reference method with values 0 or 1, $Fm_i$: is the filter function of M method with values 0 or 1, $ET_{os\,adj.\,i}$: is the adjusted mean monthly value of $ET_o$ from ASCE-short method which becomes 0 when $Fr_i$ or $Fm_i$ is 0, $YET_{os\,adj.}$: is the sum of the monthly adjusted $ET_{os\,adj.\,i}$ values, $C_i$: is the mean monthly coefficient of M method (i.e. $a_{pt}$ or $c_{hs2}$) calibrated based on ASCE-short method (results from the previous step of analysis), $C$: is the partial weighted average (p.w.a.) of the mean monthly coefficients of M method (i.e. $a_{pt}$ or $c_{hs2}$) for short reference crop and $i$: is month.

For estimating the p.w.a. of mean monthly $a_{pt}$ and $c_{hs2}$ for tall reference crop, the same procedure of Eqs.7 was followed using the mean monthly values of $ET_o$ from ASCE-tall to estimate the $Fr_i$ values in Eq.7a, while the adjusted values of ASCE-tall were also used in Eqs.7c,d,e. For $C_i$ values in Eq.7e, the mean monthly values of $a_{pt}$ or $c_{hs2}$ based on ASCE-tall method were used. Even though the mean monthly values of ASCE-tall are generally higher from ASCE-short, the threshold of 45 mm month$^{-1}$ in Eqs.7a,b remained the same since it was observed that the difference between ASCE-short and ASCE-tall is very small when $ET_{os\,i}$ falls below ~50 mm month$^{-1}$.

A similar procedure (using the set of Eqs.7) was also followed to obtain the p.w.a. of mean monthly $K_{RS}$ of H-S method for $R_s$ estimations. The $Fr_i$ values in Eq.7a were estimated using as reference the mean monthly $R_s$ values of Sheffield et al. (2006) which were also used after adjustment in Eqs.7c,d,e. The $Fm_i$ values in Eq.7b were estimated using the respective $R_s$ values of the typical H-S with $K_{RS}=0.17$. For $C_i$ values in Eq.7e, the mean monthly values of $K_{RS}$ calibrated based on $R_s$ values of Sheffield et al. (2006) were used. The threshold used for adjusting $R_s$ values in Eqs.7a,b was set equal to 3.61 MJ m$^{-2}$ d$^{-1}$ (~110 MJ m$^2$ month$^{-1}$) which is equivalent to 45 mm month$^{-1}$ (conversion of mm month$^{-1}$ to MJ m$^{-2}$ month$^{-1}$ was performed after multiplying with $\lambda=2.45$ MJ kg$^{-1}$). The threshold for $R_s$ adjustment was tested before its use and it was found that works satisfactorily excluding unrealistic monthly values of $K_{RS}$ (as unrealistic values were considered those values which were at least one order of magnitude larger or smaller from the typical value of $K_{RS}=0.17$).

*Step 3: Use of stations for the validation of the p.w.a. coefficients of P-T and H-S methods*

Stations from two databases (California Irrigation Management System - CIMIS database,



http://www.cimis.water.ca.gov, and Australian Government – Bureau of Meteorology AGBM database, http://www.bom.gov.au/), were used in this study in order to validate the p.w.a. coefficients of P-T and H-S methods. The first database includes stations from California-USA and it was selected for the following reasons: a) it has been used as a basis for the development of Hargreaves-Samani method (Hargreaves and Allen, 2002) and CIMIS method (Snyder and Pruitt, 1985, Snyder and Pruitt, 1992) and b) it provides a dense and descriptive network of stations for a specific region which combines coastal, plain, mountain and desert environments (Table 1, Fig.1a). The second database includes stations from Australia and it was selected because the stations network covers a large territory with large variety of climate classes (Table 1, Fig.1b). For the stations of AGBM database, the selection of stations was performed in such way in order to cover all the possible existing Köppen climatic types and altitude ranges. In total, 140 stations were used, 60 stations were selected from CIMIS and 80 stations from the AGBM that have at least 15 years of observations (some stations, that do not follow this rule, were selected due to their special climate Köppen class or the high altitude of their location). In the stations data, observations from years after 2000 up to 2016 (when they were available) were included in order to show that the new revised coefficients are applicable for recent periods.

**[FIGURE 1]**

**[TABLE 1]**

In the case of CIMIS stations, the monthly data for all climatic parameters were obtained including $ET_o$ estimations using the CIMIS method (Snyder and Pruitt, 1985, Snyder and Pruitt, 1992) but they require quality control before their use. Quality control signs are provided for all climatic data, indicating extreme values, while possible errors are flagged but they are not automatically excluded. For this reason, the user should consider the signs in order to prepare a robust dataset. For this study, proper control was performed and very extreme or erroneous monthly values were excluded (excluded values were less than 1‰ of the total values of all stations and parameters). The final clean dataset was subjected to a secondary but indirect quality control through the comparison between the estimated mean monthly values of $ET_o$ of ASCE-short method (Eq.1) using the clean climatic data of all USA-CA stations versus their respective mean monthly $ET_o$ values given by CIMIS database (linear regression result between mean monthly values for $n$ obs.=12×60=720: y=0.994x-1.07 with $R^2$=0.98) (see Fig.S1 in the supplementary material section S1).

The validation procedure was performed using the data of the stations in Table 1 by comparing the mean monthly values of $ET_o$ derived by the P-T (Eq.2) and H-S (Eq.4b,c) methods with the original $a_{pt}$ and $c_{hs2}$ coefficients and with the re-adjusted ones versus the ASCE method for short reference crop (Eq.1). The same procedure was also performed for the new $a_{pt}$ and $c_{hs2}$ coefficients for the tall reference crop and for the re-adjusted coefficient $K_{RS}$ in the radiation formula of H-S (Eq.3).

The following statistical criteria were used in the validation procedure: coefficient of determination $R^2$, modified coefficient of determination $bR^2$ based on y=$b$x (Krause et al., 2005), mean absolute error $MAE$, root mean square error $RMSE$, percent bias $PBIAS\%$, Nash-Sutcliffe efficiency $NSE$ (Nash and Sutcliffe, 1970), index of agreement $d$ (Willmott, 1981) and Kling-Gupta efficiency (Gupta et al., 2009). The criteria were calculated using the package {HydroGOF} in R



language (Zambrano-Bigiarini, 2015, see the package manual for formulas). Due to the use of a large number of statistical criteria between different methods, the final ranking was also verified using Taylor diagrams (Taylor, 2001). These diagrams combine the *RMSE*, Pearson *R* and standard deviations of model predictions versus observations in order to provide a ranking of the overall performance of models. The standard deviations are used in order to compare the amplitude of variation between simulated and observed values. The Taylor diagrams were developed using the {Plotrix} package in R language (Lemon et al., 2016).

One of the major problems which is usually identified in the development of climate data grids is that their accuracy is reduced with the altitude due to the respective reduction of available stations (Hijmans et al., 2005; Sheffield et al., 2006). For this reason, additional comparative analysis between the methods was also performed after splitting the validation dataset of stations in three parts based on the altitude of their location: ≤110 m, 110-500 m and >500 m. The respective number of stations for each altitude range was 70 (50%), 56 (40%), and 14 (10%). For this analysis only the *RMSE* criterion was used due to the large number of comparisons.

Finally, analysis of pixel resolution effects on the predictive accuracy of the derived maps of $a_{pt}$, $c_{hs2}$ and $K_{RS}$ coefficients was performed based on the five different pixel resolutions given in the database (30 arc-sec, 2.5 arc-min, 5 arc-min, 10 arc-min and 30 arc-min~0.5 deg.) For this analysis, only the *RMSE* criterion was used due to the large number of comparisons.

## 3. Results

### 3.1 Comparative analysis between the typical $ET_o$ formulas of ASCE, P-T and H-S and error analysis of H-S radiation formula

The global maps of mean monthly $ET_o$ at 30 arc-sec resolution for the period 1950-2000 using the methods of ASCE (Eq.1) for both reference crops (ASCE-short and ASCE-tall), the typical P-T method (Eq.2) for $a_{pt}$=1.26 and the typical H-S method (Eq.4b) for $c_{hs2} = 0.0023$ were developed. The respective mean annual $ET_o$ maps are given in Fig.2a,b,c and d, respectively. Similarly, the mean annual $R_s$ values provided by Sheffield et al. (2006) and the $R_s$ values estimated by the H-S formula (Eq.3 with $K_{RS}$=0.17) are given in Fig.3a,b, respectively.

[FIGURE 2]

[FIGURE 3]

The *MAD*% maps of ASCE-tall, typical P-T and typical H-S methods versus ASCE-short are given in Fig.4a,b,c, respectively, while in Fig.4d is also given the *MAD*% of the typical solar radiation formula of H-S versus the $R_s$ values given by Sheffield et al. (2006). The percentage globe coverage for different classes of *MAD*% and the $R^2$ and *RMSD* based on respective comparisons of the mean monthly values of $ET_o$ and $R_s$ methods are given in Table 2.

[FIGURE 4]

[TABLE 2]





The case of *MAD*% between the $ET_o$ methods of ASCE-tall and ASCE-short (Fig.4a and Table 2) indicates that there is a 21.3% of map coverage in the *MAD*% class of ±10% where the effects of reference crop type are significantly minimized (Table 2). These territories include the regions of tropical rainforests in Latin America, central Africa and Indonesia, regions of large mountain formations-ranges of high altitude and regions of taigas and tundras of North America and Asia (Fig.4a).

The low values of vapor pressure deficit is the main characteristic of these regions. On the contrary, the largest differences between the two reference crops appear in arid and semi-arid environments due to the high values of vapor pressure deficit. The high correlation $R^2$=0.96 (Table 2) between the monthly $ET_o$ values of ASCE-tall and ASCE-short suggests that it is feasible to develop reliable regional monthly coefficients or regression models, which can convert the $ET_o$ estimations from short to tall reference crop especially when the $ET_o$ of short reference crop is estimated with a method of reduced parameters

(e.g. P-T or H-S) (a paradigm has been presented by Aschonitis et al., 2012).

Even though *MAD*%, $R^2$ and *RMSD* for the typical P-T and H-S methods (Fig.4b and c, Table 2) indicate a better performance of the second one to approximate the ASCE-short in a global scale, both methods seem to be equally valuable because their proximity to ASCE-short is maximized at relatively different climatic regions. This is indicated by the difference between the absolute *MAD*% values (*DMAD*) (Eq.6) of the P-T and H-S methods (Fig.5a). The interpretation of

15 Fig.5a was performed using as a base the major climatic groups CGs of the Köppen-Geiger climate map obtained by Peel et al. (2007) (Fig.5b). The spatial extent of the major CGs of the Köppen-Geiger climate classification and the percentage prevalence of P-T versus H-S in the CGs based on the *DMAD* values are given in Table 3. According to Table 3 and Figs.5a,b, the H-S method prevails in regions of the B group (arid and semi-arid) and E group (polar/alpine/tundras), while the P-T method prevails in the regions of the A group (tropical/megathermal), C group (temperate/mesothermal climates)

and D group (continental/microthermal). Even though the P-T method seems to be more powerful in more climatic zones, in reality the H-S method prevails in the 50.5% of the regions while P-T in the 46.5% (the remaining proportion of 3.0% mainly corresponds to inner Greenland and very high mountain areas with annual $ET_o$=0 or to regions where both methods gave equal results).

**[FIGURE 5]**

**[TABLE 3]**

The spatial variation of *MAD*% for the case of $R_s$ estimations using the typical solar radiation formula of H-S for $K_{RS}$=0.17 (Eq.3) versus the mean annual $R_s$ values of Sheffield et al. (2006) is given in Fig.4d. It is indicative that the 59.6% of the territories are included in the *MAD*% range ±10%, while the 92.5% is included in the range between ±25% (Table 2). Significant deviations of $R_s$ estimations using the typical H-S method appear mostly in the region of Greenland (Fig.4d). The

30 values of $R^2$ και *RMSD* (Table 2) indicate a good performance of the method in the case of monthly estimations. The overall results indicate that the use of the typical value $K_{RS}$=0.17 can provide satisfactory indirect estimations of $R_s$ for the most part of the world only by the use of temperature data.



## 3.2 Partial weighted averages of mean monthly $a_{pt}$, $c_{hs2}$ and $K_{RS}$

The p.w.a. of mean monthly $a_{pt}$ and $c_{hs2}$ for short (p.w.a.s.) and tall (p.w.a.t.) reference crop were derived from the application of Eqs.7 and they are given in Fig.6, while the respective p.w.a. of mean monthly $K_{RS}$ values are given in Fig.7. The global means of p.w.a. of $a_{pt}$ and $c_{hs2}$ for short reference crop (presented below each map of Fig.6a,c), and the global

mean of p.w.a. of $K_{RS}$ values for $R_s$ (presented below Fig.7) approximate to the typical values of $a_{pt}$=1.26, $c_{hs2}$=0.0023 and $K_{RS}$=0.17, respectively (less than 5% difference).

**[FIGURE 6]**

**[FIGURE 7]**

As regards the spatial variation of $a_{pt}$ for short reference crop (Fig.6a), extreme deviations from the 1.26 value are

observed in the arid-desert environments with values ranging between ~2.0-3.0 (due to extremely high vapor pressure deficit) while the extremely cold environments present the lower values <0.8. Interesting cases are the alpine-tundra and extreme humid tropical environments, which present similar values between ~0.8-1.0 due to the low values of vapor pressure deficit. As concern the respective spatial variation of $a_{pt}$ for tall reference crop (Fig.6b), its value is increased proportionally with ASCE-tall $ET_o$ in comparison to the respective values of $a_{pt}$ for short reference crop.

As regards the spatial variation of $c_{hs2}$ for short reference crop (Fig.6c), extreme deviations from the 0.0023 value are observed again in the arid-desert environments with values ranging between ~0.0028-0.0038 (due to extremely high vapor pressure deficit) while the extremely cold environments present the lower values <0.0014. Similarities appear again in the case of alpine-tundra and extreme humid tropical environments, which present similar values between ~0.0014-0.0018 due to the low values of vapor pressure deficit. As concern the respective spatial variation of $c_{hs2}$ for tall reference crop (Fig.6d),

its value is increased proportionally with ASCE-tall $ET_o$ in comparison to the respective values of $c_{hs2}$ for short reference crop.

In the case of $K_{RS}$ (Fig.7), extreme deviations from the 0.17 value were observed in Greenland with values ranging above 0.21. The spatial variation of $K_{RS}$ does not follow a specific pattern in relation to climate zones while in many cases it was observed an increasing trend of its values closer to the coastlines. This increase is more evident in island complexes,

Africa, India, south Europe coastlines etc (Fig.7). Additional observations about the effect of distance from the coastlines $Dc$ is given in the discussion section.

The extremely high values of $a_{pt}$ and $c_{hs2}$ for both short and tall reference crop, which are described by the seventh class of the maps in Figs.6 (class of black colour), appear in the South Sandwich Islands (26° 40′ W, 57° 10′ S) (black bullet in Figs.6a,b,c,d). Even though the environment of these islands is considered polar, the mean monthly maximum

temperatures during the warm season are between 0-10 °C, while the wind speed is in the range of 5-7 m s$^{-1}$ all the year. Even though the high values of the coefficients may be an artefact, the high wind speed during the warm season may justify them since typical P-T and H-S methods cannot describe such conditions.




The range of values for $a_{pt}$, $c_{hs2}$ and $K_{RS}$ given in Figs.6 and 7 correspond to the 30 arc-sec resolution. The respective ranges for each coefficient using the coarser resolution maps of 0.5 degrees are the following: $a_{pt}$ for short ref. crop (0.52-2.96), $a_{pt}$ for tall ref. crop (0.52-4.55), $c_{hs2}$ for short ref. crop (0.00105-0.00427), $c_{hs2}$ for tall ref. crop (0.00101-0.00641) and $K_{RS}$ (0.109-0.381).

### 3.3 Validation of the re-adjusted $a_{pt}$, $c_{hs2}$ and $K_{RS}$ coefficients

The validation of the re-adjusted $a_{pt}$, $c_{hs2}$ coefficients for $ET_o$ estimations (for both reference crops) and the $K_{RS}$ coefficient for $R_s$ was performed taking into account the mean monthly values of the climatic parameters of all stations from Table 1. The re-adjusted coefficients for each station obtained from the 30 arc-sec and 0.5 deg resolution maps are given in

Table S1 and S2, respectively (section S2 in supplementary material). It is indicative that the mean $c_{hs2}$ and $K_{RS}$ coefficients of all the USA-CA stations are equal to 0.0024 and 0.16, respectively, approximating the typical values used in the original formulas of H-S method, which were initially based on stations of the same territory. The comparison of different methods is described in the next paragraphs while the overall results of the statistical criteria for all the examined cases are given in Table 4.

**[TABLE 4]**

The No.1 case of Table 4 and the Figs.8a,b,c show the $ET_o$ (mm month$^{-1}$) comparisons between the ASCE-short values versus the values of the typical P-T method (with $a_{pt}$=1.26) and versus the values of P-T method using the partial weighted average $a_{pt}$ coefficient for short reference crop (p.w.a.s.) from 30 arc-sec and 0.5 deg resolution maps, respectively. The results of Fig.8a,b,c together with the results of the statistical criteria (Table 4) clearly indicate a significant

improvement of the P-T method using the p.w.a.s. values of $a_{pt}$ coefficient for short reference crop. The p.w.a.s. $a_{pt}$ coefficients from the 30 arc-sec resolution map showed slightly better performance in comparison to the respective coefficients of 0.5 deg resolution map (Table 4).

                                     **[FIGURE 8]**

The No.2 case of Table 4 and the Figs.9a,b,c,d show the $ET_o$ (mm month$^{-1}$) comparisons between the ASCE-short

values versus the values of the typical H-S method (with $c_{hs2}$=0.0023), versus the values of H-S$_{DRAL}$ method and versus the values of H-S method using the partial weighted average $c_{hs2}$ coefficient for short reference crop (p.w.a.s.) from 30 arc-sec and 0.5 deg resolution maps, respectively. The results of Fig.9a,b,c,d together with the results of the statistical criteria (Table 4) clearly indicate that the H-S method using the p.w.a.s. values of $c_{hs2}$ outperforms in comparison to the typical H-S and H-S$_{DRAL}$ methods. The p.w.a.s. $a_{pt}$ coefficients from the 0.5 deg resolution map showed slightly better performance in

comparison to the respective coefficients of 30 arc-sec resolution map (Table 4).

                                     **[FIGURE 9]**

The No.3 case of Table 4 and the Figs.10a,b,c,d show the $ET_o$ (mm month$^{-1}$) comparisons between the ASCE-tall values versus the values of P-T method and H-S method using the partial weighted average $a_{pt}$ and $c_{hs2}$ coefficients for tall





reference crop (p.w.a.t.) from the 30 arc-sec and 0.5 deg resolution maps, respectively. Since there were not previous typical formulas of P-T and H-S for tall reference crop, the comparison is restricted between the two methods and the different map resolutions. The results of Fig.10a,b,c,d together with the results of the statistical criteria (Table 4) indicate a better performance of the H-S(with $c_{hs2}$=p.w.a.t.) while the performance of P-T (with $a_{pt}$=p.w.a.t.) is also satisfactory. The

prevalence of the first method can be attributed to the fact the ~87% of stations from Table 1 are located in territories with negative *DMAD* values (Fig.5a), which gives an advantage to H-S method to outperform. This may also be the reason for the better performance of the typical H-S (with $c_{hs2}$=0.0023) in comparison to the typical P-T(with $a_{pt}$=1.26) for short reference crop (Table 4). As regards the comparison between different resolution maps, the 30 arc-sec resolution map showed slightly better performance compared to the 0.5 deg in the case of P-T while the opposite was observed in the case of H-S for tall

reference crop.

[FIGURE 10]

The No.4 case of Table 4 and the Figs.11a,b,c show the comparisons between the $R_s$ (MJ m$^{-2}$ d$^{-1}$) of Sheffield et al. (2006) versus the respective values of the H-S formula for radiation $R_s$ (Eq.3) using $K_{RS}$= 0.17 and $K_{RS}$= p.w.a. from 30 arc-sec and 0.5 deg resolution maps, respectively. The results of Fig.11a,b,c together with the results of the statistical criteria

(Table 4) indicate a better performance of the H-S $R_s$ (with $K_{RS}$= p.w.a.) even though the performance of the typical H-S $R_s$ (with $K_{RS}$= 0.17) is also satisfactory. As regards the comparison between different resolution maps in the case of p.w.a. $K_{RS}$ coefficients, the 30 arc-sec resolution map showed slightly better performance compared to the 0.5 deg.

[FIGURE 11]

Despite the fact that Table 4 and Figs.8-11 provide clear evidence that the new re-adjusted coefficients provide a

better performance in comparison to the typical coefficients, it is necessary to provide a general aspect of their performance using a combined criterion such as the Taylor diagram. The comparison of all cases related to $ET_o$ for short and tall reference crop are given in Fig.12a and 12b, respectively, while the cases related to $R_s$ estimations are given in Fig.12c.

Simulated patterns that outperform lie nearest the point marked "observed" on the x-axis (Figs.12a,b,c). These models provide estimations, which have relatively high $R$ and low *RMSE* in comparison to the observed values. Models lying on the

dashed arc have the correct standard deviation (which indicates that the pattern variations are of the right amplitude). According to Fig.12a (comparisons for $ET_o$ of short reference crop), it is observed that the P-T and H-S models, which use the p.w.a.s. coefficients, are described by higher $R$ and lower *RMSE* in comparison to the other typical models. The comparison between the P-T (with $a_{pt}$=p.w.a.s.) and H-S (with $c_{hs2}$=p.w.a.s.), showed that the first one presents a more similar variation to the observed values (through st.dev.) while the second one presents better correlation $R$ and lower *RMSE*.

The same pattern is also observed in the comparison between the P-T (with $a_{pt}$=p.w.a.t.) and H-S (with $c_{hs2}$=p.w.a.t.) for the $ET_o$ of the tall reference crop (Fig.12b). For the case of $R_s$ estimations, it is observed that the H-S ($K_{RS}$= p.w.a.) model not only improves the $R$ and *RMSE* in comparison to H-S ($K_{RS}$= 0.17) but also provides a more similar variation to the observed values (Fig.12c). As concern the performance of the models, which use the p.w.a. values from the coarser resolution (0.5



deg) maps, it was observed that they also outperform in comparison to the typical methods with extremely small differences in comparison to the finer resolution maps of 30 arc-sec.

**[FIGURE 12]**

The validation analysis (based on *RMSE*) after separating the stations in the three datasets of different altitudinal range is given in Fig.13. Fig.13a shows the response of methods related to $ET_o$ estimations of short reference crop. Significant observations based on Fig.13a are the following:

a) the P-T ($a_{pt}$=p.w.a.s.) and H-S ($c_{hs2}$=p.w.a.s.) of both resolutions outperform in comparison to their respective typical methods at all altitudinal ranges.

b) the accuracy of P-T ($a_{pt}$=p.w.a.s.) and H-S ($c_{hs2}$=p.w.a.s.) is reduced in the stations of the second altitudinal zone but it is increased in the third zone when all the rest typical methods show a trend of accuracy reduction with altitude. It is indicative that the *RMSE* of P-T ($a_{pt}$=p.w.a.s.) and H-S ($c_{hs2}$=p.w.a.s.) is at least 40% lower from the *RMSE* of their respective typical methods in the high altitude zone highlighting the robustness of the new revised coefficients to describe the $ET_o$ conditions of higher altitude areas. The increase in the *RMSE* of P-T ($a_{pt}$=p.w.a.s.) and H-S ($c_{hs2}$=p.w.a.s.) in the stations of the second altitudinal zone is probably related to the fact that these stations are located to regions of higher surface slope. The mean ± standard error of % slope of the stations belonging to the groups of <110, 110-500 and <500 m altitude ranges are 0.48±0.01%, 1.24±0.03% and 0.70±0.06%, respectively. The possible relation between the surface slope of stations' location and the *RMSE* of $ET_o$ estimations using the re-adjusted coefficients was not verified when correlation analysis was performed using directly the values of slope and RMSE of each station (all correlations showed values of |R|<0.1).

c) The *RMSE* comparison between p.w.a.s. coefficients obtained from different resolution maps (30 arc-sec and 0.5 deg) showed that in the case of H-S ($c_{hs2}$=p.w.a.s.) the differences are negligible, while in the case of P-T ($a_{pt}$=p.w.a.s.) the 30 arc-sec resolution gave slightly better results in the high altitude zone.

d) the typical H-S ($c_{hs2}$=0.0023) showed better performance from the modified H-S$_{DRAL}$ in the low altitudinal zone which consists of 50% stations, while in the third altitudinal zone they present almost similar performance.

Fig.13b shows the response of P-T ($a_{pt}$=p.w.a.t.) and H-S ($c_{hs2}$=p.w.a.t.) methods to assess the $ET_o$ of tall reference crop. The accuracy of the two methods is reduced in the second altitudinal zone but it is increased in the third zone following similar patterns with the P-T ($a_{pt}$=p.w.a.s.) and H-S ($c_{hs2}$=p.w.a.s.) for short reference crop. The p.w.a.t. coefficients obtained from the high resolution maps (30 arc-sec) showed slightly better performance, compared to the 0.5 deg maps, in the two first altitudinal zones, while the opposite was observed in the third high altitude zone.

Fig.13c shows the response of H-S ($K_{RS}$= 0.17) and H-S ($K_{RS}$= p.w.a.) to assess the $R_s$. It is indicative that H-S ($K_{RS}$= p.w.a.) outperforms at all altitudinal zones. The p.w.a. coefficients obtained from the high resolution maps (30 arc-sec) showed better performance, compared to the 0.5 deg maps, at all altitudinal zones. The accuracy of p.w.a. of 30 arc-sec is increased with altitude and especially in the third zone where the *RMSE* is almost half from the one of H-S ($K_{RS}$= 0.17).

**[FIGURE 13]**

Finally, analysis of pixel resolution effects on the predictive accuracy of the derived maps of $a_{pt}$, $c_{hs2}$ and $K_{RS}$





coefficients was performed based on the five different pixel resolutions given in the database (30 arc-sec, 2.5 arc-min, 5 arc-min, 10 arc-min and 0.5 deg). For this analysis, the *RMSE* criterion was estimated separately for each station in order to estimate the respective standard deviation of *RMSE* for each method and for each pixel resolution (Fig.14). The standard deviation of *RMSE* is important since it highlights the variation within the *RMSE* average. For the case of $a_{pt}$ (p.w.a.s.)

(Fig.14a), $c_{hs2}$ (p.w.a.s.) (Fig.14b), $a_{pt}$ (p.w.a.t.) (Fig.14c) and $K_{RS}$ (p.w.a.) (Fig.14e), the best *RMSE* results are observed in the maps of 2.5 and 5 arc-min resolutions These two resolutions present also the lower st.dev. of *RMSE*. In the case of $c_{hs2}$ (p.w.a.t.) (Fig.14d), the best *RMSE* is observed in the 0.5 deg resolution but with a high value of its standard deviation. If we exclude the 0.5 deg resolution from this case, again the 2.5 and 5 arc-min resolutions present the lower *RMSE* and the lower st.dev. (the better *RMSE* performance of 0.5 deg in this case was probably a lucky shot). Significant observations related to

pixel resolution effects on the coefficients are provided in the discussion section.

**[FIGURE 14]**

## 4. Discussion

The analysis presented in this study passed through various stages before selecting the p.w.a. form of the coefficients. Some steps in the preliminary analysis were to analyze a) the strength of the derived mean monthly coefficients versus the

15 p.w.a. coefficients and versus the coefficients of the typical methods, b) the strength of potential global models based on minimum data (e.g. as Eq.4c) for adjusting regional monthly coefficients and c) the different forms of averages (e.g. mean, mode, median, geometric mean, harmonic mean etc) for deriving annual coefficients.

The case of mean monthly coefficients was examined in the same way like the p.w.a. coefficients (results not given). The results showed that the assessment of annual $ET_o$ and seasonal $ET_o$ during the warm season using the mean monthly

coefficients outperforms in comparison to the typical methods but their predictive strength was not as good as p.w.a. coefficients. Further analysis was also performed using different time intervals for calculating seasonal averages or weighting averages of the coefficients (e.g. 3-months averages or 6-months averages). These results again showed a very high predictive accuracy during the warm periods smoothing the errors observed by the use of mean monthly coefficients during the cold periods. For a global scale analysis, the p.w.a. coefficients showed more robust performance even from the

seasonal coefficients probably because they counterbalance intra-annual/intra-seasonal climatic variability. The case of mean monthly coefficients was also analyzed through modelling approaches trying to investigate the predictive strength of various formulas at global scale using limited/easily accessible data (e.g. intra-annual characteristics of precipitation and temperature, latitude, longitude, altitude, distance from the sea, climatic type etc). The predictive accuracy of such formulas at global scale was found better in comparison to the typical methods but again the p.w.a. coefficients outperformed showing

that the effects of special regional climatic characteristics related to wind speed, relative humidity etc (which usually are not considered in the models due data limitation) cannot be captured by a global model. It was also observed that a regional mean value of $a_{pt}$, $c_{hs2}$ and $K_{RS}$ coefficients derived by p.w.a. values may present even more robust performance because it probably counterbalances both the intra-annual climatic but also the spatial climatic variability. This can be attributed to the





fact that the climatic patterns (especially precipitation) may show higher variation in specific positions of a region while the respective variation is lower when considering the general patterns of the overall region. The aforementioned observation was verified by the application of H-S method for $ET_o$ of short reference crop when we used only the stations of California (this application includes also the H-S$_{DRAL}$ model in order to show the performance of a global model in a regional case).

The application showed that the use of the mean value of $c_{hs2}$=0.0024 derived by all the p.w.a. coefficients of California stations (Table S1) (this value also approximates the typical value of 0.0023) gives better results in comparison to the p.w.a.s. $c_{hs2}$ coefficients of each station and the H-S$_{DRAL}$ model (Fig.S2 and Table S3, section S3 in supplementary material). It is indicative that the H-S$_{DRAL}$ model showed the lower accuracy among the three cases (Fig.S2 and Table S3). The aforementioned observations suggest that a robust territorial segmentation based on general topographic characteristics (e.g.

altitude, slope, latitude and longitude, distance from the coastline etc) and general climatic characteristics (e.g. Köppen class, general precipitation and temperature patterns) can provide a proper zonation of large territories for deriving very robust mean values of $a_{pt}$, $c_{hs2}$ and $K_{RS}$ coefficients using the respective p.w.a. values of each zone. A method for robust zonation based on grids of mean monthly precipitation and temperature (or even mean monthly $ET_o$) can easily be performed using cluster analysis in GIS environment (Demertzi et al., 2014; Aschonitis et al., 2016).

As regards the use of weighted annual average (w.a.) of the mean monthly coefficients instead of other forms of averages (e.g. mean, mode, median, geometric mean - g.m., harmonic mean – h.m. etc), detailed analysis was performed in order to investigate their predictive strength (results not given). Using in all cases the same rule given in Eqs.7 for excluding values when $ET_o$ and $R_s$ fall below a specific threshold (partial form of all cases), the following predictive ranking was observed w.a.>h.m.>g.m.> median≈mode≈average. The w.a. outperformed in all cases because it is the only form which

considers the amplitude of the parameter under investigation giving more weight to the monthly coefficients which are related to the larger values of the parameter.

Special attention was also given in the case of $K_{RS}$ coefficient for estimating $R_s$. Even though the spatial variation of p.w.a. $a_{pt}$ and $c_{hs2}$ coefficients at global scale was linked to general climatic characteristics (Fig.5), the respective variation of p.w.a. $K_{RS}$ coefficient could not clearly be linked with a specific climatic or topographic characteristic. The only observed

dependence, which showed some relevance to the spatial variation of $K_{RS}$, was a relatively negative correlation with the distance from the coastline $Dc$. This observed dependence can be only used as a general observation and not as a basis for applying in general the empirical rule of Allen et al. (1998) ($K_{RS}$=0.16 for "interior" and $K_{RS}$ =0.19 for "coastal" locations). The inapplicability of the aforementioned rule was also indicated by Samani (2000) and it is verified by the analysis presented in Fig.S3a (section S4 in supplementary material). Fig.S3a shows a relatively negative correlation between $K_{RS}$ and

$Dc$ (for $Dc$<500 km) but also shows an extremely high variability of $K_{RS}$ close to the coastlines where $K_{RS}$ values are not necessarily higher in comparison to the values observed in the interior regions. The observed lower variability of $K_{RS}$ at interior regions is probably related to the fact that coastlines are more affected by oceanic-climatic phenomena, which anyway present high spatial variability at global scale. Samani (2000) also observed that monthly $K_{RS}$ values may be influenced by the difference between monthly maximum and minimum temperature $TD$, which is also related to humidity



characteristics (vapor pressure deficit). This effect was also investigated through correlation between the mean monthly $K_{RS}$ coefficients and the mean monthly $TD$ values of stations data (Fig.S3b, section S4 in supplementary material). The results showed that the hypothesis related to the effect of $TD$ on $K_{RS}$ may be stronger in comparison to the effect of $Dc$, but again the variation of $K_{RS}$ is extremely large in the $TD$ range between 8-15 °C (Fig.S3b) not allowing secure conclusions for a global

scale application (the result of Fig.S3b are based only on the stations of Table 1, and for this reason the variation is expected much larger in a global scale).

For the case of $ET_o$ for tall reference crop, the general observations in the behaviour of the p.w.a.t. $a_{pt}$ and $c_{hs2}$ coefficients follow the respective observations which were provided for the coefficients of the short reference crop. A very interesting observation about the tall reference crop was made based on the results of $MAD\%$ map (Fig.4a). In the $MAD\%$

class of ±10% of Fig.4a, it was observed a peculiarity where the ~2% of map coverage shows small negative $MAD\%$ values indicating slightly larger annual values of ASCE-short in comparison to ASCE-tall. This occurred in regions of extremely small vapor deficit (areas of very high altitude, either of very cold, or extremely humid conditions scattered around the world) and it is a peculiarity of Eq.1 and probably an artefact. This happens because the second term of the nominator in Eq.1 (which includes the vapor deficit term and the $C_n$ coefficient) approximates to 0 (since $e_s$-$e_a$ is close to 0) when the

vapor deficit is extremely small while the denominator of Eq.1 is always larger in ASCE-tall in comparison to ASCE-short due to the difference in $C_d$ value (0.34 for short and 0.38 for tall reference crop).

As regards the prediction accuracy of different resolution maps of the $a_{pt}$, $c_{hs2}$ and $K_{RS}$ coefficients, it was observed a generally better performance of the 2.5 and 5 arc-min resolutions. The coefficients of all resolutions gave significantly better results in comparison to the typical methods but this was quite expected since the majority of stations are located in quite

large flat areas with small spatial variation of the coefficients in their surrounding environment (it is a common practice to establish meteorological stations in such locations). For visualizing the differences in the spatial variation of the coefficients between the finer and coarser resolution, we give as example the respective resolution maps of the coefficients for the case of California Figs.S4, S5 and S6 (section S5 in supplementary material) where it is clear that the finer resolution maps provide significant details about the variation of the coefficients in the steep mountainous regions, valleys and coastlines. The

examination of different resolution maps before the use of the coefficients for a specific location is prerequisite and should be performed taking into account mainly the general topography but also the spatial variation of the precipitation and temperature patterns of the surrounding environment. The 0.5 deg resolution maps is recommended only in cases when the user wants to assess $ET_o$ values for large flat territories. For analysis at watershed scale or at specific positions (e.g. stations which measure only temperature), the finer resolutions are more appropriate depending on the watershed size and

topography. Another characteristic of the different resolution maps, which should be considered, is the pixel position-orientation in relation to the location of the study area or the station especially in regions between mountains (anomalous topographies related to small-medium valleys). An example is given in Fig.S7 (section S5 in supplementary material), where we provide the location of the 5 pixels of different resolution which were used for the derivation of the coefficients for the Sanel Valley station (CA-46, Table 1, this station showed large differences of the coefficients between the 30 arc-sec and 0.5



deg maps). According to Fig.S7, the station is located ~43 km from the coastline, inside a narrow valley between mountains where the 0.5 deg resolution pixel can not be considered adequate to provide representative coefficients. The rest pixel resolutions are also on the upper right side of the 0.5 deg pixel and describe different aspects, since the 2.5 arc-min resolution pixel captures only lowland territories similar to the territory captured by the 30 arc-sec, the 5 arc-min pixel is

expanded only to the west at slightly higher altitude territories while the 10 arc-min pixel covers a larger area expanded in the southern lowland areas and in both steep sides of the valley. Apart from the coefficients of 0.5 deg resolution, which showed significantly larger *RMSE* values from the rest resolutions, the comparison of the rest showed small differences among them, while the most descriptive coefficients were those belonging to the pixel of the 10 arc-min resolution. This pixel covers a large territory of the lowland valley, plus steep areas from both sides, which suggests a better representation of

the mean conditions through minimization of the intra-annual and spatial variability effects observed in the position and the adjacent territories where the station is located. Taking into account all the aforementioned observations, a general guideline for cases like the Sanel Valley, is to use the coefficients of the pixel resolution which best describes the position and the surrounding environment. The use of larger resolution could also allow the use of the observed minimum and maximum values of the finer 30 arc-sec resolution pixels inside the larger pixel in order to perform uncertainty analysis.

## 5. Data availability

The datasets produced in this study have been archived in PANGAEA database (https://doi.pangaea.de/10.1594/PANGAEA.868808, https://doi.org/10.1594/PANGAEA.868808) and in ESRN-database which is currently supported by the University of Ferrara (Italy), Aristotle university of Thessaloniki (Greece) and 2nd

University of Naples (Italy) (http://www.esrn-database.org/gis-data.html or http://esrn-database.weebly.com/gis-data.html). A complete list of the datasets is provided in the supplementary material S6.

## 6. Conclusions

The study provided high resolution global grids of revised annual coefficients for the Priestley-Taylor (P-T) and Hargreaves-Samani (H-S) for estimating $ET_o$ (for two reference crops) and $R_s$. The analysis was based on global gridded

climatic data of the period 1950-2000. The method for deriving annual coefficients of P-T and H-S methods was based on partial weighted averages (p.w.a.) of their mean monthly values, which eliminate the effect of monthly coefficients that occur during periods where $ET_o$ and $R_s$ fall below a specific threshold. Five resolution global grids (30 arc-sec, 2.5, 5, 10 arc-min and 0.5 deg) of annual coefficients of each method were developed. The new coefficients were validated based on data from 140 stations located at various climatic zones of USA and Australia with expanded observations up to 2016. The

validation procedure for $ET_o$ estimations of short reference crop showed that the P-T and H-S methods with the new revised coefficients outperformed in comparison to the typical methods reducing at ~30-40% the *RMSE* of estimated $ET_o$ and $R_s$ values. Finally, a raster database was built consisting of: a) global maps of revised annual coefficients for the $ET_o$ methods of




P-T and H-S for both reference crops and the $R_s$ H-S formula, b) global maps which indicate the optimum locations for using the original P-T and H-S methods and their expected error based on ASCE method for short reference crop. The provision of the database aims to improve $ET_o$ and $R_s$ estimations which are used in hydrologic/climatic applications when climatic data are limited.

**Acknowledgements.** This study was performed in the context of two Post-Doctoral research studies by Dr.Vassilis Aschonitis financed by Ferrara University (Italy) and Aristotle University of Thessaloniki (Greece).

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



**TABLES**

**Table 1.** Meteorological stations from USA-California (CIMIS database) and Australia (AGBM database).

| No. | Code | Station | Country | Altitude (m) | Lat (Dec.deg.) | Long (Dec.Deg.) | Period | Köppen Class* |
|---|---|---|---|---|---|---|---|---|
| CA-1 | 006 | Davis | USA-CA | 18 | 38.54 | -121.78 | Sep 1982 - Aug 2016 | Csa |
| CA-2 | 002 | FivePoints | USA-CA | 87 | 36.34 | -120.11 | Jun 1982 - Aug 2016 | BWk |
| CA-3 | 005 | Shafter | USA-CA | 110 | 35.53 | -119.28 | Jun 1982 - Aug 2016 | BSk |
| CA-4 | 007 | Firebaugh/Telles | USA-CA | 56 | 36.85 | -120.59 | Sep 1982 - Aug 2016 | BWk |
| CA-5 | 012 | Durham | USA-CA | 40 | 39.61 | -121.82 | Oct 1982 - Aug 2016 | Csa |
| CA-6 | 008 | Gerber | USA-CA | 76 | 40.04 | -122.17 | Sep 1982 - Aug 2014 | Csa |
| CA-7 | 015 | Stratford | USA-CA | 59 | 36.16 | -119.85 | Nov 1982 - Aug 2016 | BSk |
| CA-8 | 019 | Castroville | USA-CA | 3 | 36.77 | -121.77 | Nov 1982 - Aug 2016 | Csb |
| CA-9 | 021 | Kettleman | USA-CA | 104 | 35.87 | -119.89 | Nov 1982 - Aug 2016 | BWk |
| CA-10 | 027 | Zamora | USA-CA | 15 | 38.81 | -121.91 | Dec 1982 - May 2006 | Csa |
| CA-11 | 030 | Nicolaus | USA-CA | 10 | 38.87 | -121.55 | Jan 1983 - Dec 2011 | Csa |
| CA-12 | 032 | Colusa | USA-CA | 17 | 39.23 | -122.02 | Jan 1983 - Aug 2016 | Csa |
| CA-13 | 033 | Visalia | USA-CA | 107 | 36.30 | -119.22 | Jan 1983 - Feb 2007 | BSk |
| CA-14 | 035 | Bishop | USA-CA | 1271 | 37.36 | -118.41 | Feb 1983 - Aug 2016 | BSk |
| CA-15 | 039 | Parlier | USA-CA | 103 | 36.60 | -119.50 | May 1983 - Aug 2016 | BSk |
| CA-16 | 041 | Calipatria/Mulberry | USA-CA | -34 | 33.04 | -115.42 | Jul 1983 - Aug 2016 | BWh |
| CA-17 | 043 | McArthur | USA-CA | 1009 | 41.06 | -121.46 | Dec 1983 - Aug 2016 | Csb |
| CA-18 | 044 | U.C.Riverside | USA-CA | 311 | 33.96 | -117.34 | Jun 1985 - Aug 2016 | BSk |
| CA-19 | 047 | Brentwood | USA-CA | 14 | 37.93 | -121.66 | Nov 1985 - Aug 2016 | Csb |
| CA-20 | 049 | Oceanside | USA-CA | 15 | 33.26 | -117.32 | Mar 1986 - Oct 2003 | BSk |
| CA-21 | 054 | Blackwells Corner | USA-CA | 215 | 35.65 | -119.96 | Oct 1986 - Aug 2016 | BWk |
| CA-22 | 056 | Los Banos | USA-CA | 29 | 37.10 | -120.75 | Jun 1988 - Aug 2016 | BSk |
| CA-23 | 061 | Orland | USA-CA | 60 | 39.69 | -122.15 | May 1987 - May 2010 | Csa |
| CA-24 | 062 | Temecula | USA-CA | 433 | 33.49 | -117.23 | Nov 1986 - Aug 2016 | BSk |
| CA-25 | 064 | Santa Ynez | USA-CA | 149 | 34.58 | -120.08 | Nov 1986 - Aug 2016 | Csb |
| CA-26 | 068 | Seeley | USA-CA | 12 | 32.76 | -115.73 | May 1987 - Aug 2016 | BWh |
| CA-27 | 070 | Manteca | USA-CA | 10 | 37.83 | -121.22 | Nov 1987 - Aug 2016 | BSk |
| CA-28 | 071 | Modesto | USA-CA | 11 | 37.65 | -121.19 | Jul 1987 - Aug 2016 | BSk |
| CA-29 | 077 | Oakville | USA-CA | 58 | 38.43 | -122.41 | Jan 1989 - Aug 2016 | Csb |
| CA-30 | 075 | Irvine | USA-CA | 125 | 33.69 | -117.72 | Oct 1987 - Aug 2016 | BSk |
| CA-31 | 078 | Pomona | USA-CA | 223 | 34.06 | -117.81 | Mar 1989 - Aug 2016 | Csa |
| CA-32 | 080 | Fresno State | USA-CA | 103 | 36.82 | -119.74 | Oct 1988 - Aug 2016 | BSk |
| CA-33 | 083 | Santa Rosa | USA-CA | 24 | 38.40 | -122.80 | Jan 1990 - Aug 2016 | Csb |
| CA-34 | 084 | Browns Valley | USA-CA | 287 | 39.25 | -121.32 | Apr 1989 - Aug 2016 | Csa |
| CA-35 | 085 | Hopland F.S. | USA-CA | 354 | 39.01 | -123.08 | Sep 1989 - Apr 2016 | Csa |
| CA-36 | 086 | Lindcove | USA-CA | 146 | 36.36 | -119.06 | May 1989 - Aug 2016 | Csa |
| CA-37 | 087 | Meloland | USA-CA | -15 | 32.81 | -115.45 | Dec 1989 - Aug 2016 | BWh |
| CA-38 | 088 | Cuyama | USA-CA | 698 | 34.94 | -119.67 | May 1989 - Aug 2016 | BSk |
| CA-39 | 091 | Tulelake F.S. | USA-CA | 1230 | 41.96 | -121.47 | Mar 1989 - Aug 2016 | Dsb |
| CA-40 | 092 | Kesterson | USA-CA | 23 | 37.23 | -120.88 | Oct 1989 - Aug 2016 | BSk |



| | | | | | | | | |
|---|---|---|---|---|---|---|---|---|
| CA-41 | 094 | Goletta foothills | USA-CA | 195 | 34.47 | -119.87 | Jul 1989 - Jul 2016 | Csb |
| CA-42 | 099 | Santa Monica | USA-CA | 104 | 34.04 | -118.48 | Dec 1992 - Aug 2016 | Csb |
| CA-43 | 103 | Windsor | USA-CA | 26 | 38.53 | -122.83 | Dec 1990 - Aug 2016 | Csb |
| CA-44 | 104 | De Laveaga | USA-CA | 91 | 37.00 | -122.00 | Sep 1990 - Aug 2016 | Csb |
| CA-45 | 105 | Westlands | USA-CA | 58 | 36.63 | -120.38 | Apr 1992 - Aug 2016 | BWk |
| CA-46 | 106 | Sanel Valley | USA-CA | 160 | 38.98 | -123.09 | Feb 1991 - Aug 2016 | Csa |
| CA-47 | 57 | Buntingville | USA-CA | 1221 | 40.29 | -120.43 | June 1986 - Sep 2016 | Dsb |
| CA-48 | 90 | Alturas | USA-CA | 1343 | 41.44 | -120.48 | Apr 1989 - Sep 2016 | Dsb |
| CA-49 | 151 | Ripley | USA-CA | 77 | 33.53 | -114.63 | Dec 1998 - Sep 2016 | BWh |
| CA-50 | 183 | Owens Lake North | USA-CA | 1123 | 36.49 | -117.92 | Dec 2002 - Sep 2016 | BWk |
| CA-51 | 147 | Otay Lake | USA-CA | 177 | 32.63 | -116.94 | Apr 1999 - Sep 2016 | Csb |
| CA-52 | 175 | Palo Verde II | USA-CA | 70 | 33.38 | -114.72 | Jan 2001 - Sep 2016 | BWh |
| CA-53 | 135 | Blynthe NE | USA-CA | 84 | 33.66 | -114.56 | Jan 1997 - Sep 2016 | BWh |
| CA-54 | 155 | Bryte | USA-CA | 12 | 38.60 | -121.54 | Dec 1998 - Sep 2016 | Csa |
| CA-55 | 159 | Monrovia | USA-CA | 181 | 34.15 | -117.99 | Oct 1999 - Sep 2016 | Csa |
| CA-56 | 161 | Patterson | USA-CA | 56 | 37.44 | -121.14 | Aug 1999 - Sep 2016 | BSk |
| CA-57 | 174 | Long Beach | USA-CA | 5 | 33.80 | -118.09 | Sep 2000 - Sep 2016 | Csb |
| CA-58 | 173 | Torrey Pines | USA-CA | 102 | 32.90 | -117.25 | Nov 2000 - Sep 2016 | Csa |
| CA-59 | 150 | Miramar | USA-CA | 136 | 32.89 | -117.14 | Apr 1999 - Sep 2016 | Csa |
| CA-60 | 153 | Escondido SPV | USA-CA | 119 | 33.08 | -116.98 | Feb 1999 - Sep 2016 | Csb |
| A-1 | 32040 | Townsville Aero | Australia | 4 | -19.25 | 146.77 | (1940/1996-2016)# | Aw |
| A-2 | 33307 | Woolshed | Australia | 556 | -19.42 | 146.54 | (1990/2003-2016) | Aw |
| A-3 | 2056 | Kununurra Aero | Australia | 44 | -15.78 | 128.71 | (1971/1990-2016) | BSh |
| A-4 | 35264 | Emerald | Australia | 189 | -23.57 | 148.18 | (1990/1998-2016) | BSh |
| A-5 | 24024 | Loxton R.C. | Australia | 30 | -34.44 | 140.6 | (1984/1998-2016) | BSk |
| A-6 | 74037 | Yanco AG.I. | Australia | 164 | -34.62 | 146.43 | (1957/1999-2016) | BSk |
| A-7 | 74258 | Deniliquin Airp.AWS | Australia | 94 | -35.56 | 144.95 | (1990/2003-2016) | BSk |
| A-8 | 75041 | Griffith Airp.AWS | Australia | 134 | -34.25 | 146.07 | (1958/1990-2016) | BSk |
| A-9 | 76031 | Mildura Airp. | Australia | 50 | -34.24 | 142.09 | (1946/1993-2016) | BSk |
| A-10 | 24048 | Renmark Apt.1 | Australia | 32 | -34.2 | 140.68 | (1990/2003-2016) | BWk |
| A-11 | 40082 | University of QLD G. | Australia | 89 | -27.54 | 152.34 | (1897/1995-2016) | Cfa |
| A-12 | 40922 | Kingaroy Airp. | Australia | 434 | -26.57 | 151.84 | (1990/2003-2016) | Cfa |
| A-13 | 41359 | Oakey Aero | Australia | 406 | -27.4 | 151.74 | (1970/1996-2016) | Cfa |
| A-14 | 41522 | Dalby Airp. | Australia | 344 | -27.16 | 151.26 | (1990/2006-2016) | Cfa |
| A-15 | 41525 | Warwick | Australia | 475 | -28.21 | 152.1 | (1990/2000-2016) | Cfa |
| A-16 | 41529 | Toowoomba Airp. | Australia | 641 | -27.54 | 151.91 | (1990/1997-2016) | Cfa |
| A-17 | 80091 | Kyabram | Australia | 105 | -36.34 | 145.06 | (1964/1990-2016) | Cfa |
| A-18 | 81049 | Tatura I.S.A. | Australia | 114 | -36.44 | 145.27 | (1942/1990-2016) | Cfa |
| A-19 | 81124 | Yarrawonga | Australia | 129 | -36.03 | 146.03 | (1990/2003-2016) | Cfa |
| A-20 | 81125 | Shepparton Airp. | Australia | 114 | -36.43 | 145.39 | (1990/1996-2016) | Cfa |
| A-21 | 41175 | Applethorpe | Australia | 872 | -28.62 | 151.95 | (1966/2006-2016) | Cfb |
| A-22 | 81123 | Bendigo Airp. | Australia | 208 | -36.74 | 144.33 | (1990/2004-2016) | Cfb |
| A-23 | 85072 | East sale Airp. | Australia | 5 | -38.12 | 147.13 | (1943/1996-2016) | Cfb |
| A-24 | 85279 | Bairnsdale Airp. | Australia | 49 | -37.88 | 147.57 | (1942/2003-2016) | Cfb |
| A-25 | 85280 | Morwell L.V.Airp. | Australia | 56 | -38.21 | 146.47 | (1984/1999-2016) | Cfb |





| | | | | | | | | |
|---|---|---|---|---|---|---|---|---|
| A-26 | 85296 | Mount Moornapa | Australia | 480 | -37.75 | 147.14 | (1990/2003-2016) | Cfb |
| A-27 | 90035 | Colac | Australia | 261 | -38.23 | 143.79 | (1990/2003-2016) | Cfb |
| A-28 | 9538 | Dwellingup | Australia | 267 | -32.71 | 116.06 | (1934/1990-2016) | Csb |
| A-29 | 9617 | Bridgetown | Australia | 179 | -33.95 | 116.13 | (1990/2003-2016) | Csb |
| A-30 | 23373 | Nuriootpa Pirsa | Australia | 275 | -34.48 | 139.01 | (1990/1996-2016) | Csb |
| A-31 | 26021 | Mount Gambier Aero | Australia | 63 | -37.75 | 140.77 | (1942/1994-2016) | Csb |
| A-32 | 26091 | Coonawarra | Australia | 57 | -37.29 | 140.83 | (1985/1990-2016) | Csb |
| A-33 | 66062 | Sydney (Obs.Hill) | Australia | 39 | -33.86 | 151.205 | (1858/1990-2016) | Cfb |
| A-34 | 33002 | Ayr DPI Res.St. | Australia | 17 | -19.62 | 147.38 | (1951/1994-2016) | Cwa |
| A-35 | 7176 | Newman Aero | Australia | 524 | -23.42 | 119.8 | (1971/2003-2016) | BWh |
| A-36 | 13017 | Giles | Australia | 598 | -25.03 | 128.3 | (1956/1990-2016) | BWh |
| A-37 | 11052 | Forrest | Australia | 159 | -30.85 | 128.11 | (1990/2003-2016) | BWh |
| A-38 | 11003 | Eucla | Australia | 93 | -31.68 | 128.9 | (1876/1995-2016) | BSk |
| A-39 | 12071 | Salmon Gums | Australia | 249 | -32.99 | 121.62 | (1932/2003-2016) | BSk |
| A-40 | 7045 | Meekatharra Airp. | Australia | 517 | -26.61 | 118.54 | (1944/1992-2016) | BWh |
| A-41 | 1025 | Doongan | Australia | 385 | -15.38 | 126.31 | (1988/1990-2016) | Aw |
| A-42 | 2012 | Halls Creek Airp. | Australia | 422 | -18.23 | 127.66 | (1944/1996-2016) | BSh |
| A-43 | 13015 | Carnegie | Australia | 448 | -25.8 | 122.98 | (1942/1990-2016) | BWh |
| A-44 | 3080 | Curtin Aero | Australia | 78 | -17.58 | 123.83 | (1990/2003-2016) | BSh |
| A-45 | 6022 | Gascoyne Junction | Australia | 144 | -25.05 | 115.21 | (1907/1990-2016) | BWh |
| A-46 | 9789 | Esperance | Australia | 25 | -33.83 | 121.89 | (1969/1990-2016) | Csb |
| A-47 | 91223 | Marrawah | Australia | 107 | -40.91 | 144.71 | (1971/1990-2016) | Cfb |
| A-48 | 18106 | Nullarbor | Australia | 64 | -31.45 | 130.9 | (1986/2006-2016) | BWk |
| A-49 | 16090 | Coober Pedy Airp. | Australia | 225 | -29.03 | 134.72 | (1990/2004-2016) | BWh |
| A-50 | 16085 | Marla Police St. | Australia | 323 | -27.3 | 133.62 | (1985/1990-2016) | BWh |
| A-51 | 13011 | Warburton Airfield | Australia | 459 | -26.13 | 126.58 | (1940/2003-2016) | BWh |
| A-52 | 15528 | Yuendumu | Australia | 667 | -22.26 | 131.8 | (1952/1990-2016) | BWh |
| A-53 | 15666 | Rabbit Flat | Australia | 340 | -20.18 | 130.01 | (1990/1996-2016) | BWh |
| A-54 | 14829 | Lajamanu Airp. | Australia | 316 | -18.33 | 130.64 | (1952/1990-2016) | BSh |
| A-55 | 15135 | Tennant Creek Airp. | Australia | 376 | -19.64 | 134.18 | (1969/1992-2016) | BSh |
| A-56 | 37010 | Camooweal Township | Australia | 231 | -19.92 | 138.12 | (1891/2003-2016) | BWh |
| A-57 | 14707 | Wollogorang | Australia | 60 | -17.21 | 137.95 | (1967/1990-2016) | Aw |
| A-58 | 14938 | Mango Farm | Australia | 15 | -13.74 | 130.68 | (1980/1990-2016) | Aw |
| A-59 | 69134 | Batemans Bay | Australia | 11 | -35.72 | 150.19 | (1985/1991-2016) | Cfb |
| A-60 | 14198 | Jabiru Airp. | Australia | 27 | -12.66 | 132.89 | (1971/1990-2016) | Aw |
| A-61 | 28008 | Lockhart River Airp. | Australia | 19 | -12.79 | 143.3 | (1956/2001-2016) | Am |
| A-62 | 34084 | Charters Towers Airp. | Australia | 290 | -20.05 | 146.27 | (1990/1992-2016) | BSh |
| A-63 | 29038 | Kowanyama Airp. | Australia | 10 | -15.48 | 141.75 | (1912/1999-2016) | Aw |
| A-64 | 32078 | Ingham Composite | Australia | 12 | -18.65 | 146.18 | (1968/1990-2016) | Am |
| A-65 | 40854 | Logan City W.T.P. | Australia | 14 | -27.68 | 153.19 | (1990/1992-2016) | Cfa |
| A-66 | 8095 | Mullewa | Australia | 268 | -28.54 | 115.51 | (1896/1990-2016) | BSh |
| A-67 | 8251 | Kalbarri | Australia | 6 | -27.71 | 114.17 | (1970/1990-2016) | BSh |
| A-68 | 8225 | Eneabba | Australia | 100 | -29.82 | 115.27 | (1964/1990-2016) | Csa |
| A-69 | 7139 | Paynes find | Australia | 339 | -28.5 | 119.74 | (1919/1990-2016) | BWh |
| A-70 | 10007 | Bencubbin | Australia | 359 | -30.81 | 117.86 | (1912/1990-2016) | BSh |




| A-71 | 10092 | Merredin | Australia | 315 | -31.48 | 118.28 | (1903/1990-2016) | BSk |
| A-72 | 12038 | Kalgoorlie-Boulder Airp. | Australia | 365 | -30.78 | 121.45 | (1939/1994-2016) | BSh |
| A-73 | 16098 | Tarcoola Aero | Australia | 123 | -30.71 | 134.58 | (1990/1999-2016) | BWh |
| A-74 | 18195 | Minnipa Pirsa | Australia | 165 | -32.84 | 135.15 | (1990/2003-2016) | BSk |
| A-75 | 46126 | Tibooburra Airp. | Australia | 176 | -29.44 | 142.06 | (1990/2003-2016) | BWh |
| A-76 | 48245 | Boorke Airp. AWS | Australia | 107 | -30.04 | 145.95 | (1990/2002-2016) | BSh |
| A-77 | 55325 | Tamworth Airp. AWS | Australia | 395 | -31.07 | 150.84 | (1990/2006-2016) | Cfa |
| A-78 | 38026 | Birdsville Airp. | Australia | 47 | -25.9 | 139.35 | (1990/2001-2016) | BWh |
| A-79 | 30161 | Richmond Airp. | Australia | 206 | -20.7 | 143.12 | (1990/2003-2016) | BSh |
| A-80 | 33013 | Collinsville Airp. | Australia | 196 | -20.55 | 147.85 | (1939/1990-2016) | BSh |

*Köppen classification obtained from Peel et al. (2007).

# In the case of Australian stations, the periods of observations vary between different climatic parameters. E.g. for the example case (1939/1990-2016), the two dates separated with "/" show the starting date of the oldest and newest record of parameters used in calculations, respectively, while 2016 is the ending date of the records.

10 **Table 2.** The % coverage* of $MAD\%$ classes based on mean annual values, $R^2$ and $RMSD$ based on comparisons of the mean monthly values of $ET_o$ and $R_s$ methods.

| $MAD\%$ range | †$ET_o$ (ASCE-tall) for $C_n$=1600, $C_d$=0.38 (Eq.1) | †$ET_o$ (P-T) for $a_{pt}$=1.26 (Eq.2) | †$ET_o$ (H-S) for $c_{hs2}$ = 0.0023 (Eq.4b) | ‡$R_s$ (H-S) for $K_{RS}$=0.17 (Eq.3) |
|---|---|---|---|---|
| ≤ -50% | 0.0%* | 1.0% | 0.0% | 0.4% |
| -50 up to -25% | 0.0% | 18.1% | 5.8% | 1.0% |
| -25 up to -10% | 0.0% | 12.7% | 18.1% | 5.3% |
| -10 up to 10% | 21.3% | 23.1% | 26.3% | 59.6% |
| 10 up to 25% | 39.0% | 23.6% | 20.1% | 27.6% |
| 25 up to 50% | 39.4% | 17.3% | 26.2% | 6.2% |
| > 50% | 0.3% | 4.2% | 3.5% | 0.0% |
| $R^2$ | 0.96 | 0.75 | 0.89 | 0.87 |
| $RMSD$ | 39.6§ | 35.6§ | 25.0§ | 2.309# |

*The % coverage was estimated after conversion from WGS84 ellipsoid to projected Cylindrical Equal Area coordinate system without considering Antarctica.

† $MAD\%$ of the three $ET_o$ methods is estimated versus ASCE-short.

15 ‡ $MAD\%$ of the typical solar radiation method of H-S is estimated versus the $R_s$ data of Sheffield et al. (2006).

§ The unit of $RMSD$ for $ET_o$ is mm month$^{-1}$.

# The unit of $RMSD$ for $R_s$ is MJ m$^{-2}$ d$^{-1}$.





**Table 3.** Spatial extent of the major climatic groups CGs from Köppen-Geiger climate map (Peel et al., 2007), % prevalence of P-T versus H-S within each CG based on the *DMAD* values.

| Climatic group (CG) of Köppen-Geiger | % extent of CGs* based on Peel et al. (2007) map | P-T versus H-S prevalence % inside a CG# | | |
|---|---|---|---|---|
| | | H-S (DMAD<=-1) | Trans. Zone -1<DMAD<1† | P-T (DMAD=>1) |
| A - tropical/megathermal | 20.66% | 32.0% | 3.6% | 64.4% |
| B - arid/semi-arid | 32.90% | 86.4% | 1.3% | 12.3% |
| C - temperate/mesothermal | 14.58% | 32.8% | 3.2% | 64.1% |
| D - continental/microthermal | 27.00% | 26.9% | 2.1% | 71.0% |
| E - polar/alpine (without Antarctica) | 4.86% | 71.1% | 16.3%‡ | 12.5% |

*The % coverage was estimated after conversion from WGS84 ellipsoid to projected Cylindrical Equal Area coordinate system without considering Antarctica.

# % coverage of DMAD values were estimated after pixel resampling using the resolution of Köppen map.

†DMAD range were both methods present similar proximity to ASCE-short method (transitional zone).

‡Big part of this percentage corresponds to regions with annual $ET_o$ equal to 0 (e.g. inner Greenland). Such cases are included in the trans. zone of Fig.5a.



**Table 4.** Statistical criteria from the comparisons a) between $ET_o$ values from ASCE-short and P-T methods with typical and modified coefficients, b) between $ET_o$ values from ASCE-short and H-S methods with typical and modified coefficients, c) between $ET_o$ values from ASCE-tall and P-T or H-S methods with modified coefficients, d) $R_s$ values from Sheffield et al. (2006) and H-S method with typical and modified coefficients.

| Case | | 1 | | | 2 | | | |
|---|---|---|---|---|---|---|---|---|
| | | P-T vs. ASCE-short | | | H-S vs. ASCE-short | | | |
| Criterion | Optimum value | P-T (Eq.2) with $a_{pt}$=1.26 | P-T (Eq.2) with $a_{pt}$=p.w.a.s. (30 arc-sec) | P-T (Eq.2) with $a_{pt}$=p.w.a.s. (0.5 deg) | H-S (Eq.4b) with $c_{hs2}$=0.0023 | H-S$_{DRAL}$ (Eq.4c) | H-S (Eq.4b) with $c_{hs2}$=p.w.a.s. (30 arc-sec) | H-S (Eq.4b) with $c_{hs2}$=p.w.a.s. (0.5 deg) |
| MAE | 0 | 39.044 | 24.089* | 24.122 | 23.364 | 25.025 | 19.095 | 18.802* |
| RMSE | 0 | 52.275 | 31.082* | 31.238 | 34.074 | 33.304 | 25.250 | 25.159* |
| NRMSE% | 0 | 97.300 | 42.200* | 42.800 | 59.700 | 49.500 | 38.200 | 38.100* |
| PBIAS% | 0 | 35.400 | 8.000* | 9.100 | 12.600 | 4.800 | 3.400* | 4.000 |
| $R^2$ | 1 | 0.745 | 0.845 | 0.847* | 0.837 | 0.795 | 0.881 | 0.883* |
| $bR^2$ | 1 | 0.568 | 0.816* | 0.810 | 0.741 | 0.773 | 0.854 | 0.852* |
| NSE | 1 | 0.053 | 0.822* | 0.817 | 0.643 | 0.755 | 0.854 | 0.855* |
| d | 1 | 0.834 | 0.953* | 0.952 | 0.929 | 0.940 | 0.966* | 0.965 |
| KGE | 1 | 0.493 | 0.884* | 0.878 | 0.700 | 0.864 | 0.889* | 0.887 |

| Case | | 3 | | | | 4 | | |
|---|---|---|---|---|---|---|---|---|
| | | P-T & H-S vs. ASCE-tall | | | | H-S $R_s$ (Eq.3) vs. $R_s$ from Sheffield et al. (2006) | | |
| Criterion | Optimum value | P-T (Eq.2) with $a_{pt}$=p.w.a.t. (30 arc-sec) | P-T (Eq.2) with $a_{pt}$=p.w.a.t. (0.5 deg) | H-S (Eq.4b) with $c_{hs2}$=p.w.a.t. (30 arc-sec) | H-S (Eq.4b) with $c_{hs2}$=p.w.a.t. (0.5 deg) | H-S $R_s$ (Eq.3) with $K_{RS}$=0.17 | H-S $R_s$ (Eq.3) with $K_{RS}$=p.w.a. (30 arc-sec) | H-S $R_s$ (Eq.3) with $K_{RS}$=p.w.a. (0.5 deg) |
| MAE | 0 | 43.052 | 42.977* | 35.359 | 34.746*‡ | 1.641 | 0.987* | 1.047 |
| RMSE | 0 | 55.546* | 55.578 | 46.893 | 46.543*‡ | 1.985 | 1.318* | 1.426 |
| NRMSE% | 0 | 53.100* | 53.700 | 49.800 | 49.600*‡ | 29.600 | 20.700* | 22.300 |
| PBIAS% | 0 | 9.100* | 10.600 | 4.800*‡ | 5.800 | -4.500 | -0.799* | -0.800 |
| $R^2$ | 1 | 0.757 | 0.761* | 0.803 | 0.809*‡ | 0.930 | 0.959* | 0.952 |
| $bR^2$ | 1 | 0.733* | 0.727 | 0.775*‡ | 0.774 | 0.885 | 0.952* | 0.944 |
| NSE | 1 | 0.718* | 0.712 | 0.752 | 0.754*‡ | 0.912 | 0.957* | 0.950 |
| d | 1 | 0.926* | 0.925 | 0.941 | 0.942*‡ | 0.977 | 0.989* | 0.988 |
| KGE | 1 | 0.841* | 0.834 | 0.847*‡ | 0.843 | 0.932 | 0.973* | 0.972 |

*The asterisk is used for each case (1,2,3,4) to indicate the best value of each criterion. In case 3, the comparison is between pixel resolutions separately for each method (P-T or H-S).

‡ Applicable only in Case 3. The symbol denotes the best value of each criterion between both pixel resolutions and both methods (P-T and H-S).

**FIGURES**

**Figure 1.** Position of stations from **(a)** California-USA obtained by CIMIS database and **(b)** Australia obtained by AGBM database (the numbers indicate the No. of stations from Table 1 without the abbreviations CA- and A-).







**Figure 2.** Mean annual values (mm year$^{-1}$) of $ET_o$ for the period 1950-2000 using **(a)** the ASCE-short method, **(b)** the ASCE-tall method, **(c)** the typical P-T method for $a_{pt}$=1.26 and **(d)** the typical H-S method for $c_{hs2}$=0.0023 (30 arc-sec resolution maps).



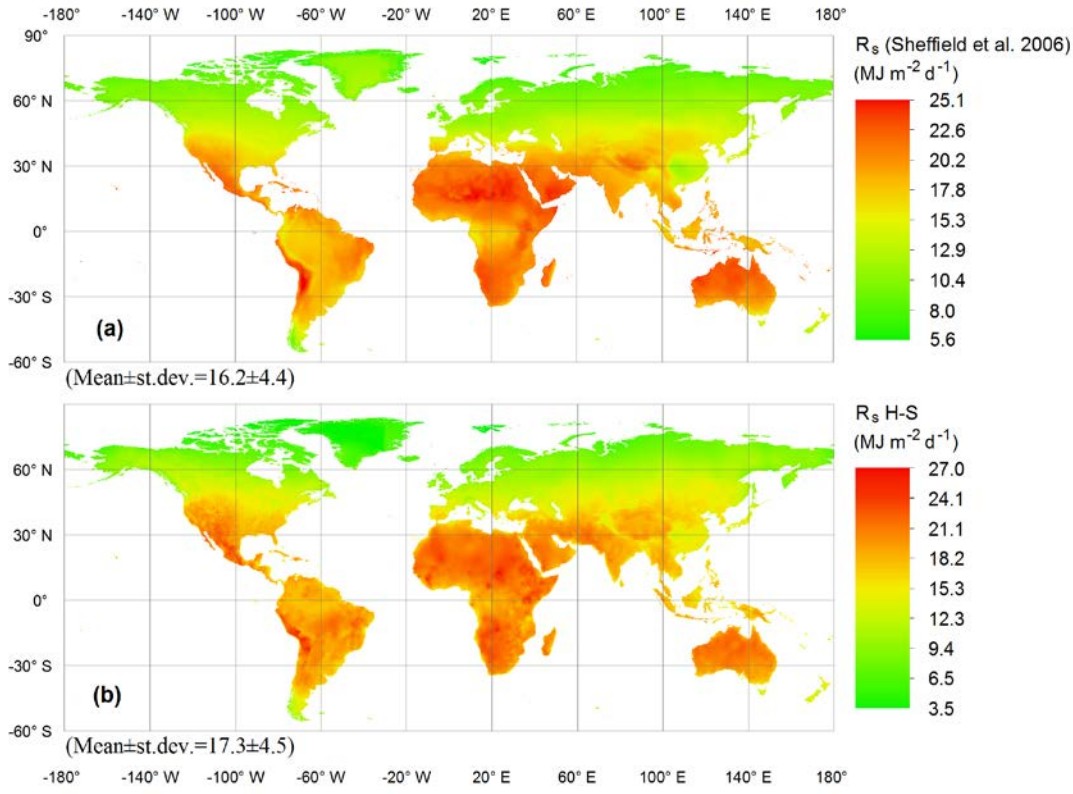

**Figure 3.** Mean annual values of $R_s$ (MJ m$^{-2}$ d$^{-1}$) for the period 1950-2000 **(a)** from the database of Sheffield et al. (2006) and **(b)** estimated using the typical H-S method for $K_{RS}$=0.17 (Eq.3) (30 arc-sec resolution maps).





**Figure 4.** Mean annual difference % (*MAD*%) of $ET_o$ between the ASCE-short and **(a)** the ASCE-tall method, **(b)** the typical P-T method for $a_{pt}$=1.26, **(c)** the typical H-S method for $c_{hs2}$ = 0.0023 and **(d)** *MAD*% between $R_s$ values of Sheffield et al. (2006) and the typical solar radiation formula of H-S for $K_{RS}$=0.17 (30 arc-sec resolution maps).





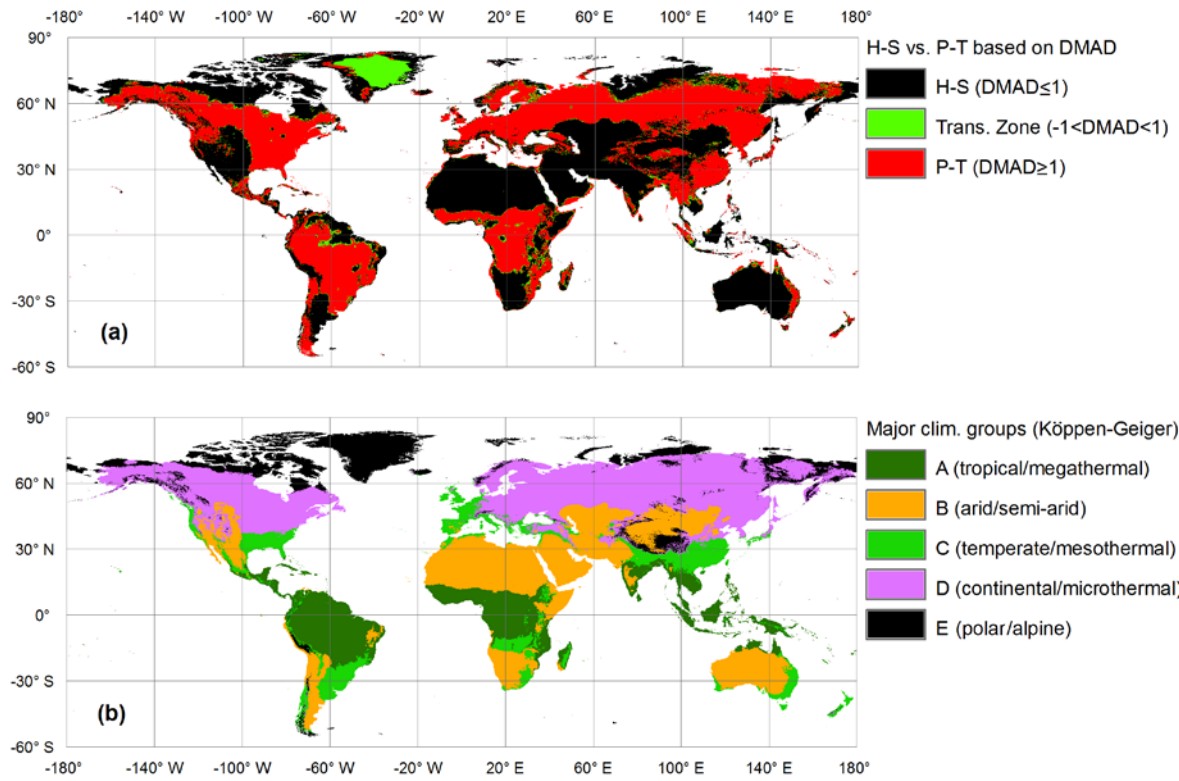

**Figure 5**. **(a)** P-T versus H-S prevalence according to their proximity to ASCE-short method expressed by the *DMAD* values (30 arc-sec resolution map) and **(b)** Spatial extent of the major climatic groups of the Köppen-Geiger climate classification according to Peel et al. (2007).



**Figure 6.** Partial weighted averages of mean monthly **(a)** $a_{pt}$ for short reference crop, **(b)** $a_{pt}$ for tall reference crop, **(c)** $c_{hs2}$ for short reference crop and **(d)** $c_{hs2}$ for tall reference crop (values in parenthesis below each map where estimated excluding pixels of 0 value) (30 arc-sec resolution maps).




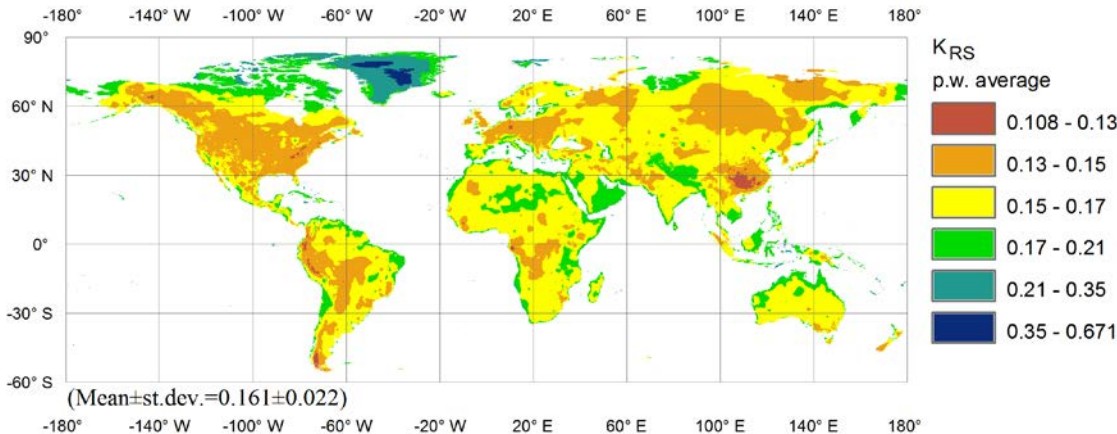

**Figure 7.** Partial weighted averages of mean monthly $K_{RS}$ (30 arc-sec resolution map).





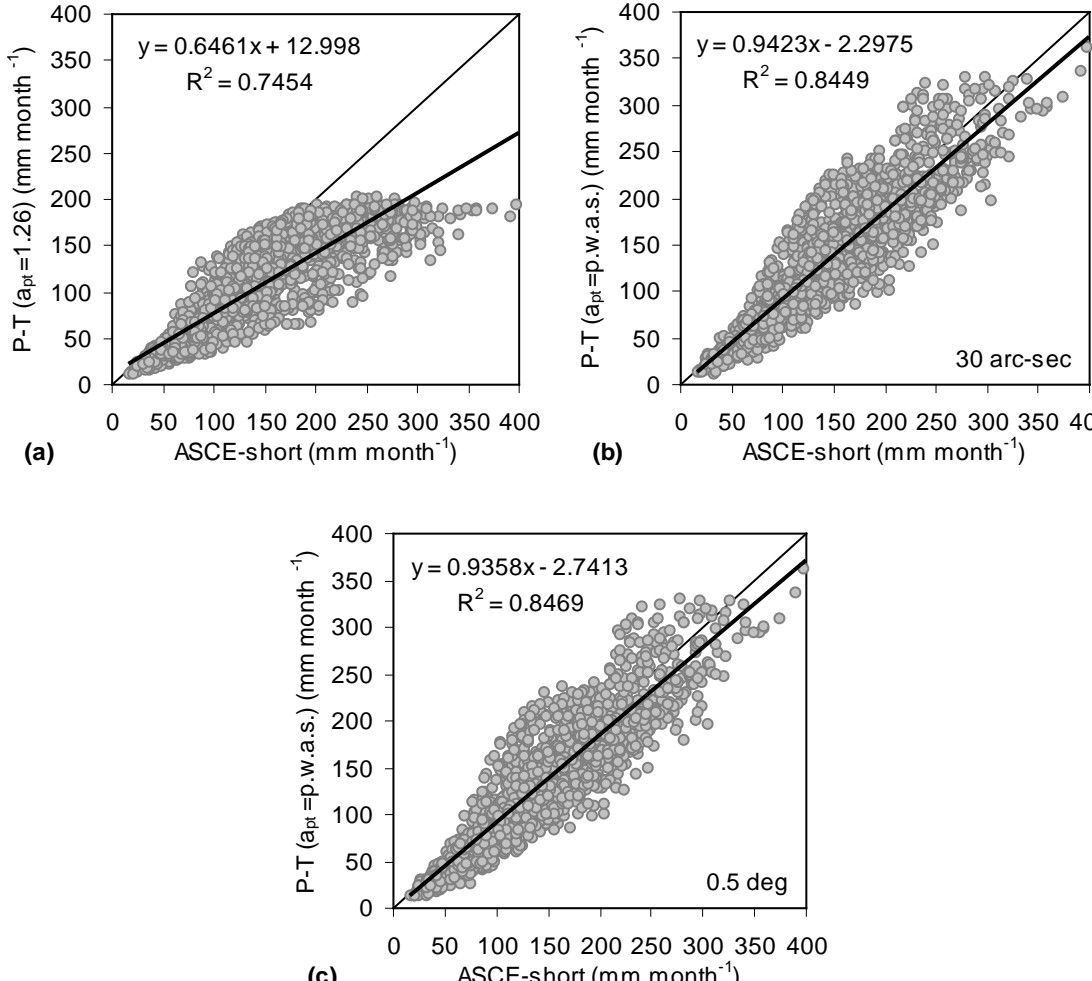

**Figure 8.** Comparative 1:1 plots between the results of ASCE-short versus **(a)** the typical P-T method with $a_{pt}$=1.26, **(b)** the P-T method with $a_{pt}$=p.w.a.s. from 30 arc-sec resolution map and **(c)** the P-T method with $a_{pt}$=p.w.a.s. from 0.5 deg. resolution map.





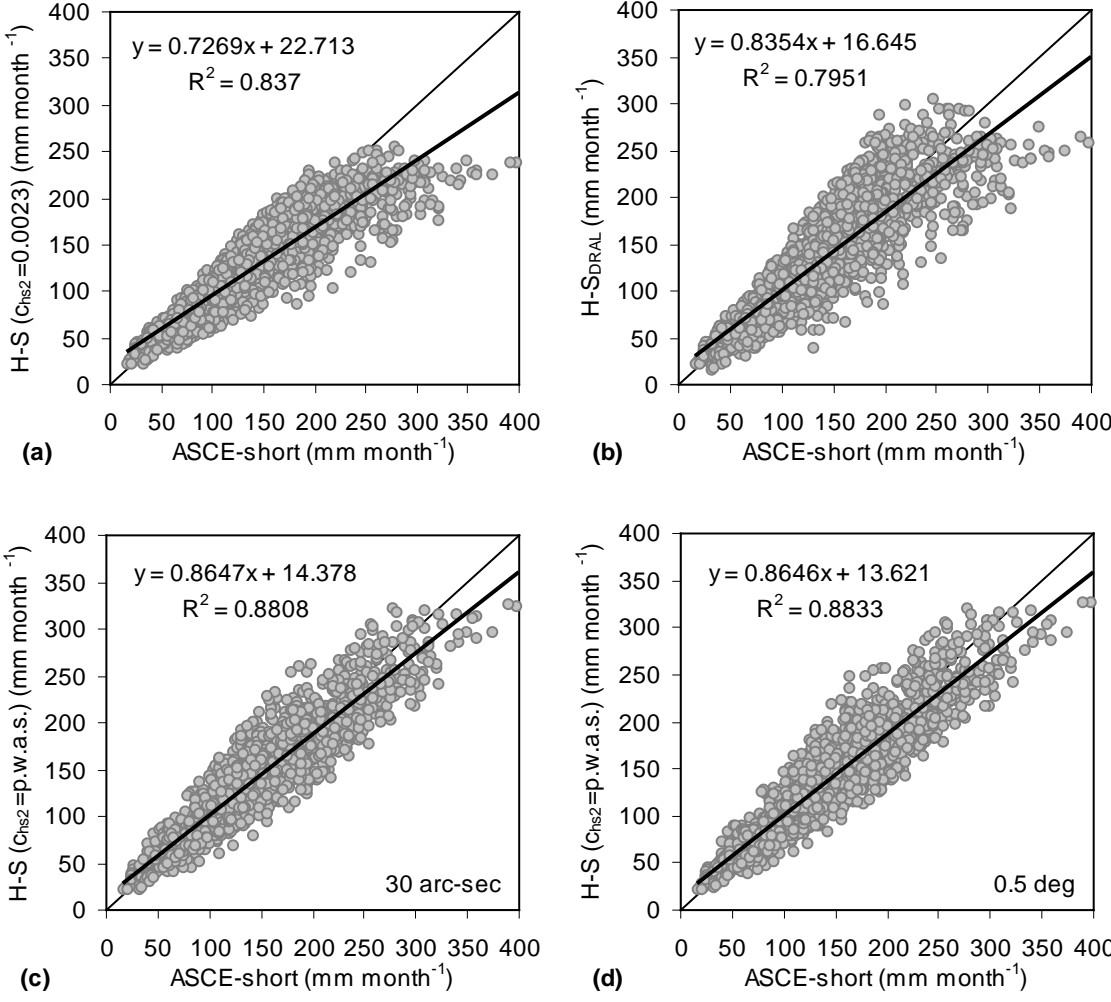

**Figure 9.** Comparative 1:1 plots between the results of ASCE-short versus **(a)** the typical H-S method with $c_{hs2}$=0.0023, **(b)** the H-S$_{DRAL}$ method, **(c)** the H-S method with $c_{hs2}$=p.w.a.s. from 30 arc-sec resolution map and **(d)** the H-S method with $c_{hs2}$=p.w.a.s. from 0.5 deg resolution map.




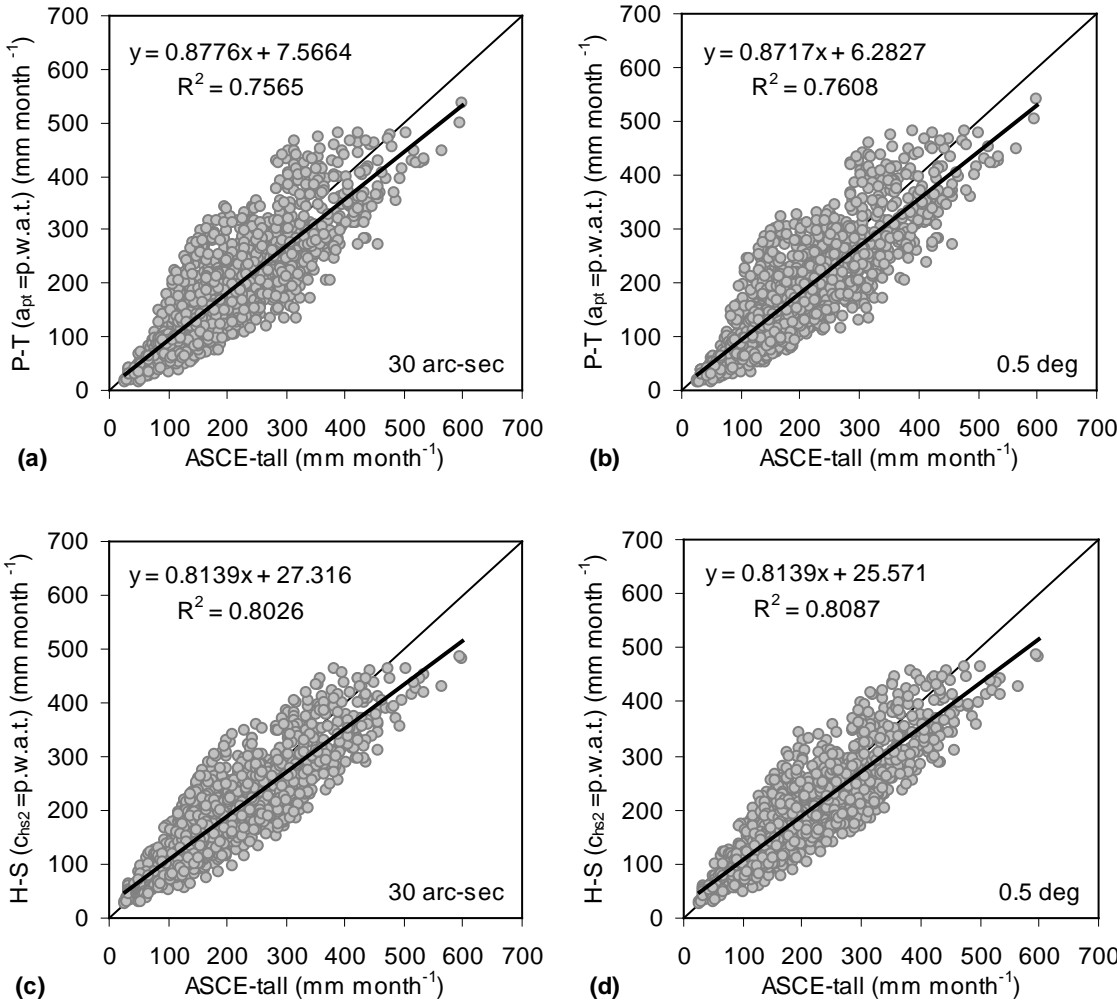

**Figure 10.** Comparative 1:1 plots between the results of ASCE-tall versus **(a)** the P-T method with $a_{pt}$=p.w.a.t. from 30 arc-sec resolution map, **(b)** the P-T method with $a_{pt}$=p.w.a.t. from 0.5 deg resolution map, **(c)** the H-S method with $c_{hs2}$=p.w.a.t. from 30 arc-sec resolution map and **(d)** the H-S method with $c_{hs2}$=p.w.a.t. from 0.5 deg resolution map.





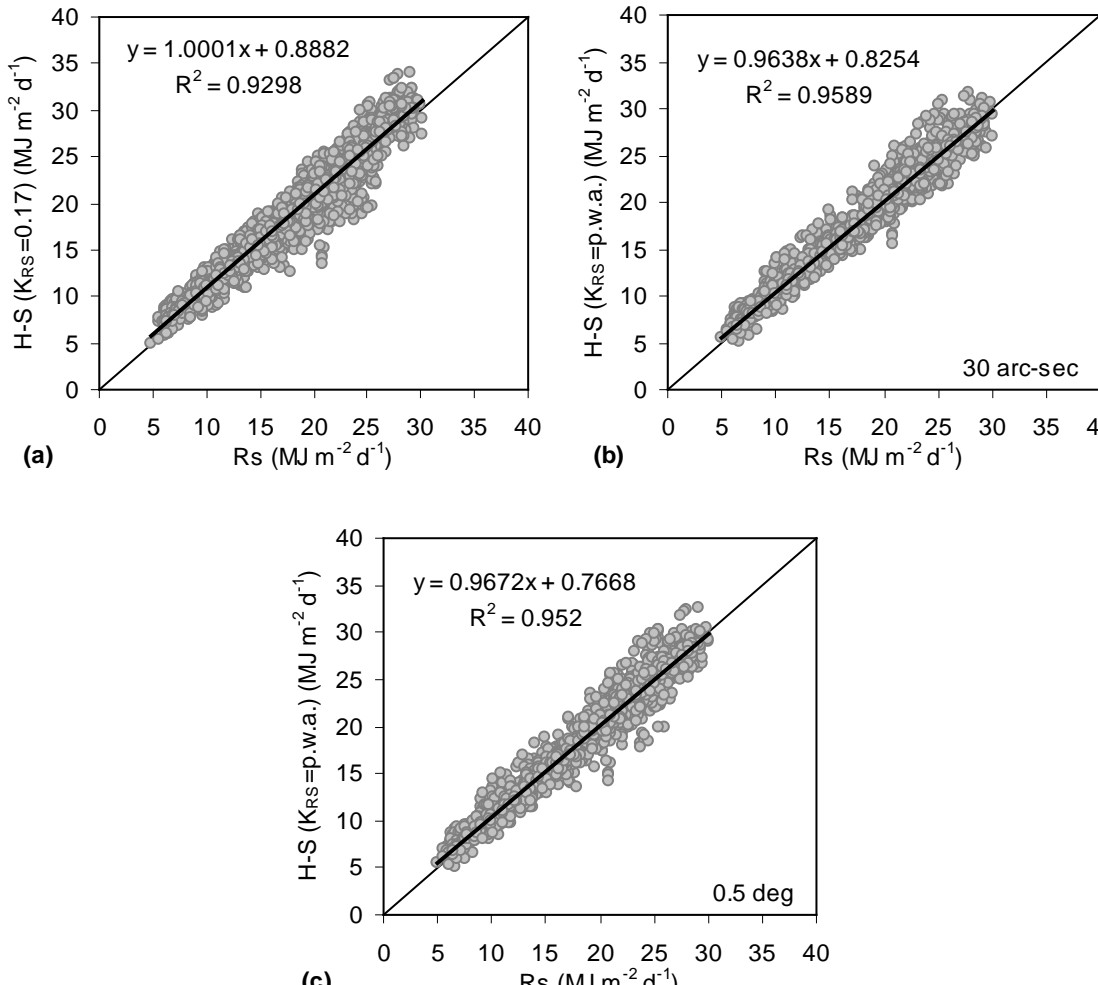

**Figure 11.** Comparative 1:1 plots between the results of $R_s$ (MJ m$^{-2}$ d$^{-1}$) of Sheffield et al. (2006) versus the H-S formula for radiation $R_s$ **(a)** with $K_{RS}$= 0.17,  **(b)** with $K_{RS}$= p.w.a. from 30 arc-sec resolution map and **(c)** with $K_{RS}$= p.w.a. from 0.5 deg resolution map.





**Figure 12.** Taylor diagrams **(a)** for $ET_o$ estimations of short reference crop, **(b)** for $ET_o$ estimations of tall reference crop and **(c)** for $R_s$ estimations.



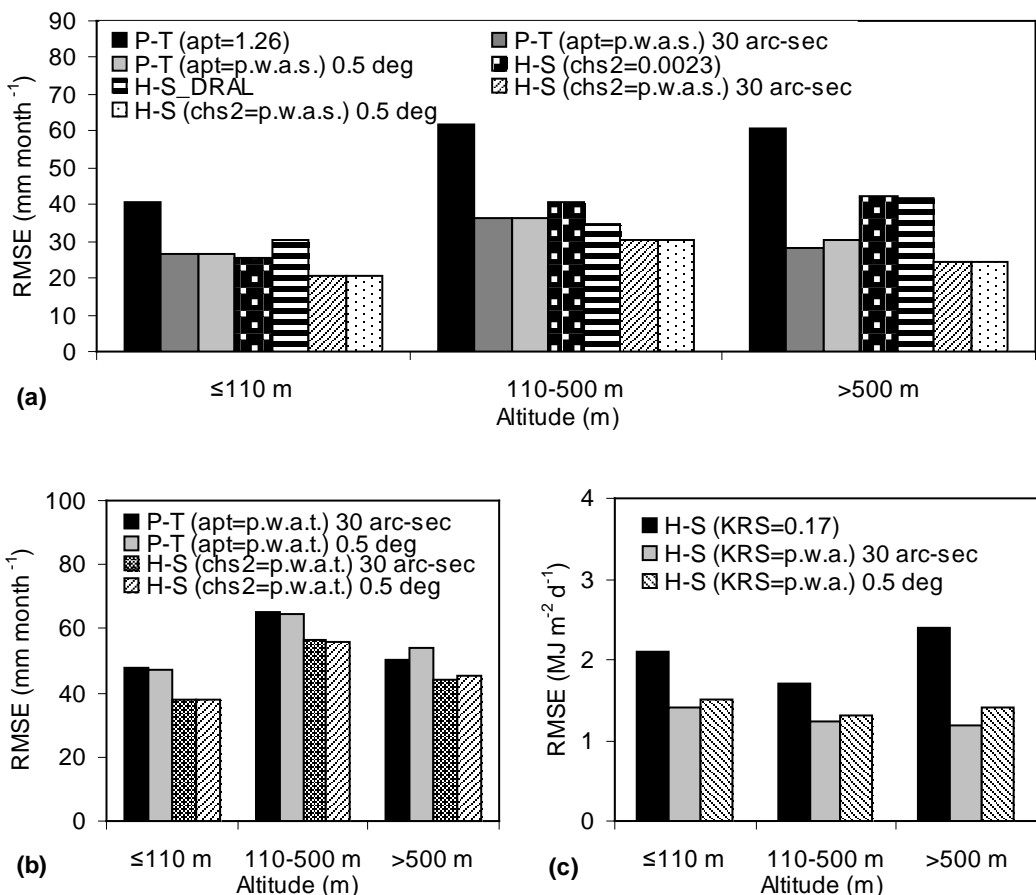

**Figure 13.** Comparison of RMSEs between different methods describing **(a)** $ET_o$ for short reference crop, **(b)** $ET_o$ for tall reference crop and **(c)** and $R_s$, after splitting the validation data of stations to three groups belonging to <110, 110-500 and <500 m altitude ranges.



(a)

(b)

(c)

(d)





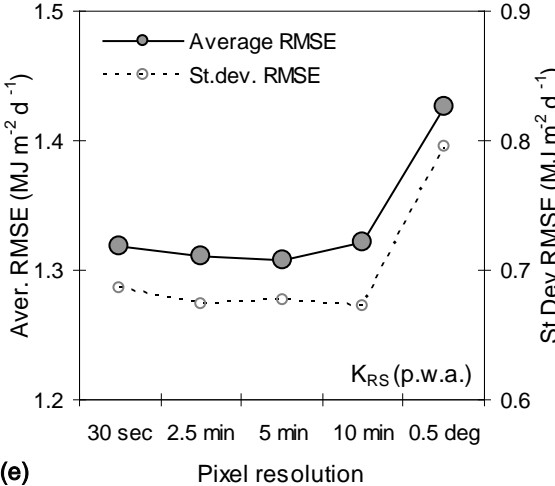

**(e)**

**Figure 14.** Comparison of the average and standard deviation of *RMSEs* of the validation dataset between different pixel resolutions (30 arc-sec, 2.5 arc-min, 5 arc-min, 10 arc-min and 0.5 deg) for the **(a)** $a_{pt}$ (p.w.a.s.), **(b)** $c_{hs2}$ (p.w.a.s.), **(c)** $a_{pt}$ (p.w.a.t.), **(d)** $c_{hs2}$ (p.w.a.t.), and **(e)** $K_{RS}$ (p.w.a.).