# Peer review of "High resolution global grids of revised Priestley-Taylor and Hargreaves-Samani coefficients for assessing ASCE-standardized reference crop evapotranspiration and solar radiation"

_Earth System Science Data, 2016_

## Referee Comment (RC1) · Anonymous Referee #1 · 15 Feb 2017

I appreciate the Editor to give me a chance to review an interesting and valuable paper. I found some merits in the both methodology and results. In my opinion, this paper has a good potential to be published in the journal. However, I have also some concerns on the different parts of the manuscript. If only the author(s) address carefully to all of my comments, I'll recommend publication of the manuscript in the journal: • What was the criterion to select the stations? Why the USA and Australia? Are they covering all climates? • Lns 7-8, cite also these three useful papers to enhance the literature:

[Figure]

Selecting the best model to estimate potential evapotranspiration with respect to climate change and magnitudes of extreme events Temporal analysis of reference evapotranspiration to detect variation factors Analysis of potential evapotranspiration using limited weather data • Lns 15-16, cite also these two useful papers to enhance the literature: Application of new mass transfer formulae for computation of evapotranspiration Ability of Box-Jenkins Models to Estimate of Reference Potential Evapotranspiration (A Case Study: Mehrabad Synoptic Station, Tehran, Iran) • In the last paragraph of the Introduction, the authors should clearly mention the weakness point of former works (identification of the gaps) and describe the novelties of the current investigation to justify us the paper deserves to be published in this journal. • Compare the results with modified/calibrated H-S and P-T models presented by other researchers in all of the world (particularly in the USA and Australia). • The Discussion section should be broken to sub-sections for better understanding readers. • Explain the variations of the spatial extent of the major climatic groups CGs from Köppen-Geiger climate map. • Discuss the comparison of the average and standard deviation of RMSEs of the validation dataset between different pixel Resolutions more thoroughly. • What are the strategies/recommendations to reduce uncertainties in this study? • At the end of the manuscript, explain the implications and future works considering the outputs of current study. • The quality of the language needs to improve by a native English speaker for grammatically style and word use.

Please also note the supplement to this comment:
http://www.earth-syst-sci-data-discuss.net/essd-2016-59/essd-2016-59-RC1-supplement.pdf
* * *

---

## Referee Comment (RC2) · Anonymous Referee #2 · 17 Apr 2017

Revision of High resolution global grids of revised Priestley-Taylor and Hargreaves-Samani coefficients for assessing ASCE-standardized reference crop evapotranspiration and solar radiation.

This study analyses the global relationship between ETo annual maps obtained by means of three different methods (ASCE PM, P-T and H-S), with the purpose of determining the accuracy of the methods based on poor data availability (P-T and H-S). In addition, the manuscript includes a calibration exercise to obtain revised coefficients

at the global scale for P-T and H-S equations based on the obtained ASCE PM data. The research topic is highly relevant given the relevance of estimating the atmospheric evaporative demand (AED) with accuracy since AED is an important hydroclimatic variable with strong implications in aridity conditions and climate change processes. The manuscript is in general well written, the figures show high quality and it has a good structure. The authors use a high amount of data for analysis and validation, including gridded datasets and meteorological networks in California and Australia.

The manuscript is a bit long and sometimes it is different to follow but independently of formal issues I find major methodological problems in the manuscript, which are related to the treatment of the data used, the spatial resolution of the gridded products and the assessment of the uncertainty in the ETo estimations. I am including some detailed issues below about these issues. I would recommend the authors to work at coarser spatial resolution to reduce the strong uncertainty associated to the selected high resolution (1 km) of final products.

Page 4: I find highly problematic to interpolate the low resolution 0.5° data for wind speed, humidity and solar radiation to 1 km. The results of the bilinear interpolation of the 0.5° data does not really increase the necessary spatial resolution of these variables to be compared with the high resolution of tmax and tmin data (in any case high resolution temperature data from the global dataset used is also affected by spatial errors and uncertainties, which should be also taken into account). The 1 km interpolated wind speed, humidity and solar radiation has a spatial resolution completely unreal. These variables are essential to be taken into account to estimate ETo spatial patterns since ETo is usually more sensitive to these variables than to temperature (McVicar et al., 2012a and b). For this reason, I consider that 1 km gridded maps generated in this study show high uncertainty, which is not quantified/provided in this study.

The authors are computing Eto by PM equation as reference to be compared with H-S and P-T methods, but there is not any assessment of the error in the PM estimations
related to the data inaccuracies and the poor resolution of the input climate data. I think these problems would be solved (not completely since an assessment of uncertainty should be taken into account) if authors consider to focus at coarse (0.5°) spatial resolution, which avoids unnecessary interpolation of wind speed, radiation and humidity variables and the outputs would be useful for continental to global assessments. Thus, the results of figures 8-10 confirms that interpolation of low resolution variables have strong influence on the comparability of different ETo estimations, which can be associated to the poor interpolation approach applied to the coarse climate variables.

I have also doubts on the use of the coefficients calculated in this study to calculate ETo using H-S and P-T equations. The authors obtain the calibration coefficients for the period 1950-2000 and assume stationary climate conditions. Nevertheless, under climate scenarios in which input climate variables change (I refer to wind speed, relative humidity and incoming solar radiation) under a non-stationary scenario, the obtained coefficients would not be useful to calculate ETo based on scarce climate data. Different studies have showed recent changes in solar radiation (Wild et al., 2013), wind speed (McVicar et al., 2012b) and atmospheric humidity (Willet et al. 2014).

Given that the main objective of this study is the re-calibration of the H-S and P-T equations, it would be necessary that authors provide not only the recalibrated coefficients but also a measure of the accuracy considering the errors in the interpolated variables used in P-M calculations.

Page 8. Really I do not find useful the annual coefficients in areas that show strong climate seasonality (as in the majority of world regions). In addition, there are not seasonal accuracy statistics, which can be much more relevant than annual ones.

McVicar, T.R., Roderick, M.L., Donohue, R.J., Van Niel, T.G., (2012a): Less bluster ahead? ecohydrological implications of global trends of terrestrial near-surface wind speeds. Ecohydrology 5 (4), 381–388. McVicar, T.R., Roderick, M.L., Donohue, R.J., et al., (2012b). Global review and synthesis of trends in observed terrestrial near-surface

wind speeds: implications for evaporation. J. Hydrol. 416–417, 182–205. Wild, M., D. Folini, C. Schaer, N. Loeb, E. G. Dutton, and G. Koning-Langlo (2013), The global energy balance from a surface perspective, Clim. Dyn., 40, 3107–3134. Willett, K.M., Dunn, R.J.H., Thorne, P.W. et al. (2014): HadISDH land surface multi-variable humidity and temperature record for climate monitoring. Climate of the Past 10: 1983-2006.

---

## Author Comment (AC1) · 21 Jun 2017

Comment R1.1: I appreciate the Editor to give me a chance to review an interesting and valuable paper. I found some merits in the both methodology and results. In my opinion, this paper has a good potential to be published in the journal. However, I have also some concerns on the different parts of the manuscript. If only the author(s) address carefully to all of my comments, I'll recommend publication of the manuscript

in the journal.

Response: We would like to thank the reviewer for the constructive comments. We followed his suggestions for improving the manuscript. Our responses to specific comments are given below. We also suggest the reviewer to check carefully the responses to the comments of Reviewer 2, since his suggestions led to substantial changes in the manuscript.

Comment R1.2: What was the criterion to select the stations? Why the USA and Australia? Are they covering all climates?

Response: The reasons for choosing the CIMIS-database of California (USA) and AGBM database of Australia are: - The first database includes stations from California-USA and it was selected because: a) it has been used as a basis for the development of Hargreaves-Samani method (Hargreaves and Samani, 1985; Hargreaves and Allen, 2002) and CIMIS method (Snyder and Pruitt, 1985, Snyder and Pruitt, 1992) and b) provides a dense and descriptive network of stations for a specific region that combines coastal, plain, mountain and desert environments (Table 1, Fig.1a in the manuscript). The second database includes stations from Australia and it was selected because the stations network covers a large territory with large variety of climate classes (Table 1, Fig.1b in the manuscript), but also because the Priestley-Taylor method has been calibrated for locations of eastern Australia (Priestley and Taylor, 1972). For the stations of AGBM database, the selection of stations was performed in such way in order to cover all the possible existing Köppen climatic types and elevation ranges of Australian continent (Table 1 in the manuscript). (see text in Page 9, lines 25-35, Page 10, lines 0-5)

- Additional reason for choosing these two databases was that they provide a large number of stations with complete data for estimating ASCE-ETo covering large observation periods before and after the year 2000 (Table 1 in the manuscript). This was a prerequisite in this study because the rasters of the new coefficients were developed

based on mean monthly climatic parameters of 1950-2000. Thus, using stations with many available data after 2000, we could prove that the derived coefficients also work for current conditions. (see text in Page 10, lines 8-10)

- We have to mention that the combination of the databases provided a wide range of the mean monthly values of the parameters used for the ETo estimations (this is an additional reason for their selection). The general statistics of the aforementioned parameters are also given in Table R1 below, which was included in the supplementary material as Table S1 (reference for Table S1 exist in the text in Page 10, line 24). In order to show the high variability in the parameters of the validation dataset, we provide frequency diagrams of mean monthly Tmax, Tmin, Rs, RH%, u2 and P are given below in Fig.R1a,b,c,d,e,f, respectively.

- Taking into account the last column of Table 1 (Köppen-Geiger classification), we provided the climatic classification of each station. According to Table 1, from the 140 stations, 9 belong to A Köppen-Geiger group (tropical/megathermal), 69 to B group (Arid/semi-arid), 59 to C group (temperate/mesothermal) and 3 to D group (continental/microthermal). We believe, that apart from the D group, the number of stations for the rest climatic groups are enough for validating the results. As concern the D group, we couldn't find more stations with adequate data inside the aforementioned databases. Other databases, which may provide data for stations from other parts of the world, were not used in order to fully exploit the two aforementioned databases but also to give the opportunity to other scientists to test our revised coefficients for their territories using other complete databases and not selected stations from various databases. Many existing databases of observed data may show differences in the methods used for measuring and presenting data. Such differences were also observed in the CIMIS and AGBM databases and they were used to justify many uncertainties observed during the implementation of this work (see response to comment R2.3 of reviewer 2 and the new section in the discussion with title "Uncertainties in the data used for calibrating and validating the revised coefficients of P-T and H-S methods". It would be difficult to identify such uncertainties using stations from multiple databases.

Table R1. General statistics* of the mean monthly observed values of climatic parameters from the 140 stations of California-USA and Australia that participate in the estimation of reference evapotranspiration with the ASCE method.

[TABLE R1, PLACE HERE]

*The statistics are based on 1680 values (140 stations $\times$ 12 months)

Fig.R1 Frequency diagrams (number of monthly values) based on the mean monthly data of the 140 stations of CA-USA and Australia for a) maximum temperature Tmax, b) minimum temperature Tmin, c) solar radiation Rs, d) relative humidity RH, e) wind speed u2 at 2 m height and f) precipitation P. The frequencies are based on a total number of observations equal to 1680 (140 stations $\times$ 12 months).

[FIGURE R1, PLACE HERE]

Comment R1.3: Lns 7-8, cite also these three useful papers to enhance the literature: - Selecting the best model to estimate potential evapotranspiration with respect to climate change and magnitudes of extreme events. - Temporal analysis of reference evapotranspiration to detect variation factors. - Analysis of potential evapotranspiration using limited weather data.

Response: The proposed citations were added at the proposed locations in the text and in the references list.

Comment R1.4: Lns 15-16, cite also these two useful papers to enhance the literature: - Application of new mass transfer formulae for computation of evapotranspiration - Ability of Box-Jenkins Models to Estimate of Reference Potential Evapotranspiration (A Case Study: Mehrabad Synoptic Station, Tehran, Iran)

Response: The proposed citations were added at the proposed locations in the text

and in the references list.

Comment R1.5: In the last paragraph of the Introduction, the authors should clearly mention the weakness point of former works (identification of the gaps) and describe the novelties of the current investigation to justify us the paper deserves to be published in this journal.

Response: The most significant novelty of the study is that provides, for the first time, global maps of revised coefficients for the P-T and H-S evapotranspiration methods and revised coefficients for the H-S radiation formula. Such attempt has never been made in the past at the global scale, despite the fact that many studies for recalibrating the respective coefficients have been presented for many parts of the world. The final maps allow the comparison of the revised coefficients among regions under a common base since they were built using common datasets and using the same technique (Eqs.7), while they provide a global overview of the variation in these coefficients. Other novelties of the study are: • the development of global maps for the revised coefficients for P-T and H-S evapotranspiration methods for tall reference crop. • the development of global maps for the possible mean annual error, when the P-T and H-S are applied using the standard coefficients of the original methods (maps of MAD% parameter, Fig.4b,c,d in the manuscript). These maps provide information about the uncertainty when the standard H-S and P-T methods are used. These maps were also combined to derive a new map (map of DMAD parameter, Fig.5a in the manuscript), which indentifies the optimum locations for the application of the standard H-S and P-T formulas based on their proximity to the results of ASCE for short reference crop. The DMAD map is an important tool, which can give a solution when someone has to choose between the two methods. • the proposal of a method for deriving annual coefficients for the P-T and H-S methods. The procedure described by the set of Eqs.7, which estimates the partial weighted averages of the coefficients based on their monthly values, is a newly proposed method that can be easily applied in GIS environment, while it provides a solution when annual coefficients have to be derived under

a common base for many stations or for global applications using raster data. This technique is proposed as an alternative of optimization methods, which are difficult to be incorporated in GIS environment (see more details about this comment in Page 18, lines 13-24 and the respective subsection of the discussion). The method calibrates the basic coefficients without modifying or adding parameters in the original P-T and H-S equations. Finally, we have to stress that the aim of the study is to provide tools for facilitating the estimation of ETo and solar radiation for regions (especially those of developing countries), which face serious shortage of climatic data and not to propose the revised coefficients as an alternative method for regions, which have complete meteorological stations that provide detailed sets of all climatic variables. Of course, the use of such stations to validate the coefficients is the only solution. All the aforementioned aspects related to the novelties and the aims of the study are included in brief in the revised paragraph at the end of the introduction.

Comment R1.6: Compare the results with modified/calibrated H-S and P-T models presented by other researchers in all of the world (particularly in the USA and Australia).

Response: The requested task is quite difficult and there are many problems of comparability since either the majority of authors have modified the initial form of H-S and P-T models and not only the main coefficients, but also because the majority of the works provide H-S and P-T models, which are calibrated for regions outside California and Australia. For example, the popular recalibrated Priestley-Taylor model of Abtew (1996) for Florida-USA uses a recalibrated coefficient equal to 1.18, which is almost equal to our revised coefficient (1.17 from 0.5 degree resolution), but the model gives bad results in the validation procedure because Florida has a completely different environment from California and cannot cover the climatic variation of Australia stations. Other examples are the modified Makking model and other models given by Castañeda and Rao (2005), which were calibrated only for one station of southern California using 4 years of observations and the modified Hargreaves-Samani models and other models of Azhar and Perera (2011), which were recalibrated for three stations of southeastern

Australia with very few years of observations. Similar problems were observed for too many other cases that we examined using a large list of models provided by Valipour (2015a,b; 2017) and Valipour et al. (2017) and by models obtained from the works cited in the introduction. Thus, it is unfair to examine the accuracy of such models using the complete validation dataset (both California and Australian stations), while it is not feasible in the context of this article to examine one by one all the modified models published in the international literature. Of course, the analysis of modified H-S and P-T models calibrated for other parts of the world was rejected from the beginning because the validation dataset does not include stations from these regions. The only models, which are absolutely comparable are the modified H-S models by Droogers and Allen (2002) because they have been calibrated using global datasets. In order to partly satisfy the request of the reviewer, we selected some models of reduced parameters, which have similar or additional requirements from the standard H-S and P-T models. The final selected models were also those who showed a) the best performance after examining an extremely large list of models using both California-USA and Australia stations data and b) a good performance to other studies using other datasets. The comparison with such models will also contribute to verify the value of our coefficients as alternative options for ETo estimations with fewer variables. Based on the afore-mentioned observations, the following 8 models were selected for comparisons with the standard and re-adjusted H-S and P-T models: 1. Two modified models of H-S by Droogers and Allen (2002) where the second one uses precipitation as additional parameter. The models were based on calibrations using global data. 2. Three models of reduced parameters given by Valiantzas (2013a,b; 2014) that were calibrated using 535 stations from Europe, Asia, Africa. The first model uses temperature and radiation data, while the other two use temperature, radiation, and humidity data. The models have been tested for California (Valiantzas, 2013c) and Australia conditions (Ahoogha-landari et al., 2017). 3. Two models of reduced parameters by Ahooghalandari et al. (2016) calibrated using 18 stations from various locations of Australia. The models use temperature and relative humidity data. Ahooghalandari et al. (2017) also made recalibration of Valiantzas equations and other models but for a restricted region of Western Australia considering 8 stations, and for this reason these modified versions were not used. 4. The Copais model of Alexandris et al. (2006) that uses temperature, radiation and humidity data. The model was calibrated/validated using data from Greece, California and Oregon-USA, while it has shown a very good response to many other regions of the world including Australia (Ahooghalandari et al., 2017). The aforementioned observations and the description of the additional models used for comparisons were added as a new paragraph in the page 10 (lines 32-35) and 11 (lines 0-25) together with the new Table 2, which gives the equations of the additional models. The results of the models were added in Page 14 (lines 25-35), and 15 (lines 0-15). The comparisons of the models were included in new Fig.8 and their statistics in the new Table 5.

Comment R1.7: The Discussion section should be broken to sub-sections for better understanding of readers.

Response: we followed the suggestion of the reviewer and the discussion section was broken to sub-sections.

Comment R1.8: Explain the variations of the spatial extent of the major climatic groups CGs from Köppen-Geiger climate map.

Response: The variations of the spatial extent of the major climatic groups CGs from Köppen-Geiger climate map, which are given in the first column of Table 4 in the revised manuscript, are results obtained from the respective raster Köppen-Geiger climate map of Peel et al. (2007) and they are described in detailed in their article. We believe that is beyond the scope of the paper to discuss the observed % of CGs in the map of Peel et al. (2007) (keep in mind that % values are re-adjusted after excluding Antarctica and they are slightly different from those given by Peel). If the reviewer means to explain the results of the other columns of new Table 4 (i.e. why for example the typical H-S method is better from the typical P-T method in more arid environments or why P-T is better in

more humid environments) the explanation is based on the fact that the standard H-S method was calibrated for the conditions of California, which include arid and semi-arid environments, while the standard coefficient 1.26 of Priestley-Taylor method was obtained based on experiments of more humid environments. This justification was added in the text (see Page 13, lines 8-11).

Comment R1.9: Discuss the comparison of the average and standard deviation of RMSEs of the validation dataset between different pixel Resolutions more thoroughly.

Response: The comparison between different pixel resolutions was removed based on the objections of Reviewer 2. Reviewer 2 suggested not to use the resolutions below 0.5 degree due to interpolation limitations in the initial raster data for solar radiation, humidity and wind speed, which were at 0.5 degree. See comments of Reviewer 2. All the manuscript was revised from the beginning using only the results of 0.5 degree resolution.

Comment R1.10: What are the strategies/recommendations to reduce uncertainties in this study?

Response: a new subsection was added in the Discussion. See the new subsection with title "Recommendations for reducing the uncertainties when the re-adjusted coefficients of P-T and H-S models are used".

Comment R1.11: At the end of the manuscript, explain the implications and future works considering the outputs of current study.

Response: we added new text about it (see the last paragraph in the Conclusions section)

Comment R1.12: The quality of the language needs to improve by a native English speaker for grammatically style and word use.

Response: We examined carefully the language, and English corrections were made with the help of native English speaker.

Other major corrections made in the text: 1. Some affiliations changed because some authors were transferred to other institutions or because one of the Institutions changed name. 2. The abstract reformed in order to be more descriptive. 3. Any analysis related to finer resolutions below 0.5 degrees was removed from the text following the comments of reviewer 2. For this reason, the 30 arc-sec resolution maps given in Figs.2,3,4,5,6,7 were substituted with the ones of 0.5 degree resolution with respective changes in the range of values in their legends. Any discussion about the comparison of different resolutions was also removed from the discussion section. Additionally, all the results and tables changed based on 0.5 degree resolution. Similar changes were also made in the supplementary material. The only reference about the finer resolutions is given in section 5. Data availability, where we added the following text: "Apart from the 0.5 degree resolution raster datasets, the database contains the same datasets at finer resolution (30 arc-sec, 2.5 arc-min, 5 arc-min and 10 arc-min). These finer datasets are provided in order to cover the observed resolution range in the initial climatic data (e.g. the temperature data of Hijmans et al. (2005) are provided at 30 arc-sec resolution). The finer resolutions were produced using bilinear interpolation on solar radiation, humidity and wind speed data of Sheffield et al. (2006). This interpolation method is not the most appropriate for such purposes. The data of finer resolutions can only be used as a tool to assess uncertainties associated to temperature variation effects within a 0.5 degree pixel or to estimate average values of the coefficients for larger territories in order to capture a better representation of the coastlines or islands that do not exist in 0.5 degree resolution (use of values from individual pixels is not recommended). A complete list of the datasets is provided in the Table S5." 4. Reviewer 2 also commented that the manuscript is quite long. For this reason we removed the accuracy analysis by splitting the stations based on their elevation, and we also removed the Taylor diagrams analysis since the criteria that we give in Table 5 are more than enough. 5. The Discussion section was completely reformed based on the comments of Reviewer 1. 6. We added another 8 models of short reference crop evapotranspiration for comparative purposes after the request of Reviewer 1. 7. An

error was found in the coordinates of Australian station Paynes Find station (A-69) of the validation dataset and the associated coefficients extracted from the specific coordinates. The position of the station was corrected in Fig.1 and any information related to the station was corrected. An additional arithmetic error was found and corrected in the ETo ASCE estimations of Australian stations. We performed a detailed check for all stations data, all the calculations/equations used for rasters development, all the calculations/equations used for analyzing stations data.

References Abtew, W.: Evapotranspiration measurements and methoding for three wetland systems in South Florida. J. Am. Water Resour. Assoc. 32, 465-473, 1996. Ahooghalandari, M., Khiadani, M., Jahromi, M. E.: Developing Equations for Estimating Reference Evapotranspiration in Australia. Water Resour. Manage., 30, 3815-3828, 2016. Ahooghalandari, M., Khiadani, M., Jahromi, M. E.: Calibration of Valiantzas' reference evapotranspiration equations for the Pilbara region, Western Australia. Theor. Appl. Climat. 128, 845-856, 2017. Alexandris, S., Kerkides, P., Liakatas, A.: Daily reference evapotranspiration estimates by the "Copais" approach. Agr. Water Manage. 82:371-386, 2006. Azhar, A. H., Perera, B. J. C.: Evaluation of reference evapotranspiration estimation methods under Southeast Australian conditions. J. Irrig. Drain Eng., 137, 268-279, 2011. Castañeda, L., Rao, P.: Comparison of methods for estimating reference evapotranspiration in Southern California. J. Environ. Hydrol. 13, 1-10, 2005. Droogers, P., and Allen, R. G.: Estimating reference evapotranspiration under inaccurate data conditions. Irrig. Drain. Syst., 16, 33-45, 2002. Hargreaves, G. H., and Samani, Z. A.: Reference crop evapotranspiration from ambient air temperature. American Society of Agricultural Engineers, 12 pp, 1985. http://libcatalog.cimmyt.org/download/reprints/97977.pdf Hargreaves, G. H., and Allen, R. G.: History and evaluation of Hargreaves evapotranspiration equation. J. Irrig. Drain Eng. ASCE, 129 (1), 53-63, 2002. Peel, M. C., Finlayson, B. L., and McMahon, T. A.: Updated world map of the Köppen-Geiger climate classification. Hydrol. Earth Syst. Sci., 11, 1633-1644, 2007. Priestley, C. H. B., and Taylor, R. J.: On the assessment of surface heat flux and evaporation using large-scale parameters. Mon. Weather
Rev., 100, 81-92, 1972. Snyder, R. L., and Pruitt. W. O.: Estimating reference evapotranspiration with hourly data. VII-1-VII-3. R. Snyder, D. W. Henderson, W. O., Pruitt, and A. Dong (eds), Calif. Irrig. Mgmt. Systems, Final Rep., Univ. Calif., Davis, 1985. Snyder, R. L., and Pruitt. W. O.: Evapotranspiration data management in California. Presented at the Amer. Soc. of Civil Engr. Water Forum '92', Aug. 2-6, 1992, Baltimore, MD, 1992. Valiantzas, J. D.: Simple ETo forms of Penman's equation without wind and/or humidity data. I: Theoretical development. J. Irrig. Drain Eng., 139, 1-8, 2013a. Valiantzas, J. D.: Simplified reference evapotranspiration formula using an empirical impact factor for penman's aerodynamic term. J. Hydrol. Eng., 18, 108-114, 2013b. Valiantzas, J. D.: Closure to "Simple ETo forms of Penman's equation without wind and/or humidity data. I: Theoretical development" by John D. Valiantzas. J. Irrig. Drain Eng., 140, art. no. 07014017, 2014. Valipour, M.: Investigation of Valiantzas' evapotranspiration equation in Iran. Theor. Appl. Clim., 121, 267-278, 2015a. Valipour, M.: Evaluation of radiation methods to study potential evapotranspiration of 31 provinces. Meteorol. Atmos. Phys., 127, 289-303, 2015b. Valipour, M.: Analysis of potential evapotranspiration using limited weather data. Appl. Water Sci., 7, 187-197, 2017. Valipour, M., Gholami Sefidkouhi, M. A., and Raeini−Sarjaz, M.: Selecting the best model to estimate potential evapotranspiration with respect to climate change and magnitudes of extreme events. Agr. Water Manage., 180, 50-60, 2017.

Please also note the supplement to this comment:
http://www.earth-syst-sci-data-discuss.net/essd-2016-59/essd-2016-59-AC1-supplement.pdf

[Figure]

| Parameter | $T_{max}$ | $T_{min}$ | $R_s$ | RH | $u_2$ | P | $ET_o$ ASCE-short | $ET_o$ ASCE-tall |
|---|---|---|---|---|---|---|---|---|
| Unit | °C | °C | MJ m$^{-2}$ d$^{-1}$ | % | m s$^{-1}$ | mm month$^{-1}$ | mm month$^{-1}$ | mm month$^{-1}$ |
| Average | 25.3 | 11.4 | 18.8 | 56.4 | 2.6 | 41.5 | 138.4 | 190.5 |
| Minimum | 5.3 | -7.2 | 4.9 | 19.0 | 0.9 | 0.0 | 17.9 | 26.2 |
| Lower quartile | 19.7 | 6.5 | 13.5 | 45.5 | 1.8 | 11.7 | 82.2 | 112.7 |
| Upper quartile | 31.1 | 15.8 | 24.4 | 68.2 | 3.2 | 50.6 | 186.9 | 254.2 |
| Maximum | 41.2 | 26.3 | 30.1 | 90.3 | 6.8 | 470.4 | 377.5 | 563.8 |
| **Range** | **35.9** | **33.5** | **25.2** | **71.3** | **5.9** | **470.4** | **359.6** | **537.6** |
| Standard deviation | 7.1 | 6.4 | 6.5 | 15.4 | 1.0 | 51.5 | 69.5 | 98.9 |
| Coeff. of variation % | 28.11% | 56.13% | 34.32% | 27.36% | 37.05% | 123.90% | 50.17% | 51.93% |

**Fig. 1.** TABLE R1

**Fig. 2.** FIGURE R1

**Supplement:**

**Comment R1.1:** *I appreciate the Editor to give me a chance to review an interesting and valuable paper. I found some merits in the both methodology and results. In my opinion, this paper has a good potential to be published in the journal. However, I have also some concerns on the different parts of the manuscript. If only the author(s) address carefully to all of my comments, I'll recommend publication of the manuscript in the journal.*

**Response:** We would like to thank the reviewer for the constructive comments. We followed his suggestions for improving the manuscript. Our responses to specific comments are given below. We also suggest the reviewer to check carefully the responses to the comments of Reviewer 2, since his suggestions led to substantial changes in the manuscript.

**Comment R1.2:** *What was the criterion to select the stations? Why the USA and Australia? Are they covering all climates?*

**Response:** The reasons for choosing the CIMIS-database of California (USA) and AGBM database of Australia are:

- The first database includes stations from California-USA and it was selected because: a) it has been used as a basis for the development of Hargreaves-Samani method (Hargreaves and Samani, 1985; Hargreaves and Allen, 2002) and CIMIS method (Snyder and Pruitt, 1985, Snyder and Pruitt, 1992) and b) provides a dense and descriptive network of stations for a specific region that combines coastal, plain, mountain and desert environments (Table 1, Fig.1a in the manuscript). The second database includes stations from Australia and it was selected because the stations network covers a large territory with large variety of climate classes (Table 1, Fig.1b in the manuscript), but also because the Priestley-Taylor method has been calibrated for locations of eastern Australia (Priestley and Taylor, 1972). For the stations of AGBM database, the selection of stations was performed in such way in order to cover all the possible existing Köppen climatic types and elevation ranges of Australian continent (Table 1 in the manuscript). (see text in Page 9, lines 25-35, Page 10, lines 0-5)

- Additional reason for choosing these two databases was that they provide a large number of stations with complete data for estimating ASCE-$ET_o$ covering large observation periods before and after the year 2000 (Table 1 in the manuscript). This was a prerequisite in this study because the rasters of the new coefficients were developed based on mean monthly climatic parameters of 1950-2000. Thus, using stations with many available data after 2000, we could prove that the derived coefficients also work for current conditions. (see text in Page 10, lines 8-10)

- We have to mention that the combination of the databases provided a wide range of the mean monthly values of the parameters used for the $ET_o$ estimations (this is an additional reason for their selection). The general statistics of the aforementioned parameters are also given in Table R1 below, which was included in the supplementary material as Table S1 (reference for Table S1 exist in the text in Page 10, line 24). In order to show the high variability in the parameters of the validation dataset, we provide frequency diagrams of mean monthly $T_{max}$, $T_{min}$, $R_s$, $RH\%$, $u_2$ and $P$ are given below in Fig.R1a,b,c,d,e,f, respectively.

- Taking into account the last column of Table 1 (Köppen-Geiger classification), we provided the climatic classification of each station. According to Table 1, from the 140 stations, 9 belong to A Köppen-Geiger group (tropical/megathermal), 69 to B group (Arid/semi-arid), 59 to C group (temperate/mesothermal) and 3 to D group (continental/microthermal). We believe, that apart from the D group, the number of stations for the rest climatic groups are enough for validating the results. As concern the D group, we couldn't find more stations with

adequate data inside the aforementioned databases. Other databases, which may provide data for stations from other parts of the world, were not used in order to fully exploit the two aforementioned databases but also to give the opportunity to other scientists to test our revised coefficients for their territories using other complete databases and not selected stations from various databases. Many existing databases of observed data may show differences in the methods used for measuring and presenting data. Such differences were also observed in the CIMIS and AGBM databases and they were used to justify many uncertainties observed during the implementation of this work (see response to comment R2.3 of reviewer 2 and the new section in the discussion with title "*Uncertainties in the data used for calibrating and validating the revised coefficients of P-T and H-S methods*"). It would be difficult to identify such uncertainties using stations from multiple databases.

**Table R1.** General statistics* of the mean monthly observed values of climatic parameters from the 140 stations of California-USA and Australia that participate in the estimation of reference evapotranspiration with the ASCE method.

| Parameter | $T_{max}$ | $T_{min}$ | $R_s$ | RH | $u_2$ | P | $ET_o$ ASCE-short | $ET_o$ ASCE-tall |
|---|---|---|---|---|---|---|---|---|
| Unit | °C | °C | MJ m$^{-2}$ d$^{-1}$ | % | m s$^{-1}$ | mm month$^{-1}$ | mm month$^{-1}$ | mm month$^{-1}$ |
| Average | 25.3 | 11.4 | 18.8 | 56.4 | 2.6 | 41.5 | 138.4 | 190.5 |
| Minimum | 5.3 | -7.2 | 4.9 | 19.0 | 0.9 | 0.0 | 17.9 | 26.2 |
| Lower quartile | 19.7 | 6.5 | 13.5 | 45.5 | 1.8 | 11.7 | 82.2 | 112.7 |
| Upper quartile | 31.1 | 15.8 | 24.4 | 68.2 | 3.2 | 50.6 | 186.9 | 254.2 |
| Maximum | 41.2 | 26.3 | 30.1 | 90.3 | 6.8 | 470.4 | 377.5 | 563.8 |
| **Range** | **35.9** | **33.5** | **25.2** | **71.3** | **5.9** | **470.4** | **359.6** | **537.6** |
| Standard deviation | 7.1 | 6.4 | 6.5 | 15.4 | 1.0 | 51.5 | 69.5 | 98.9 |
| Coeff. of variation % | 28.11% | 56.13% | 34.32% | 27.36% | 37.05% | 123.90% | 50.17% | 51.93% |

*The statistics are based on 1680 values (140 stations × 12 months)

[Figure]

**Fig.R1** Frequency diagrams (number of monthly values) based on the mean monthly data of the 140 stations of CA-USA and Australia for a) maximum temperature $T_{max}$, b) minimum temperature $T_{min}$, c) solar radiation $R_s$, d) relative humidity $RH$, e) wind speed $u_2$ at 2 m height and f) precipitation $P$. The frequencies are based on a total number of observations equal to 1680 (140 stations × 12 months).

**Comment R1.3:** *Lns 7-8, cite also these three useful papers to enhance the literature:*
*- Selecting the best model to estimate potential evapotranspiration with respect to climate change and magnitudes of extreme events.*
*- Temporal analysis of reference evapotranspiration to detect variation factors.*
*- Analysis of potential evapotranspiration using limited weather data.*
**Response:** The proposed citations were added at the proposed locations in the text and in the references list.

**Comment R1.4:** *Lns 15-16, cite also these two useful papers to enhance the literature:*
*- Application of new mass transfer formulae for computation of evapotranspiration*
*- Ability of Box-Jenkins Models to Estimate of Reference Potential Evapotranspiration (A Case Study: Mehrabad Synoptic Station, Tehran, Iran)*
**Response:** The proposed citations were added at the proposed locations in the text and in the references list.

**Comment R1.5:** *In the last paragraph of the Introduction, the authors should clearly mention the weakness point of former works (identification of the gaps) and describe the novelties of the current investigation to justify us the paper deserves to be published in this journal.*
**Response:** The most significant novelty of the study is that provides, for the first time, global maps of revised coefficients for the P-T and H-S evapotranspiration methods and revised coefficients for the H-S radiation formula. Such attempt has never been made in the past at the global scale, despite the fact that many studies for recalibrating the respective coefficients have been presented for many parts of the world. The final maps allow the comparison of the revised coefficients among regions under a common base since they were built using common datasets and using the same technique (Eqs.7), while they provide a global overview of the variation in these coefficients.
Other novelties of the study are:

- the development of global maps for the revised coefficients for P-T and H-S evapotranspiration methods for tall reference crop.
- the development of global maps for the possible mean annual error, when the P-T and H-S are applied using the standard coefficients of the original methods (maps of *MAD%* parameter, Fig.4b,c,d in the manuscript). These maps provide information about the uncertainty when the standard H-S and P-T methods are used. These maps were also combined to derive a new map (map of *DMAD* parameter, Fig.5a in the manuscript), which indentifies the optimum locations for the application of the standard H-S and P-T formulas based on their proximity to the results of ASCE for short reference crop. The *DMAD* map is an important tool, which can give a solution when someone has to choose between the two methods.
- the proposal of a method for deriving annual coefficients for the P-T and H-S methods. The procedure described by the set of Eqs.7, which estimates the partial weighted averages of the coefficients based on their monthly values, is a newly proposed method that can be easily applied in GIS environment, while it provides a solution when annual coefficients have to be derived under a common base for many stations or for global applications using raster data. This technique is proposed as an alternative of optimization methods, which are difficult to be incorporated in GIS environment (see more details about this comment in Page 18, lines 13-24 and the respective subsection of the

discussion). The method calibrates the basic coefficients without modifying or adding parameters in the original P-T and H-S equations.

Finally, we have to stress that the aim of the study is to provide tools for facilitating the estimation of $ET_o$ and solar radiation for regions (especially those of developing countries), which face serious shortage of climatic data and not to propose the revised coefficients as an alternative method for regions, which have complete meteorological stations that provide detailed sets of all climatic variables. Of course, the use of such stations to validate the coefficients is the only solution. All the aforementioned aspects related to the novelties and the aims of the study are included in brief in the revised paragraph at the end of the introduction.

**Comment R1.6:** *Compare the results with modified/calibrated H-S and P-T models presented by other researchers in all of the world (particularly in the USA and Australia).*
**Response:** The requested task is quite difficult and there are many problems of comparability since either the majority of authors have modified the initial form of H-S and P-T models and not only the main coefficients, but also because the majority of the works provide H-S and P-T models, which are calibrated for regions outside California and Australia. For example, the popular recalibrated Priestley-Taylor model of Abtew (1996) for Florida-USA uses a recalibrated coefficient equal to 1.18, which is almost equal to our revised coefficient (1.17 from 0.5 degree resolution), but the model gives bad results in the validation procedure because Florida has a completely different environment from California and cannot cover the climatic variation of Australia stations. Other examples are the modified Makking model and other models given by Castañeda and Rao (2005), which were calibrated only for one station of southern California using 4 years of observations and the modified Hargreaves-Samani models and other models of Azhar and Perera (2011), which were recalibrated for three stations of southeastern Australia with very few years of observations. Similar problems were observed for too many other cases that we examined using a large list of models provided by Valipour (2015a,b; 2017) and Valipour et al. (2017) and by models obtained from the works cited in the introduction. Thus, it is unfair to examine the accuracy of such models using the complete validation dataset (both California and Australian stations), while it is not feasible in the context of this article to examine one by one all the modified models published in the international literature. Of course, the analysis of modified H-S and P-T models calibrated for other parts of the world was rejected from the beginning because the validation dataset does not include stations from these regions. The only models, which are absolutely comparable are the modified H-S models by Droogers and Allen (2002) because they have been calibrated using global datasets. In order to partly satisfy the request of the reviewer, we selected some models of reduced parameters, which have similar or additional requirements from the standard H-S and P-T models. The final selected models were also those who showed a) the best performance after examining an extremely large list of models using both California-USA and Australia stations data and b) a good performance to other studies using other datasets. The comparison with such models will also contribute to verify the value of our coefficients as alternative options for $ET_o$ estimations with fewer variables.

Based on the aforementioned observations, the following 8 models were selected for comparisons with the standard and re-adjusted H-S and P-T models:

1. Two modified models of H-S by Droogers and Allen (2002) where the second one uses precipitation as additional parameter. The models were based on calibrations using global data.

2. Three models of reduced parameters given by Valiantzas (2013a,b; 2014) that were calibrated using 535 stations from Europe, Asia, Africa. The first model uses temperature and radiation data, while the other two use temperature, radiation, and humidity data. The models have been tested for California (Valiantzas, 2013c) and Australia conditions (Ahooghalandari et al., 2017).

3. Two models of reduced parameters by Ahooghalandari et al. (2016) calibrated using 18 stations from various locations of Australia. The models use temperature and relative humidity data. Ahooghalandari et al. (2017) also made recalibration of Valiantzas equations and other models but for a restricted region of Western Australia considering 8 stations, and for this reason these modified versions were not used.

4. The Copais model of Alexandris et al. (2006) that uses temperature, radiation and humidity data. The model was calibrated/validated using data from Greece, California and Oregon-USA, while it has shown a very good response to many other regions of the world including Australia (Ahooghalandari et al., 2017).

The aforementioned observations and the description of the additional models used for comparisons were added as a new paragraph in the page 10 (lines 32-35) and 11 (lines 0-25) together with the new Table 2, which gives the equations of the additional models. The results of the models were added in Page 14 (lines 25-35), and 15 (lines 0-15). The comparisons of the models were included in new Fig.8 and their statistics in the new Table 5.

**Comment R1.7:** *The Discussion section should be broken to sub-sections for better understanding of readers.*
**Response:** we followed the suggestion of the reviewer and the discussion section was broken to sub-sections.

**Comment R1.8:** *Explain the variations of the spatial extent of the major climatic groups CGs from Köppen-Geiger climate map.*
**Response:** The variations of the spatial extent of the major climatic groups CGs from Köppen-Geiger climate map, which are given in the first column of Table 4 in the revised manuscript, are results obtained from the respective raster Köppen-Geiger climate map of Peel et al. (2007) and they are described in detailed in their article. We believe that is beyond the scope of the paper to discuss the observed % of CGs in the map of Peel et al. (2007) (keep in mind that % values are re-adjusted after excluding Antarctica and they are slightly different from those given by Peel). If the reviewer means to explain the results of the other columns of new Table 4 (i.e. why for example the typical H-S method is better from the typical P-T method in more arid environments or why P-T is better in more humid environments) the explanation is based on the fact that the standard H-S method was calibrated for the conditions of California, which include arid and semi-arid environments, while the standard coefficient 1.26 of Priestley-Taylor method was obtained based on experiments of more humid environments. This justification was added in the text (see Page 13, lines 8-11).

**Comment R1.9:** *Discuss the comparison of the average and standard deviation of RMSEs of the validation dataset between different pixel Resolutions more thoroughly.*
**Response:** The comparison between different pixel resolutions was removed based on the objections of Reviewer 2. Reviewer 2 suggested not to use the resolutions below 0.5 degree due to interpolation limitations in the initial raster data for solar radiation, humidity and wind speed, which were at 0.5 degree. See comments of Reviewer 2. All the manuscript was revised from the beginning using only the results of 0.5 degree resolution.

**Comment R1.10:** *What are the strategies/recommendations to reduce uncertainties in this study?*
**Response:** a new subsection was added in the Discussion. See the new subsection with title "*Recommendations for reducing the uncertainties when the re-adjusted coefficients of P-T and H-S models are used*".

**Comment R1.11:** *At the end of the manuscript, explain the implications and future works considering the outputs of current study.*
**Response:** we added new text about it (see the last paragraph in the Conclusions section)

**Comment R1.12:** *The quality of the language needs to improve by a native English speaker for grammatically style and word use.*
**Response:** We examined carefully the language, and English corrections were made with the help of native English speaker.

**Other major corrections made in the text:**
1. Some affiliations changed because some authors were transferred to other institutions or because one of the Institutions changed name.
2. The abstract reformed in order to be more descriptive.
3. Any analysis related to finer resolutions below 0.5 degrees was removed from the text following the comments of reviewer 2. For this reason, the 30 arc-sec resolution maps given in Figs.2,3,4,5,6,7 were substituted with the ones of 0.5 degree resolution with respective changes in the range of values in their legends. Any discussion about the comparison of different resolutions was also removed from the discussion section. Additionally, all the results and tables changed based on 0.5 degree resolution. Similar changes were also made in the supplementary material. The only reference about the finer resolutions is given in section 5. Data availability, where we added the following text: *"Apart from the 0.5 degree resolution raster datasets, the database contains the same datasets at finer resolution (30 arc-sec, 2.5 arc-min, 5 arc-min and 10 arc-min). These finer datasets are provided in order to cover the observed resolution range in the initial climatic data (e.g. the temperature data of Hijmans et al. (2005) are provided at 30 arc-sec resolution). The finer resolutions were produced using bilinear interpolation on solar radiation, humidity and wind speed data of Sheffield et al. (2006). This interpolation method is not the most appropriate for such purposes. The data of finer resolutions can only be used as a tool to assess uncertainties associated to temperature variation effects within a 0.5 degree pixel or to estimate average values of the coefficients for larger territories in order to capture a better representation of the coastlines or islands that do not exist in 0.5 degree resolution (use of values from individual pixels is not recommended). A complete list of the datasets is provided in the Table S5."*
4. Reviewer 2 also commented that the manuscript is quite long. For this reason we removed the accuracy analysis by splitting the stations based on their elevation, and we also removed the Taylor diagrams analysis since the criteria that we give in Table 5 are more than enough.
5. The Discussion section was completely reformed based on the comments of Reviewer 1.
6. We added another 8 models of short reference crop evapotranspiration for comparative purposes after the request of Reviewer 1.

7. An error was found in the coordinates of Australian station Paynes Find station (A-69) of the validation dataset and the associated coefficients extracted from the specific coordinates. The position of the station was corrected in Fig.1 and any information related to the station was corrected. An additional arithmetic error was found and corrected in the $ET_o$ ASCE estimations of Australian stations. We performed a detailed check for all stations data, all the calculations/equations used for rasters development, all the calculations/equations used for analyzing stations data.

was found significantly different between seasons and it was negatively correlated to net solar radiation and/or temperature. The general trends of $a_{pt}$ led to the conclusion that colder-drier conditions due to low net radiation and high vapour pressure deficit tend to increase its values.

The Hargreaves-Samani (H-S) method requires only temperature data, including four empirical factors (or three depending on the formula). A part of the equation empirically describes the incident solar radiation $R_s$. A basic problem of the Hargreaves-Samani method is that it tends to underestimate $ET_o$ under high wind conditions ($u_2$>3 m s$^{-1}$) and to overestimate $ET_o$ under conditions of high relative humidity (Allen et al., 1998). The last years, many scientists have performed analysis and re-calibration of the Hargreaves-Samani method for various climates (Trajkovic, 2007; Tabari, 2010; Tabari and Talaee, 2011; Azhar and Perera, 2011; Aschonitis et al., 2012; Mohawesh and Talozi, 2012; Rahimikhoob et al., 2012; Ravazzani et al., 2012; Bachour et al., 2013;  Long et al., 2013; Mendicino and Senatore, 2013; Ngongondo et al., 2013; Berti et al., 2014; Heydari and Heydari, 2014), which indicates a global interest for simplified methods, mainly driven by the lack of data.

The analysis of $ET_o$ at global scale is of special interest since it provides a general view about the spatiotemporal variation of this parameter, while (together with rainfall) provides significant information about the aridity of terrestrial systems. A basic limitation of global analysis is the lack of homogeneously distributed meteorological stations around the globe and especially in mountainous regions. The last years, climatic models, advanced interpolation and other methods have succeeded to generate datasets of various climatic parameters (Hijmans et al., 2005; Sheffield et al., 2006; Osborn and Jones, 2014; Brinckmann et al., 2016), facilitating the attempts to develop $ET_o$ maps. Significant works of global $ET_o$ estimations have been performed from various scientists. Mintz and Walker (1993) used the Thorthwaite (1948) method and provided isoline maps of $ET_o$. Tateishi and Ahn (1996) used the Priestley-Taylor method and provided $ET_o$ maps at 0.5 degree resolution. Droogers and Allen (2002) used FAO-56 Penman-Monteith method, providing $ET_o$ maps at 10 arc-min resolution and a modified Hargreaves-Samani method, which considers rainfall. Weiß and Menzel (2008) compared four different methods (Priestley-Taylor, Kimberly Penman, FAO-56 Penman-Monteith and Hargreaves-Samani) and provided $ET_o$ maps at 0.5 degree resolution. Zomer et al. (2008) used Hargreaves-Samani method and provided the highest resolution (30 arc-sec) available for $ET_o$ maps.

The objectives of the study are: a) to develop mean monthly maps of $ET_o$ for the period 1950-2000 at global scale using the most precise ASCE-standardized method for both reference crops (short clipped grass and tall alfalfa); b) to develop global maps that provide the possible annual error in $ET_o$ estimations using the standard P-T and H-S evapotranspiration methods in comparison to ASCE method for short reference crop and the possible annual error in solar radiation estimations using the temperature-based H-S radiation formula (this attempt will allow to identify the optimum locations for the application of the standard H-S and P-T evapotranspiration formulas based on their proximity to the results of ASCE for short reference crop); c) to develop global maps of re-adjusted annual coefficients for the H-S and P-T evapotranspiration methods for both short and tall reference crop based on a new method that estimates partial weighted averages of the monthly coefficients (the same procedure was also followed for the coefficients of the H-S radiation

formula); d) to validate the results of the re-adjusted P-T and H-S coefficients using data from meteorological stations from different locations with different climatic conditions; and e) to compare the predictive ability of the re-adjusted P-T and H-S coefficients for short reference crop evapotranspiration with the respective predictions obtained from other models that have low data requirements. The analysis and the produced datasets of this study were based on mean monthly climatic data of 0.5

5    degree resolution for the period 1950-2000. The final datasets of revised H-S and P-T coefficients will provide a global overview of the variation in their values and a common base for comparing the values of different regions since they are calibrated using common datasets and using the same technique. The produced global datasets of this study can support estimations of $ET_o$ and solar radiation for locations where climatic data are limited while it can support studies, which require such estimations at larger scales (e.g. country, continent, world).

10  **2 Data and methods**

**2.1 Global climatic data**

   The analysis presented in this study was based on global climatic data obtained from the following databases:

- The database of Hijmans et al. (2005) provides mean monthly values for the parameters of precipitation, maximum, minimum and mean temperature at 30 arc-sec spatial resolution. The data are provided as grids of mean monthly values

15    of the period 1950-2000 (http://www.worldclim.org/). The database also includes a revised version of the GTOPO30 DEM based on SRTM DEM at 30 arc-sec spatial resolution, which was used for the estimation of atmospheric pressure. The DEM was also used as a base to calculate the distance from the coastlines in raster format at 30 arc-sec spatial resolution based on the Euclidean distance.

- The database of Sheffield et al. (2006) provides monthly values of parameters such as wind speed at the height of 10 m

20    above the ground surface, solar radiation, specific humidity, precipitation and temperature for the period 1948-2006 at 0.5 degree spatial resolution. The data are available in the form of netcdf files of monthly values of each year for the period 1948-2006 (http://hydrology.princeton.edu/data.pgf.php).

- The database of Peel et al. (2007) provides the revised global Köppen-Geiger climate map. The data are provided in raster form with 0.1 degree spatial resolution. The climate map was developed using the GHCN version 2.0 dataset

25    (Peterson and Vose, 1997), which includes precipitation data from 12396 stations and temperature data from 4844 stations data for the periods 1909-1991 and 1923-1993, respectively. The Köppen-Geiger map was used to obtain the climatic type of the meteorological stations used in the validation dataset.

   In this study, the $ET_o$ is estimated combining the databases of Hijmans et al. (2005) and Sheffield et al. (2006), as follows: a) mean monthly values of maximum, minimum, mean temperature and precipitation were obtained from Hijmans

30  et al. (2005); while b) wind speed, specific humidity and incident solar radiation were obtained from Sheffield et al. (2006) database. The specific humidity was converted to actual vapour pressure using the equation given by Peixoto and Oort (1996). The final results and analysis presented in this study is based on the coarser 0.5 degree resolution.

[revised manuscript text omitted]

stations for a specific region that combines coastal, plain, mountain and desert environments (Table 1, Fig.1a). The second database includes stations from Australia and it was selected because the stations network covers a large territory with large variety of climate classes (Table 1, Fig.1b) but also because the Priestley-Taylor method has been calibrated for locations of eastern Australia (Priestley and Taylor, 1972). The selection of stations from AGBM database was performed in such way in

5    order to cover all the possible existing Köppen climatic types and elevation ranges of Australian continent (Table 1). In total, 140 stations were used, 60 stations were selected from CIMIS and 80 stations from the AGBM that have at least 15 years of observations (some stations, that do not follow this rule, were selected due to their special climate Köppen class or the high elevation of their location). Observations from years after 2000 up to 2016 were included (when they were available) in the stations data, in order to show that the new revised coefficients are applicable for recent periods.

10                                                    **[FIGURE 1]**

**[TABLE 1]**

   In the case of CIMIS stations, the monthly data for all climatic parameters were obtained, including $ET_o$ estimations using the CIMIS method (Snyder and Pruitt, 1985, Snyder and Pruitt, 1992), but they required quality control before their use. Quality control signs are provided by the database for all climatic data, indicating extreme values, while possible errors

15   are flagged but they are not automatically excluded. For this reason, the user should consider the signs in order to prepare a robust dataset. For this study, proper control was performed and very extreme or erroneous monthly values were excluded. Excluded values were less than 1‰ of the total values of all stations and all parameters. The final clean dataset was subjected to a secondary but indirect quality control through the comparison between the estimated mean monthly values of $ET_o$ of ASCE-short method (Eq.1) using the clean climatic data of all USA-CA stations versus the respective mean monthly

20   $ET_o$ values given by CIMIS database (linear regression result between mean monthly values for $n$ obs.=12×60=720: y=0.994x-1.07 with $R^2$=0.98) (see Fig.S1 in the supplementary material). Data cleaning was not followed in the case of Australia stations, since the AGBM database provides the mean monthly values of the climatic parameters for the total periods of observation and not for individual years. The general statistics of the mean monthly observed values of climatic parameters obtained from the 140 stations of California-USA and Australia are given in Table S1 of the supplementary

25   material. A comparison of $T_{max}$, $T_{min}$, $R_s$, $R_n$, $DE$ (vapour pressure deficit) and $u_2$ parameters between the rasters (0.5 degree) and the stations data are provided in Figs.S2a,b,c,d,e,f of the Supplementary material.

   The validation procedure was performed using the data of the stations in Table 1 by comparing the mean monthly values of $ET_o$ derived by the P-T (Eq.2) and H-S (Eq.4b,c) methods with the standard $a_{pt}$ and $c_{hs2}$ coefficients and with the re-adjusted ones versus the ASCE method for short reference crop (Eq.1). The same procedure was also performed for the

30   new $a_{pt}$ and $c_{hs2}$ coefficients for the tall reference crop and for the re-adjusted coefficient $K_{RS}$ in the radiation formula of H-S (Eq.3). For the case of ASCE method for short reference crop, additional models of reduced parameters were used from the literature in order to perform comparisons with the standard and re-adjusted P-T and H-S models. The selection of these models was made in such way in order to satisfy the following criteria/characteristics:

● The selected models have been calibrated either using global data or a representative amount of data from California or

Australia. Models that have been tested for California and Australia and showed good performance were also included.

- The selected models showed better performance when tested using the validation datasets of California and Australia stations in comparison to other tested models but also a good performance to other regions based on studies from the literature. It has to be mentioned that an extremely large amount of models were examined taking into account the modified H-S and P-T models obtained from works that have been already mentioned in the introduction and the large lists of models presented in the works of Valipour (2015a,b; 2017) and Valipour et al. (2017). Strict modifications of P-T and H-S models with fixed coefficients calibrated for local conditions were not used because they cannot adapt their coefficients to the large climatic variability of the validation dataset.

- The majority of the selected models require additional parameters in comparison to P-T and H-S models. This criterion was used in order to compare the strength of the re-adjusted P-T and H-S coefficients versus such models.

Based on the aforementioned criteria, the following eight models were selected for comparisons with the standard and re-adjusted H-S and P-T models (Table 2):

- Two modified models of H-S by Droogers and Allen (2002), where the second one uses precipitation as additional parameter. The models were calibrated using global data.

- Three models of reduced parameters given by Valiantzas (2013a,b; 2014), which were calibrated using 535 stations from Europe, Asia, Africa. The first model uses temperature and radiation data, while the other two use temperature, radiation, and humidity data. The models have been tested for California (Valiantzas, 2013c) and Australia conditions (Ahooghalandari et al., 2017).

- Two models of reduced parameters by Ahooghalandari et al. (2016) calibrated/validated using stations from various locations of Australia. The models use temperature and relative humidity data.

- The Copais model of Alexandris et al. (2006) that uses temperature, radiation and humidity data. The model was calibrated/validated using data from Greece, California and Oregon-USA while it has shown a very good response to many other regions of the world including Australia (Ahooghalandari et al., 2017).

**[TABLE 2]**

[revised manuscript text omitted]

[FIGURE 6]

[FIGURE 7]

As regards the spatial variation of $a_{pt}$ for short reference crop (Fig.6a), the higher values were observed in extremely arid and desert environments exceeding the value of 1.8 (due to extremely high vapour pressure deficit), while the extremely cold and extremely humid environments presented values <1.0. Interesting cases are the alpine-tundra and extreme humid tropical environments, which presented similar values between ~0.8-1.0, due to the low values of vapour pressure deficit. Values of $a_{pt}$ below 0.8 were observed in sub-polar areas. The spatial variation of $a_{pt}$ for tall reference crop (Fig.6b) follows

similar patterns with $a_{pt}$ of short reference crop but with increased values, which can be described by the following relationship $a_{pt(p.w.a.t.)}=1.73 \cdot a_{pt(p.w.a.s.)} - 0.58$, $R^2=0.996$, $p<0.0001$. This relationship is valid for $a_{pt(p.w.a.s.)}>0.8$ for preserving $a_{pt(p.w.a.t.)}\geq a_{pt(p.w.a.s.)}$.

As regards the spatial variation of $c_{hs2}$ for short reference crop (Fig.6c), the higher values were observed in extremely
5    arid and desert environments exceeding 0.0026 (due to extremely high vapour pressure deficit), while the extremely cold and extremely humid environments presented values <0.0018. Similarities appear again in the case of alpine-tundra and extreme humid tropical environments, which presented values between ~0.0014-0.0018, due to the low values of vapour pressure deficit. Values of $c_{hs2}$ below 0.0014 were observed in sub-polar areas. The spatial variation of $c_{hs2}$ for tall reference crop (Fig.6d) follows similar patterns with $c_{hs2}$ of short reference crop but with increased values, which can be described by the
10    following relationship $c_{hs2(p.w.a.t.)}=1.793 \cdot c_{hs2(p.w.a.s.)} - 0.00114$, $R^2=0.967$, $p<0.0001$. This relationship is valid for $c_{hs2(p.w.a.s.)}>0.0014$ for preserving $c_{hs2(p.w.a.t.)}\geq c_{hs2(p.w.a.s.)}$.

In the case of $K_{RS}$ (Fig.7), extreme deviations from the value of 0.17 were observed in Greenland with values above 0.21 and in south-east China with values below 0.13 (regions of Chongqing, Guizhou, Hunan, Jiangxi, Guangxi). The spatial variation of $K_{RS}$ does not follow a specific pattern in relation to climate zones, while in many cases, it was observed an
15    increasing trend of its values closer to the coastlines (Fig.7). Additional observations about the effect of distance from the coastline $Dc$ on $K_{RS}$ are given in the discussion section.

**3.3 Validation of the re-adjusted $a_{pt}$, $c_{hs2}$ and $K_{RS}$ coefficients**

The validation of the re-adjusted $a_{pt}$, $c_{hs2}$ coefficients for $ET_o$ estimations (for both reference crops) and the $K_{RS}$
20    coefficient for $R_s$ was performed taking into account the mean monthly values of the climatic parameters of all stations from Table 1. The re-adjusted coefficients for each station obtained from the 0.5 degree resolution maps are given in Table S2 of the Supplementary material while the comparison between $ET_o$ estimations (for both reference crops) between rasters and stations is provided in Figs.S2g,h, respectively. The comparison of different methods is described in the next paragraphs, while the overall results of the statistical criteria for all the examined cases are given in Table 5.
25    **[TABLE 5]**

Table 5a and Fig.8 show the $ET_o$ (mm month$^{-1}$) comparisons between the ASCE-short values versus the values of the P-T and H-S methods with the standard and the re-adjusted (p.w.a.s.) coefficients and versus the values of the additional models given in Table 2. From the results of Fig.8 together with the results of the statistical criteria (Table 5a), the following observations were derived:
30    • The P-T(p.w.a.s.) and H-S(p.w.a.s.) models (Fig.8b,d) outperformed to all the statistical criteria (Table 5a) in comparison to the respective standard P-T(1.26) and H-S(0.0023) models (Fig.8a,c) reducing the $RMSE$ values at 40 and 25%, respectively.

- The comparison of statistical criteria between H-S(0.0023), H-S(p.w.a.s.), DRAL1 and DRAL2, which follow the general formula of H-S method and are based on calibrations with global data, showed the following order of accuracy H-S(p.w.a.s.)>DRAL1> DRAL2>H-S(0.0023).

- The standard P-T(1.26) showed the worst results to all criteria (Table 4a), while the use of P-T(p.w.a.s.) succeeded to improve the predictions giving better results from H-S(0.0023), DRLA2, VAL1 and AKJ2 models.

- The H-S(p.w.a.s.) provided better results from DRAL1, DRAL2, VAL1, AKJ1, AKJ2 where the latter four require data for more climatic parameters.

- The order of accuracy of the models was the following: VAL3>VAL2>Copais>H-S(p.w.a.s.)>AKJ1>P-T(p.w.a.s.)>DRAL1> DRAL2>H-S(0.0023)>AKJ2>VAL1>P-T(1.26) (the order was based on absolute comparisons of the accuracy rankings for each criterion, see Table S3 in Supplementary material). The *RMSE* difference between H-S(p.w.a.s.) and the best VAL3 model was 6.8 mm month$^{-1}$ (or 0.23 mm d$^{-1}$), while the respective difference between P-T(p.w.a.s.) and VAL3 was 13.5 mm month$^{-1}$ (or 0.45 mm d$^{-1}$). These differences are satisfactory, especially for the case of H-S(p.w.a.s.), which uses less climatic data from VAL3. Of course, these differences are even smaller when compared to VAL2 and Copais, which also use more climatic parameters. Justifications for the less satisfactory performance of P-T(p.w.a.s.) are given in the Discussion section.

**[FIGURE 8]**

Table 5b and Fig.9a,b show the $ET_o$ (mm month$^{-1}$) comparisons between the ASCE-tall values versus the values of P-T and H-S method using the readjusted $a_{pt}$ and $c_{hs2}$ coefficients for tall reference crop (p.w.a.t.), respectively. Since there are not currently other methods of reduced parameters calibrated based on ASCE $ET_o$ for tall reference crop, the comparison is restricted between the two methods. The results of Fig.9a,b together with the results of the statistical criteria (Table 5b) indicate a better performance of the H-S (with $c_{hs2}$=p.w.a.t.). The higher errors observed in H-S(p.w.a.t.) and P-T(p.w.a.t) in comparison to the respective errors of H-S(p.w.a.s.) and P-T(p.w.a.s) for short reference crop is justified by the fact that ASCE-tall is significantly higher from ASCE-short, especially in the drier environments (ASCE-tall was found ~28% higher from ASCE-short at global scale based on the mean values given in Fig.2a,b, and ~38% higher based on the comparison of the total mean values estimated by the California-USA and Australia stations data).

**[FIGURE 9]**

Table 5c and Fig.10a,b show the comparisons between the $R_s$ (MJ m$^{-2}$ d$^{-1}$) of stations data versus the respective values of standard radiation formula of H-S (Eq.3) with $K_{RS}$= 0.17 and with $K_{RS}$= p.w.a, respectively. The results of Fig.10a,b together with the results of the statistical criteria (Table 5c) indicate a better performance of the H-S $R_s$ with $K_{RS}$= p.w.a. even though the performance of the standard H-S $R_s$ is also satisfactory.

**[FIGURE 10]**

**4. Discussion**

*Uncertainties in the data used for calibrating and validating the revised coefficients of P-T and H-S methods*

The re-calibrated coefficients of the H-S and P-T methods were estimated using raster datasets that cover the period 1950-2000 assuming stationary climate conditions, while the validation datasets of California-USA and Australia stations are expanded up to 2016. Recent studies have shown changes/anomalies after 2000 in temperature (Hansen et al., 2010; Sun et al., 2017), solar radiation (Wild et al., 2013), wind speed (McVicar et al., 2012a,b) and atmospheric humidity (Willet et al., 2014) and such changes could affect the validity of the revised coefficients. The comparisons of $T_{max}$, $T_{min}$, $R_s$, $R_n$, $DE$ (vapour pressure deficit), and $u_2$ values between the rasters data and the stations data, showed a very good correspondence for the case of $T_{max}$, $T_{min}$, $R_s$, $R_n$ (Fig.S2a,b,c,d) and a relatively good correspondence for the case of $DE$ (Fig.S2e). In the case of $u_2$, a discrepancy was observed between the rasters and stations data (Fig.S2f). The separate examination of $u_2$ for the CA-USA and Australia stations (Fig.S3), showed that the total average of mean monthly $u_2$ values of CA-USA stations was lower from the rasters data of Sheffield et al. (2006) (data extracted from the stations' positions) while the opposite trend was found for $u_2$ values of Australia stations. This discrepancy between the $u_2$ values of rasters and stations can be justified by:

- Possible changes in wind speeds after 2000, since the majority of wind speed data in the stations datasets correspond to periods after 2000.

- Uncertainties in the Sheffield et al. (2006) wind data due to the scarce existing wind data for calibrating their model at global scale during the period of 1950-2000 and especially during the years belonging to the first half of the simulation period.

- The effect of the equation $u_2=4.87u_z/\ln(67.8z-5.42)$ (Allen et al., 1998; 2005), which was used to adjust the wind rasters of Sheffield et al. (2006) and the wind data of Australia stations from z=10 to 2 m height. The degree of accuracy of the aforementioned equation to convert wind data at 2 m is unknown. This equation is usually not calibrated for meteorological stations with anemometers positioned above 2 m height, while the uncertainty is even larger when is applied at global scale and for a pixel of 0.5 degree resolution, which may contain high topographic variability.

- The bias that may have been introduced after cleaning extreme wind values in the data of CA-USA stations, which may be associated to hurricane events. The region of California is strongly affected by hurricanes and the higher wind speeds in the rasters of Sheffield et al. (2006) data may partly occurred because they have included such events in their climatic simulations.

- The bias that may have been introduced by the wind data of Australia stations. The AGBM database (Australian Government – Bureau of Meteorology) provides 12 values of mean monthly wind speeds of the total observation periods for 9am and another 12 values for 3pm local time. In order to get the mean monthly wind speeds of the stations, the average value of 9 am and 3 pm conditions was used for each month.

- Combinations of all the aforementioned cases.

Uncertainties may also exist in the case of $DE=e_s-e_a$ (Fig.S2e), since Sheffield et al. (2006) provides data of specific humidity that were directly converted to actual vapour pressure $e_a$ using the equation of Peixoto and Oort (1996), which uses the additional parameter of atmospheric pressure as internal parameter. The atmospheric pressure in the case of rasters was estimated based on elevation data of 1 km resolution (30 arc-sec), which were further converted to 0.5 degree resolution. The use of $e_a$ data from 0.5 degree resolution pixels may also added additional error, especially when there is large topographic variability within the 0.5 degree pixel. On the other hand, the $e_a$ of stations was estimated by relative humidity and temperature data.

Thus, uncertainties exist in both rasters and stations data. In future studies, further improvements in the revised coefficients can be made by using global raster data, which incorporate the conditions after 2000, and by solving many of the aforementioned problems related to both stations data and raster data produced by climatic models.

*Reasons for using annual p.w.a. coefficients instead of monthly or seasonal ones in the case of H-S and P-T methods*

The analysis presented in this study passed through various stages before the selection of the annual p.w.a. form of the coefficients (Eqs.7). Some steps in the preliminary analysis were to analyse: (a) the different forms of averages (e.g. mean, mode, median, geometric mean, harmonic mean etc) for deriving annual coefficients, and (b) the strength of the derived mean monthly and seasonal coefficients versus the annual p.w.a. coefficients and versus the coefficients of the standard methods,.

As regards the use of weighted annual average (w.a.) of the mean monthly coefficients instead of other forms of averages (e.g. mean, mode, median, geometric mean - g.m., harmonic mean – h.m.), preliminary analysis was performed using data extracted by the climatic rasters from many positions of the world. During this analysis, trials to derive annual coefficients were made using an optimization algorithm separately for each position. The results showed that the optimized annual values were always closer to the monthly coefficients of the warmer months since the optimization algorithms try to reduce the total error, which is mainly dominated by the months that show larger $ET_o$ values (or $R_s$ for the case of $K_{RS}$ calibration). The optimized values were also compared to the different types of annual averages (e.g. mean, mode, median, g.m., h.m., w.m.), which were estimated after excluding values of monthly coefficients associated to months with $ET_o$ and $R_s$ values <45 mm month$^{-1}$ (for $R_s$ the equivalent is 3.61 MJ m$^{-2}$ d$^{-1}$) The w.a. outperformed in all cases because it is the only form that considers the amplitude of the parameter under investigation ($ET_o$ and $R_s$) (Eq.7), giving more weight to the monthly coefficients that are related to the warmer months. This attribute of w.a. is extremely significant since it is the only type that considers the seasonal observed differences in monthly $ET_o$ (for $a_{pt}$ and $c_{hs2}$) and $R_s$ (for $K_{RS}$) minimizing the possible errors during warmer months.

The case of mean monthly coefficients was also examined (results not shown). The results showed that the assessment of annual $ET_o$ and seasonal $ET_o$ during the warm season using the mean monthly coefficients outperforms in comparison to the standard methods, but their predictive strength was not as good as p.w.a. coefficients especially during

cold season. Similar findings were observed when different time intervals for calculating seasonal averages of the coefficients were used (e.g. 3-months averages or 6-months averages). The basic observed problem with monthly/seasonal coefficients associated to the global scale application of this study was that many parts of the world presented unreasonably high or low monthly/seasonal values of the coefficients (at least one order of magnitude larger or smaller from the standard values) during cold seasons. This problem occurred because P-T and H-S evapotranspiration models do not include the effect of humidity and wind, which becomes greater when temperature is low (in very low temperatures even the ASCE results can be questioned). Such values may lead to significant errors in monthly/seasonal $ET_o$ estimations during cold periods when there are deviations of climatic conditions (seasonal shifts/disturbances or climate changes in general) from those used for calibrating the coefficients. These were the reasons for using the threshold of 45 mm month$^{-1}$ to exclude such values from p.w.a. of the coefficients. Thus, the pw.a. annual values were chosen as the best solution for a global application because they counterbalance the errors that could be introduced by intra-annual/intra-seasonal climatic variability or other errors such as those described in the previous section of the Discussion (errors associated to the data).

It is also important to note that the derivation of annual coefficients is a pure optimization problem when stations data are used. For example, Cristea et al. (2013) derived coefficients of the P-T method for 106 stations that represent a range of climates across the contiguous USA. The coefficients were estimated for each station by minimizing the sum of the squared residuals between the benchmark FAO-56 and P-T using data only for the period April-September. The obtained optimized values of the coefficients were interpolated in order to make a map of the $a_{pt}$ coefficient. In this study, the maps of the coefficients were produced based on raster data and not stations data, which means that optimization should be performed pixel by pixel (~62000 pixels globally for the 0.5 degree resolution excluding Antarctica). This procedure would require special programming since readily available tool to perform this procedure does not exist in commercial or free GIS software packages. This is the main reason for using as an alternative method the Eqs.7 in GIS environment, since it can be calculated easily in raster calculators incorporated in the GIS packages. A solution could be the development of a tool for GIS purposes using rasters data that could be able to run using 24 rasters; 12 for the benchmark $ET_o$ and another 12 for the P-T or H-S $ET_o$ formula without the 1.26 and 0.0023 factors, respectively, in order to provide optimized annual values of their coefficients (for a global application filters to remove unreasonable values are also required).

*Observations derived by the application of H-S radiation formula*

Special attention was also given in the case of $K_{RS}$ coefficient for estimating $R_s$. Although there were indications that the spatial variation of p.w.a. $a_{pt}$ and $c_{hs2}$ coefficients at global scale may be linked to general climatic characteristics (Fig.5), the respective variation of p.w.a. $K_{RS}$ coefficient could not clearly be linked with a specific climatic or topographic characteristic. The only observed dependence, which showed some relevance to the spatial variation of $K_{RS}$, was a relatively negative correlation with the distance from the coastline $Dc$. This observed dependence can be only used as a general observation and not as a basis for applying in general the empirical rule of Allen et al. (1998) ($K_{RS}$=0.16 for "interior" and $K_{RS}$ =0.19 for "coastal" locations). The large uncertaintiy in the aforementioned rule was also indicated by Samani (2000)

and it is verified by the analysis presented in Fig.S4a of the supplementary material. Fig.S4a shows a relatively negative correlation between $K_{RS}$ and $Dc$ (for $Dc<500$ km) but also shows an extremely high variability of $K_{RS}$ close to the coastlines where $K_{RS}$ values are not necessarily higher in comparison to the values observed in the interior regions. The observed lower variability of $K_{RS}$ at interior regions is probably related to the fact that coastlines are more affected by oceanic-climatic

5   phenomena, which anyway present high spatial variability at global scale. The raster data of $K_{RS}$ (Fig.7) can be used as indicator to control the validity of the rule but also to control the validity of the given values 0.16-0.19 for a specific region. Samani (2000) also observed that the monthly $K_{RS}$ values may be influenced by the difference between monthly maximum and minimum temperature $TD$. This effect was also investigated through correlation between the mean monthly $K_{RS}$ coefficients and the mean monthly $TD$ values of the stations data (Fig.S4b, in supplementary material). The results showed

10   that the hypothesis related to the effect of $TD$ on $K_{RS}$ may be stronger locally in comparison to the effect of $Dc$, but again the variation of $K_{RS}$ is extremely large in the $TD$ range between 8-15 $^{\circ}$C (Fig.S4b), not allowing secure conclusions for a global scale application. The result of Fig.S4b is based only on the stations of Table 1, and for this reason the variation in a global scale is expected much larger.

15   *Recommendations for reducing the uncertainties when the re-adjusted coefficients of P-T and H-S models are used*

The uncertainties, which may be introduced by climate disturbances/changes or other uncertainties related to the data used for calibrating the coefficients, can be reduced taking into account some of the following observations and recommendations.

A separate analysis using only the stations of California showed that a regional mean value of the coefficients derived

20   by p.w.a. values may present even better performance because it probably counterbalances other uncertainties associated to the spatial climatic variability within a specific region. A factor for such uncertainties may be rainfall, which may not show significant seasonal deviations or deviations from the expected annual values for a large region but may show different spatial patterns every year within the region affecting the accuracy of the coefficients. The aforementioned observation was verified by the application of H-S method for $ET_o$ of short reference crop for the stations of California when the average

25   value of $c_{hs2}$=0.0024 obtained from the respective p.w.a.s values of the stations (Table S.2) was used (this value also approximates the standard value of 0.0023). The average value of sixty p.w.a.s. coefficients of the CA-USA stations gave better results from the individual coefficients (Fig.S5 and Table S4, in supplementary material). The aforementioned observations suggest that a robust territorial segmentation based on general topographic characteristics (e.g. elevation, slope, latitude and longitude, distance from the coastline etc) and general climatic characteristics (e.g. Köppen class, general

30   precipitation and temperature patterns) can provide a proper zonation of large territories for deriving very robust mean values of $a_{pt}$, $c_{hs2}$ and $K_{RS}$ coefficients using the respective p.w.a. values of each zone. Robust zonations based on grids of mean monthly precipitation and temperature using the data of Hijmans et al. (2005), or the mean monthly $ET_o$ rasters provided by this study can easily be performed using cluster analysis in GIS environment (Demertzi et al., 2014; Aschonitis et al., 2016a,b).

The comparison between P-T and H-S evapotranspiration methods with re-adjusted coefficients but also their comparison with the other models of Table 2 also provided significant information. From the comparison between P-T and H-S with re-adjusted coefficients, it was observed that H-S provided better results in both short and tall reference crop. The prevalence of H-S can be attributed to the fact that more than ~80% of stations from Table 1 are located in territories with

5    negative *DMAD* values (Fig.5a) giving a general advantage to H-S method for more robust estimations. This observation can justify the better performance of the standard H-S (with $c_{hs2}$=0.0023) in comparison to the standard P-T (with $a_{pt}$=1.26) for short reference crop (Table 5a) and indirectly validates the *DMAD* map. Considering these observations, it is recommended to take into account both the *MAD* (Fig.4,b,c) and *DMAD* (Fig.5a) maps before selecting one of the two methods either using the standard or the p.w.a. coefficients. From the comparisons with the other models of Table 2, it was observed that three

10   models, which use temperature, radiation and humidity data (i.e. VAL3, VAL2, Copais, and especially VAL3), provided better estimations. These models have shown very good performance using data from other case studies (Pan et al., 2011; Shiri et al., 2014; Kisi, 2014; Gao et al., 2015; Valipour, 2015a,2015c; Djaman et al., 2015, 2016, 2017; Ahooghalandari et al., 2017), and their use is recommended instead of the P-T and H-S with re-adjusted coefficients, when the only missing climatic parameter is wind speed.

15   A very interesting observation was also made about the tall reference crop based on the results of *MAD%* map (Fig.4a). In the *MAD%* class of ±10% of Fig.4a were observed some small negative values, which correspond to the ~2% of map coverage. These values indicate slightly larger annual values of ASCE-short in comparison to ASCE-tall. This result was observed in regions of extremely small vapour pressure deficit (areas of very high elevation, either of very cold, or extremely humid conditions scattered around the world) and it is a peculiarity of Eq.1 and probably an artefact. This result

20   occurred because the second term of the nominator in Eq.1 (which includes the vapour pressure deficit term and the $C_n$ coefficient) approximates to 0 when $e_s$-$e_a$ becomes extremely small, while the denominator of Eq.1 is always larger in ASCE-tall in comparison to ASCE-short due to the difference in $C_d$ value (0.34 for short and 0.38 for tall reference crop). A recommendation for partly solving this problem for tall reference crop applications is to use the revised coefficients of P-T and H-S methods derived for short reference crop in the places were the annual value of ASCE-tall is lower from ASCE-

25   short. This recommendation is based on the fact that annual ASCE-tall is expected to be always larger from the respective value of ASCE-short. This peculiarity was not corrected in the ASCE-tall maps and the respective $a_{pt}$ and $c_{hs2}$ coefficients for tall reference crop in order to show the absolute estimations of the ASCE-tall and the respective coefficients. Taking into account the *MAD* map (Fig.4a), the users can found the location of these pixels.

**5. Data availability**

30   The produced datasets of this study have been archived in PANGAEA database (https://doi.pangaea.de/10.1594/PANGAEA.868808) and in ESRN-database, which is currently supported by the University of Ferrara (Italy), Aristotle university of Thessaloniki (Greece) and University of Campania "Luigi Vanvitelli" (Italy) (http://www.esrn-database.org/gis-data.html or http://esrn-database.weebly.com/gis-data.html). Apart from the 0.5 degree

resolution raster datasets, the database contains the same datasets at finer resolution (30 arc-sec, 2.5 arc-min, 5 arc-min and 10 arc-min). These finer datasets are provided in order to cover the observed resolution range in the initial climatic data (e.g. the temperature data of Hijmans et al. (2005) are provided at 30 arc-sec resolution). The finer resolutions were produced using bilinear interpolation on solar radiation, humidity and wind speed data of Sheffield et al. (2006). This interpolation

5    method is not the most appropriate for such purposes. The data of finer resolutions can only be used as a tool to assess uncertainties associated to temperature variation effects within a 0.5 degree pixel or to estimate average values of the coefficients for larger territories in order to capture a better representation of the coastlines or islands that do not exist in 0.5 degree resolution (use of values from individual pixels is not recommended). A complete list of the datasets is provided in the Table S5.

**6. Conclusions**

The study provided  global grids of revised annual coefficients for the Priestley-Taylor (P-T) and Hargreaves-Samani (H-S) methods for estimating $ET_o$ for both short and tall reference crop. The coefficients were calibrated using respective grids of $ET_o$ estimated with the ASCE-standardized method. Respective grids of annual coefficients were also derived for the

15    radiation formula of H-S. The calibration procedures were based on global gridded climatic data of the period 1950-2000. The method for deriving annual coefficients of P-T and H-S methods was based on partial weighted averages (p.w.a.) of the respective mean monthly coefficients. This method estimates the annual values considering the amplitude of the parameter under investigation ($ET_o$ and $R_s$) giving more weight to the monthly coefficients of the months with higher $ET_o$ values (or $R_s$ values for the case of H-S radiation formula). The method also eliminates the effect of unreasonable monthly coefficients

20    that may occur during periods when $ET_o$ and $R_s$ fall below a specific threshold. The new coefficients were validated based on data from 140 stations located at various climatic zones of USA and Australia with expanded observations up to 2016. Additional tests were also performed for the case of short reference crop evapotranspiration using additional models with low requirements for climatic data. The validation procedure for $ET_o$ estimations of short reference crop showed that the P-T and H-S methods with revised coefficients outperformed the standard methods reducing the estimated $RMSE$ in $ET_o$ values

25    by 40% and 25%, respectively. The estimations of $R_s$ using the H-S formula with revised coefficients reduced the $RMSE$ by 28% in comparison to the standard formula. The comparisons with other models of short reference crop, showed that the P-T and H-S methods with revised coefficients can compete models of additional climatic parameters. In the case where only wind speed is missing from available data, the use of VAL2, VAL3 and Copais methods (temperature, radiation and humidity data requirements) is recommended. Finally, a raster database of 0.5 degree resolution was built consisting of: (a)

30    global maps for the mean monthly $ET_o$ values estimated by ASCE-standardized method for both reference crops, (b) global maps for the revised annual coefficients of the P-T and H-S evapotranspiration methods for both reference crops and a global map for the revised annual coefficients of the H-S radiation formula, (c) global maps that indicate the optimum locations for using the standard P-T and H-S methods and their possible annual errors based on reference values (*MAD%* and *DMAD*

maps). The online free availability of the database can support estimations of $ET_o$ and solar radiation for locations where climatic data are limited while it can support studies, which require such estimations at larger scales (e.g. country, continent, world).

The methods used in this study, their respective results and the observed uncertainties can be used as a base for future works focusing on: (a) the validation of the coefficients for other parts of the world, especially using climatic data obtained after 2000, and the comparison with other models of low data requirements (b) the recalibration of the coefficients using data from climatic models that include observations from more recent years and analysis of climate change effects on the coefficients, (c) the use of the available climatic datasets obtained from climatic models in order to calibrate models of the coefficients for various locations and not fixed values such as the ones given in this study, (d) analysis of alternative methods for deriving annual coefficients that approximate optimized values or incorporation of optimization algorithms in GIS environment for capturing the optimum solution per pixel, (e) the confrontation of uncertainties related to the data used for calibration and validation (e.g. low representativity of interpolated climatic parameters due to the lack of data in many parts of the world, errors associated to commonly used equations; such as the one used for adjusting wind data at 2 m height; uncertainties associated to the observed data etc).

**Supplementary material.** Supplementary information related to the article is given in the following supplementary file (to be added by the journal).

**Acknowledgements.** This study was performed in the context of two Post-Doctoral research studies by Dr.Vassilis Aschonitis financed by Ferrara University (Italy) and Aristotle University of Thessaloniki (Greece).

| Reference | Abbreviation | Formula | Climate data requirements* |
|---|---|---|---|
| Droogers and Allen (2002) | DRAL1 (Eq.8) | $ET_o = 0.00102 R_a \left(T_{mean} + 16.8\right) \cdot \left(TD\right)^{0.5}$ | $T_{max}, T_{min}$ |
| Droogers and Allen (2002) | DRAL2 (Eq.9) | $ET_o = 0.0005304 R_a \left(T_{mean} + 17.0\right) \cdot \left(TD - 0.0123P\right)^{0.76}$ | $T_{max}, T_{min}, P$ |
| Alexandris et al. (2006) | Copais (Eq.10) | $ET_o = 0.057 + 0.227C_2 + 0.643C_1 + 0.0124C_1C_2$ $C_1 = 0.6416 - 0.00784RH + 0.372R_s - 0.00264R_sRH$ $C_2 = -0.0033 + 0.00812T_{mean} + 0.101R_s + 0.00584R_sT_{mean}$ | $T_{mean}, R_s, RH$ |
| Valiantzas (2013a, 2014) | VAL1 (Eq.11) | $ET_o = 0.0393R_s\sqrt{T_{mean} + 9.5} - 0.19R_s^{0.6}\varphi^{0.15}$ $+ 0.0061\left(T_{mean} + 20\right)\left(1.12T_{mean} - T_{min} - 2\right)^{0.7}$ | $T_{mean}, T_{min}, R_s$ |
| Valiantzas (2013a; 2014) | VAL2 (Eq.12) | $ET_o = 0.0393R_s\sqrt{T_{mean} + 9.5} - 0.19R_s^{0.6}\varphi^{0.15}$ $+ 0.078\left(T_{mean} + 20\right)\left(1 - \dfrac{RH}{100}\right)$ | $T_{mean}, R_s, RH$ |
| Valiantzas (2013b) | VAL3 (Eq.13) | $ET_o = 0.0393R_s\sqrt{T_{mean} + 9.5} - 2.4\left(\dfrac{R_s}{R_a}\right)^2$ $+ Cu\left(T_{mean} + 20\right)\left(1 - \dfrac{RH}{100}\right)$ | $T_{mean}, R_s, RH$ ($Cu$=0.054 for $RH$>65% and $Cu$=0.083 for $RH \leq 65\%$) |
| Ahooghalandari et al. (2016) | AKJ1 (Eq.14) | $ET_o = 0.252 \cdot 0.408R_a + 0.221T_{mean}\left(1 - \dfrac{RH}{100}\right)$ | $T_{mean}, RH$ |
| Ahooghalandari et al. (2016) | AKJ2 (Eq.15) | $ET_o = 0.29 \cdot 0.408R_a + 0.15T_{max}\left(1 - \dfrac{RH}{100}\right)$ | $T_{max}, RH$ |

\* $T_{mean, max, min}$: Mean, maximum and minimum temperature ($^{o}$C), $TD$: difference between maximum and minimum temperature ($^{o}$C), $R_s$: incident solar radiation (MJ m$^{-2}$ d$^{-1}$), $R_a$: extraterrestrial solar radiation (MJ m$^{-2}$ d$^{-1}$), $RH$: relative humidity (%), $\varphi$: absolute value of latitude (rads), $P$: precipitation (mm month$^{-1}$)

**Table 3.** The % coverage* of *MAD%* classes based on mean annual values (according to Figs. 4), $R^2$ and *RMSD* based on comparisons of the mean monthly values of $ET_o$ and $R_s$ methods (comparisons based on 0.5 degree resolution maps).

| *MAD%* range | †$ET_o$ (ASCE-tall) for $C_n$=1600, $C_d$=0.38 (Eq.1) | †$ET_o$ (P-T) for $a_{pt}$=1.26 (Eq.2) | †$ET_o$ (H-S) for $c_{hs2}$ = 0.0023 (Eq.4b) | ‡$R_s$ (H-S) for $K_{RS}$=0.17 (Eq.3) |
|---|---|---|---|---|
| ≤ -50% | 0.0%* | 0.8% | 0.0% | 1.0% |
| -50 up to -25% | 0.0% | 14.8% | 5.2% | 2.2% |
| -25 up to -10% | 0.0% | 10.8% | 15.4% | 7.1% |
| -10 up to 10% | 25.2% | 21.3% | 24.8% | 55.3% |
| 10 up to 25% | 40.9% | 22.5% | 19.6% | 32.8% |
| 25 up to 50% | 33.6% | 21.9% | 29.2% | 1.6% |
| > 50% | 0.3% | 7.9% | 5.8% | 0.0% |
| $R^2$ | 0.98 | 0.77 | 0.89 | 0.92 |
| *RMSD* | 39.6§ | 36.0§ | 24.5§ | 2.4# |

*The % coverage was estimated after conversion from WGS84 ellipsoid to projected Cylindrical Equal Area coordinate system without considering Antarctica.

† *MAD%* of the three $ET_o$ methods is estimated versus ASCE-short.

‡ *MAD%* of the standard solar radiation method of H-S is estimated versus the $R_s$ data of Sheffield et al. (2006).

§ The unit of *RMSD* for $ET_o$ is mm month$^{-1}$.

**The unit of *RMSD* for $R_s$ is MJ m$^{-2}$ d$^{-1}$.**

**Table 4.** Spatial extent of the major climatic groups CGs from Köppen-Geiger climate map (Peel et al., 2007), % prevalence of P-T versus H-S within each CG based on the *DMAD* values.

| Climatic group (CG) of Köppen-Geiger | % extent of CGs* based on Peel et al. (2007) map | P-T versus H-S prevalence % inside a CG# | | |
|---|---|---|---|---|
| | | H-S (DMAD≤-1) | Trans. Zone -1<DMAD<1† | P-T (DMAD≥1) |
| A - tropical/megathermal | 20.66% | 32.0% | 3.6% | 64.4% |
| B - arid/semi-arid | 32.90% | 86.4% | 1.3% | 12.3% |
| C - temperate/mesothermal | 14.58% | 32.8% | 3.2% | 64.1% |
| D - continental/microthermal | 27.00% | 26.9% | 2.1% | 71.0% |
| E - polar/alpine (without Antarctica) | 4.86% | 71.1% | 16.3%‡ | 12.5% |

*The % coverage was estimated after conversion from WGS84 ellipsoid to projected Cylindrical Equal Area coordinate system without considering Antarctica.

**% coverage of *DMAD* values were estimated after pixel resampling using the resolution of Köppen map.**

†*DMAD* range were both methods present similar proximity to ASCE-short method (transitional zone).

‡Big part of this percentage corresponds to regions with annual $ET_o$ equal to 0 (e.g. inner Greenland). Such cases are included in the trans. zone of Fig.5a.

**Table 5.** Statistical criteria from the comparisons **(a)** between $ET_o$ values from ASCE-short and the methods used for estimating short reference crop evapotranspiration (i.e. P-T with standard and re-adjusted coefficients, H-S with standard and re-adjusted coefficients and all equations given in Table 2), **(b)** between $ET_o$ values from ASCE-tall and P-T, H-S methods with re-adjusted coefficients for tall reference crop, **(c)** $R_s$ values from stations and $R_s$ obtained from the H-S radiation formula with standard and re-adjusted coefficients.

| Case | Criterion | MAE | RMSE | NRMSE% | PBIAS% | $R^2$ | $bR^2$ | NSE | d | KGE |
|---|---|---|---|---|---|---|---|---|---|---|
| | Optimum value | 0 | 0 | 0 | 0 | 1 | 1 | 1 | 1 | 1 |
| a | P-T (Eq.2) with $a_{pt}$=1.26 | 36.92 | 48.87 | 90.9 | 33.3 | 0.763 | 0.591 | 0.173 | 0.849 | 0.539 |
| | P-T (Eq.2) with $a_{pt}$=p.w.a.s. | 22.71 | 29.43 | 40.3 | 7.5 | 0.856 | 0.832 | 0.837 | 0.956 | 0.883 |
| | H-S (Eq.4b) with $c_{hs2}$=0.0023 | 21.19 | 30.36 | 53.2 | 10.8 | 0.858 | 0.772 | 0.717 | 0.941 | 0.746 |
| | H-S (Eq.4b) with $c_{hs2}$=p.w.a.s. | 17.13 | 22.72 | 34.4 | 2.5 | 0.895 | 0.878 | 0.881 | 0.971 | 0.921 |
| | DRAL1 (Eq.8) | 19.53 | 27.05 | 44.5 | 4.8 | 0.859 | 0.818 | 0.802 | 0.955 | 0.833 |
| | DRAL2 (Eq.9) | 22.92 | 30.28 | 45.0 | 3.2 | 0.818 | 0.808 | 0.798 | 0.949 | 0.894 |
| | Copais (Eq.10) | 14.49 | 20.70 | 34.3 | 7.3 | 0.940 | 0.870 | 0.882 | 0.974 | 0.829 |
| | VAL1 (Eq.11) | 21.36 | 31.87 | 59.8 | 15.1 | 0.888 | 0.763 | 0.642 | 0.932 | 0.657 |
| | VAL2 (Eq.12) | 12.13 | 17.96 | 29.3 | 4.2 | 0.948 | 0.900 | 0.914 | 0.981 | 0.859 |
| | VAL3 (Eq.13) | 11.45 | 15.94 | 24.1 | 1.4 | 0.949 | 0.934 | 0.942 | 0.986 | 0.940 |
| | AKJ1 (Eq.14) | 21.17 | 24.24 | 42.0 | -10.6 | 0.955 | 0.887 | 0.824 | 0.964 | 0.771 |
| | AKJ2 (Eq.15) | 30.36 | 33.69 | 59.5 | -16.3 | 0.938 | 0.820 | 0.645 | 0.931 | 0.718 |
| b | P-T (Eq.2) with $a_{pt}$=p.w.a.t. | 40.43 | 52.38 | 50.6 | 8.4 | 0.770 | 0.754 | 0.743 | 0.930 | 0.845 |
| | H-S (Eq.4b) with $c_{hs2}$=p.w.a.t. | 31.87 | 42.34 | 45.2 | 3.7 | 0.823 | 0.806 | 0.795 | 0.950 | 0.885 |
| c | H-S $R_s$ (Eq.3) with $K_{RS}$=0.17 | 1.64 | 1.99 | 29.6 | -4.5 | 0.930 | 0.885 | 0.912 | 0.977 | 0.932 |
| | H-S $R_s$ (Eq.3) with $K_{RS}$=p.w.a. | 1.05 | 1.43 | 22.3 | -0.8 | 0.952 | 0.944 | 0.950 | 0.988 | 0.972 |

**FIGURES**

[Figure]

**Figure 1.** Position of stations **(a)** from California-USA obtained by CIMIS database and **(b)** from Australia obtained by
5    AGBM database (the numbers indicate the No. of stations from Table 1 without the abbreviations CA- and A-).

[Figure]

**Figure 2.** Mean annual values (mm year$^{-1}$) of $ET_o$ for the period 1950-2000 using **(a)** the ASCE-short method, **(b)** the ASCE-tall method, **(c)** the standard P-T method for $a_{pt}$=1.26 and **(d)** the standard H-S method for $c_{hs2}$=0.0023 (0.5 degree resolution maps, mean±st.dev. are estimated after conversion from WGS84 to Cylindrical Equal Area coordinate system).

[Figure]

**Figure 3.** Mean annual values of $R_s$ (MJ m$^{-2}$ d$^{-1}$) for the period 1950-2000 **(a)** from the database of Sheffield et al. (2006) and **(b)** estimated using the standard H-S radiation formula for $K_{RS}$=0.17 (Eq.3) (0.5 degree resolution maps, mean±st.dev. are estimated after conversion from WGS84 to Cylindrical Equal Area coordinate system).

[Figure]

**Figure 4.** Mean annual difference % (*MAD*%) of *ET$_o$* between the ASCE-short and **(a)** the ASCE-tall method, **(b)** the standard P-T method for *a$_{pt}$*=1.26, **(c)** the standard H-S method for *c$_{hs2}$* = 0.0023, and **(d)** *MAD*% between *R$_s$* values of Sheffield et al. (2006) and the standard solar radiation formula of H-S for *K$_{RS}$*=0.17 (0.5 degree resolution maps, mean±st.dev. are estimated after conversion from WGS84 to Cylindrical Equal Area coordinate system).

[Figure]

**Figure 5**. **(a)** P-T versus H-S prevalence according to their proximity to ASCE-short method expressed by the *DMAD* values (0.5 degree resolution map) and **(b)** Spatial extent of the major climatic groups of the Köppen-Geiger climate classification according to Peel et al. (2007).

[Figure]

**Figure 6.** Partial weighted averages of mean monthly **(a)** $a_{pt}$ for short reference crop, **(b)** $a_{pt}$ for tall reference crop, **(c)** $c_{hs2}$ for short reference crop and **(d)** $c_{hs2}$ for tall reference crop (0.5 degree resolution maps, mean±st.dev. are estimated after conversion from WGS84 to Cylindrical Equal Area coordinate system excluding pixels of 0 value).

[Figure]

5 **Figure 7.** Partial weighted averages of mean monthly $K_{RS}$ (0.5 degree resolution maps, mean±st.dev. are estimated after conversion from WGS84 to Cylindrical Equal Area coordinate system).

[Figure]

**Figure 8.** Comparative 1:1 plots between the results of $ET_o$ ASCE-short (mm month[-1]) versus **(a)** the standard P-T method with $a_{pt}$=1.26, **(b)** the P-T method with $a_{pt}$=p.w.a.s. (0.5 degree resolution), **(c)** the standard H-S method with $c_{hs2}$=0.0023. **(d)** the H-S method with $c_{hs2}$=p.w.a.s. (0.5 degree resolution), **(e)** DRAL1 (Eq.8)**, (f)** DRAL2 (Eq.9), **(g)** Copais (Eq.10), **(h)** VAL1 (Eq.11), **(i)** VAL2 (Eq.12), **(j)** VAL3 (Eq.13), **(k)** AKJ1 (Eq.14), **(l)** AKJ2 (Eq.15).

[Figure]

5 **Figure 9.** Comparative 1:1 plots between the results of $ET_o$ ASCE-tall (mm month$^{-1}$) versus **(a)** the P-T method with $a_{pt}$=p.w.a.t. (0.5 degree resolution), **(b)** the H-S method with $c_{hs2}$=p.w.a.t. (0.5 degree resolution).

[Figure]

**Figure 10.** Comparative 1:1 plots between the $R_s$ (MJ m$^{-2}$ d$^{-1}$) values of CA-USA and Australia stations versus the results of
10 H-S radiation formula (Eq.3) **(a)** with $K_{RS}$= 0.17, **(b)** with $K_{RS}$=p.w.a. (0.5 degree resolution).

**Supplementary Material**

*Indirect verification of the data cleaning that was performed in the derived data from CIMIS database.*

[Figure]

**Fig.S1** Comparison between the mean monthly $ET_o$ values of ASCE-short method using the final clean climatic data from CIMIS database versus the provided mean monthly values of $ET_o$ by the database using the CIMIS evapotranspiration method.

*General statistics of meteorological stations data (validation data) and comparison with the raster data (calibration data) used for developing the global maps of $ET_o$ with ASCE method.*

**Table S1**. General statistics* of the mean monthly observed values of climatic parameters from the 140 stations of California-USA and Australia that participate in the estimation of reference evapotranspiration with the ASCE method.

| Parameter | $T_{max}$ | $T_{min}$ | $R_s$ | RH | $u_2$ | P | $ET_o$ ASCE-short | $ET_o$ ASCE-tall |
|---|---|---|---|---|---|---|---|---|
| Unit | °C | °C | MJ m$^{-2}$ d$^{-1}$ | % | m s$^{-1}$ | mm month$^{-1}$ | mm month$^{-1}$ | mm month$^{-1}$ |
| Average | 25.3 | 11.4 | 18.8 | 56.4 | 2.6 | 41.5 | 138.4 | 190.5 |
| Minimum | 5.3 | -7.2 | 4.9 | 19.0 | 0.9 | 0.0 | 17.9 | 26.2 |
| Lower quartile | 19.7 | 6.5 | 13.5 | 45.5 | 1.8 | 11.7 | 82.2 | 112.7 |
| Upper quartile | 31.1 | 15.8 | 24.4 | 68.2 | 3.2 | 50.6 | 186.9 | 254.2 |
| Maximum | 41.2 | 26.3 | 30.1 | 90.3 | 6.8 | 470.4 | 377.5 | 563.8 |
| Range | 35.9 | 33.5 | 25.2 | 71.3 | 5.9 | 470.4 | 359.6 | 537.6 |
| Standard deviation | 7.1 | 6.4 | 6.5 | 15.4 | 1.0 | 51.5 | 69.5 | 98.9 |
| Coeff. of variation % | 28.11% | 56.13% | 34.32% | 27.36% | 37.05% | 123.90% | 50.17% | 51.93% |

*The statistics are based on 1680 values (140 stations × 12 months)

[Figure]

**Fig.S2** Comparison of $T_{max}$, $T_{min}$, $R_s$, $R_n$, $DE$ (vapour pressure deficit), $u_2$, $ET_o$ ASCE-short, and $ET_o$ ASCE-tall between the rasters (0.5 degree resolution) and the stations data.

[Figure]

**Fig.S3** Comparison of total averages of mean monthly $u_2$ values through Box-Whisker plots: a) between rasters (Sheffield et al., 2006) and California-USA stations, b) between rasters (Sheffield et al., 2006) and Australia stations.

*Extracted values of the p.w.a. coefficients for each station in the validation dataset.*

**Table S2.** Partial weighted averages of mean monthly coefficients ($a_{pt}$, $c_{hs2}$, $K_{RS}$) for each station extracted by the 0.5 degree resolution maps.

| No. | Code | Station | Country | $a_{pt}$ p.w.a.s. (0.5 deg) | $a_{pt}$ p.w.a.t. (0.5 deg) | $c_{hs2}$ p.w.a.s. (0.5 deg) | $c_{hs2}$ p.w.a.t. (0.5 deg) | $K_{RS}$ p.w.a. (0.5 deg) |
|---|---|---|---|---|---|---|---|---|
| CA-1 | 006 | Davis | USA-CA | 1.45 | 1.93 | 0.0022 | 0.0029 | 0.16 |
| CA-2 | 002 | FivePoints | USA-CA | 1.53 | 2.06 | 0.0023 | 0.0030 | 0.16 |
| CA-3 | 005 | Shafter | USA-CA | 1.48 | 1.97 | 0.0023 | 0.0031 | 0.16 |
| CA-4 | 007 | Firebaugh/Telles | USA-CA | 1.48 | 1.99 | 0.0022 | 0.0029 | 0.15 |
| CA-5 | 012 | Durham | USA-CA | 1.49 | 2.01 | 0.0024 | 0.0031 | 0.16 |
| CA-6 | 008 | Gerber | USA-CA | 1.46 | 1.96 | 0.0023 | 0.0031 | 0.16 |
| CA-7 | 015 | Stratford | USA-CA | 1.47 | 1.95 | 0.0023 | 0.0030 | 0.16 |
| CA-8 | 019 | Castroville | USA-CA | 1.20 | 1.53 | 0.0023 | 0.0029 | 0.18 |
| CA-9 | 021 | Kettleman | USA-CA | 1.49 | 1.99 | 0.0022 | 0.0030 | 0.15 |
| CA-10 | 027 | Zamora | USA-CA | 1.45 | 1.93 | 0.0022 | 0.0029 | 0.16 |
| CA-11 | 030 | Nicolaus | USA-CA | 1.45 | 1.93 | 0.0022 | 0.0029 | 0.16 |
| CA-12 | 032 | Colusa | USA-CA | 1.49 | 2.01 | 0.0023 | 0.0030 | 0.15 |
| CA-13 | 033 | Visalia | USA-CA | 1.48 | 1.96 | 0.0023 | 0.0031 | 0.16 |
| CA-14 | 035 | Bishop | USA-CA | 1.71 | 2.38 | 0.0026 | 0.0036 | 0.15 |
| CA-15 | 039 | Parlier | USA-CA | 1.45 | 1.92 | 0.0023 | 0.0030 | 0.16 |
| CA-16 | 041 | Calipatria/Mulberry | USA-CA | 1.79 | 2.50 | 0.0025 | 0.0036 | 0.15 |
| CA-17 | 043 | McArthur | USA-CA | 1.31 | 1.70 | 0.0022 | 0.0029 | 0.15 |
| CA-18 | 044 | U.C.Riverside | USA-CA | 1.68 | 2.35 | 0.0025 | 0.0035 | 0.16 |
| CA-19 | 047 | Brentwood | USA-CA | 1.45 | 1.94 | 0.0023 | 0.0030 | 0.16 |
| CA-20 | 049 | Oceanside | USA-CA | 1.62 | 2.26 | 0.0029 | 0.0040 | 0.18 |
| CA-21 | 054 | Blackwells Corner | USA-CA | 1.49 | 1.99 | 0.0022 | 0.0030 | 0.15 |
| CA-22 | 056 | Los Banos | USA-CA | 1.47 | 1.95 | 0.0023 | 0.0030 | 0.16 |
| CA-23 | 061 | Orland | USA-CA | 1.45 | 1.94 | 0.0023 | 0.0030 | 0.16 |
| CA-24 | 062 | Temecula | USA-CA | 1.62 | 2.26 | 0.0029 | 0.0040 | 0.18 |
| CA-25 | 064 | Santa Ynez | USA-CA | 1.36 | 1.81 | 0.0024 | 0.0032 | 0.17 |
| CA-26 | 068 | Seeley | USA-CA | 1.93 | 2.76 | 0.0026 | 0.0037 | 0.15 |
| CA-27 | 070 | Manteca | USA-CA | 1.43 | 1.89 | 0.0023 | 0.0030 | 0.16 |
| CA-28 | 071 | Modesto | USA-CA | 1.43 | 1.89 | 0.0023 | 0.0030 | 0.16 |

| | | | | | | | | |
|---|---|---|---|---|---|---|---|---|
| CA-29 | 077 | Oakville | USA-CA | 1.37 | 1.82 | 0.0023 | 0.0030 | 0.16 |
| CA-30 | 075 | Irvine | USA-CA | 1.65 | 2.29 | 0.0027 | 0.0038 | 0.17 |
| CA-31 | 078 | Pomona | USA-CA | 1.72 | 2.39 | 0.0027 | 0.0038 | 0.16 |
| CA-32 | 080 | Fresno State | USA-CA | 1.45 | 1.92 | 0.0023 | 0.0030 | 0.16 |
| CA-33 | 083 | Santa Rosa | USA-CA | 1.24 | 1.63 | 0.0021 | 0.0027 | 0.16 |
| CA-34 | 084 | Browns Valley | USA-CA | 1.45 | 1.93 | 0.0024 | 0.0031 | 0.17 |
| CA-35 | 085 | Hopland F.S. | USA-CA | 1.38 | 1.87 | 0.0021 | 0.0028 | 0.15 |
| CA-36 | 086 | Lindcove | USA-CA | 1.48 | 1.96 | 0.0023 | 0.0031 | 0.16 |
| CA-37 | 087 | Meloland | USA-CA | 1.91 | 2.71 | 0.0025 | 0.0036 | 0.14 |
| CA-38 | 088 | Cuyama | USA-CA | 1.37 | 1.81 | 0.0025 | 0.0033 | 0.17 |
| CA-39 | 091 | Tulelake F.S. | USA-CA | 1.39 | 1.81 | 0.0022 | 0.0029 | 0.15 |
| CA-40 | 092 | Kesterson | USA-CA | 1.47 | 1.95 | 0.0023 | 0.0030 | 0.16 |
| CA-41 | 094 | Goletta foothills | USA-CA | 1.37 | 1.81 | 0.0025 | 0.0033 | 0.17 |
| CA-42 | 099 | Santa Monica | USA-CA | 1.63 | 2.24 | 0.0027 | 0.0037 | 0.17 |
| CA-43 | 103 | Windsor | USA-CA | 1.28 | 1.68 | 0.0021 | 0.0028 | 0.16 |
| CA-44 | 104 | De Laveaga | USA-CA | 1.20 | 1.53 | 0.0023 | 0.0029 | 0.18 |
| CA-45 | 105 | Westlands | USA-CA | 1.48 | 1.97 | 0.0023 | 0.0030 | 0.16 |
| CA-46 | 106 | Sanel Valley | USA-CA | 1.10 | 1.39 | 0.0019 | 0.0024 | 0.16 |
| CA-47 | 57 | Buntingville | USA-CA | 1.55 | 2.11 | 0.0023 | 0.0031 | 0.15 |
| CA-48 | 90 | Alturas | USA-CA | 1.33 | 1.74 | 0.0023 | 0.0030 | 0.15 |
| CA-49 | 151 | Ripley | USA-CA | 2.01 | 2.88 | 0.0028 | 0.0040 | 0.16 |
| CA-50 | 183 | Owens Lake North | USA-CA | 1.43 | 1.89 | 0.0026 | 0.0034 | 0.17 |
| CA-51 | 147 | Otay Lake | USA-CA | 1.71 | 2.39 | 0.0026 | 0.0037 | 0.15 |
| CA-52 | 175 | Palo Verde II | USA-CA | 1.98 | 2.84 | 0.0027 | 0.0038 | 0.15 |
| CA-53 | 135 | Blynthe NE | USA-CA | 2.01 | 2.88 | 0.0028 | 0.0040 | 0.16 |
| CA-54 | 155 | Bryte | USA-CA | 1.45 | 1.93 | 0.0022 | 0.0029 | 0.16 |
| CA-55 | 159 | Monrovia | USA-CA | 1.72 | 2.39 | 0.0027 | 0.0038 | 0.16 |
| CA-56 | 161 | Patterson | USA-CA | 1.48 | 1.98 | 0.0023 | 0.0030 | 0.16 |
| CA-57 | 174 | Long Beach | USA-CA | 1.52 | 2.08 | 0.0029 | 0.0040 | 0.20 |
| CA-58 | 173 | Torrey Pines | USA-CA | 1.62 | 2.26 | 0.0029 | 0.0040 | 0.18 |
| CA-59 | 150 | Miramar | USA-CA | 1.62 | 2.26 | 0.0029 | 0.0040 | 0.18 |
| CA-60 | 153 | Escondido SPV | USA-CA | 1.62 | 2.24 | 0.0025 | 0.0035 | 0.16 |
| A-1 | 32040 | Townsville Aero | Australia | 1.28 | 1.66 | 0.0026 | 0.0033 | 0.19 |
| A-2 | 33307 | Woolshed | Australia | 1.28 | 1.66 | 0.0026 | 0.0033 | 0.19 |
| A-3 | 2056 | Kununurra Aero | Australia | 1.56 | 2.11 | 0.0025 | 0.0034 | 0.18 |
| A-4 | 35264 | Emerald | Australia | 1.29 | 1.63 | 0.0021 | 0.0027 | 0.16 |
| A-5 | 24024 | Loxton R.C. | Australia | 1.63 | 2.21 | 0.0024 | 0.0032 | 0.15 |
| A-6 | 74037 | Yanco AG.I. | Australia | 1.48 | 1.95 | 0.0023 | 0.0031 | 0.16 |
| A-7 | 74258 | Deniliquin Airp.AWS | Australia | 1.49 | 1.99 | 0.0023 | 0.0030 | 0.16 |
| A-8 | 75041 | Griffith Airp.AWS | Australia | 1.51 | 2.02 | 0.0024 | 0.0032 | 0.16 |
| A-9 | 76031 | Mildura Airp. | Australia | 1.67 | 2.30 | 0.0025 | 0.0034 | 0.16 |
| A-10 | 24048 | Renmark Apt.1 | Australia | 1.63 | 2.21 | 0.0024 | 0.0032 | 0.15 |
| A-11 | 40082 | University of QLD G. | Australia | 1.27 | 1.63 | 0.0021 | 0.0027 | 0.16 |
| A-12 | 40922 | Kingaroy Airp. | Australia | 1.23 | 1.56 | 0.0021 | 0.0026 | 0.16 |
| A-13 | 41359 | Oakey Aero | Australia | 1.23 | 1.55 | 0.0021 | 0.0026 | 0.16 |
| A-14 | 41522 | Dalby Airp. | Australia | 1.26 | 1.60 | 0.0021 | 0.0026 | 0.16 |
| A-15 | 41525 | Warwick | Australia | 1.22 | 1.55 | 0.0021 | 0.0027 | 0.16 |
| A-16 | 41529 | Toowoomba Airp. | Australia | 1.25 | 1.58 | 0.0021 | 0.0026 | 0.16 |
| A-17 | 80091 | Kyabram | Australia | 1.43 | 1.88 | 0.0022 | 0.0030 | 0.16 |
| A-18 | 81049 | Tatura I.S.A. | Australia | 1.43 | 1.88 | 0.0022 | 0.0030 | 0.16 |
| A-19 | 81124 | Yarrawonga | Australia | 1.39 | 1.80 | 0.0022 | 0.0028 | 0.15 |
| A-20 | 81125 | Shepparton Airp. | Australia | 1.43 | 1.88 | 0.0022 | 0.0030 | 0.16 |
| A-21 | 41175 | Applethorpe | Australia | 1.20 | 1.49 | 0.0021 | 0.0026 | 0.16 |
| A-22 | 81123 | Bendigo Airp. | Australia | 1.43 | 1.89 | 0.0023 | 0.0030 | 0.15 |
| A-23 | 85072 | East sale Airp. | Australia | 1.34 | 1.80 | 0.0023 | 0.0031 | 0.16 |
| A-24 | 85279 | Bairnsdale Airp. | Australia | 1.40 | 1.88 | 0.0024 | 0.0032 | 0.16 |
| A-25 | 85280 | Morwell L.V.Airp. | Australia | 1.38 | 1.86 | 0.0023 | 0.0031 | 0.15 |

| A-26 | 85296 | Mount Moornapa | Australia | 1.43 | 1.94 | 0.0023 | 0.0031 | 0.15 |
|------|-------|----------------|-----------|------|------|--------|--------|------|
| A-27 | 90035 | Colac | Australia | 1.46 | 2.00 | 0.0024 | 0.0033 | 0.16 |
| A-28 | 9538 | Dwellingup | Australia | 1.36 | 1.80 | 0.0023 | 0.0031 | 0.17 |
| A-29 | 9617 | Bridgetown | Australia | 1.32 | 1.73 | 0.0022 | 0.0029 | 0.16 |
| A-30 | 23373 | Nuriootpa Pirsa | Australia | 1.54 | 2.07 | 0.0024 | 0.0032 | 0.16 |
| A-31 | 26021 | Mount Gambier Aero | Australia | 1.38 | 1.85 | 0.0024 | 0.0032 | 0.16 |
| A-32 | 26091 | Coonawarra | Australia | 1.49 | 2.03 | 0.0023 | 0.0032 | 0.15 |
| A-33 | 66062 | Sydney (Obs.Hill) | Australia | 1.18 | 1.52 | 0.0022 | 0.0029 | 0.17 |
| A-34 | 33002 | Ayr DPI Res.St. | Australia | 1.22 | 1.54 | 0.0023 | 0.0029 | 0.18 |
| A-35 | 7176 | Newman Aero | Australia | 2.04 | 2.94 | 0.0031 | 0.0044 | 0.18 |
| A-36 | 13017 | Giles | Australia | 2.18 | 3.20 | 0.0032 | 0.0046 | 0.17 |
| A-37 | 11052 | Forrest | Australia | 1.78 | 2.52 | 0.0027 | 0.0038 | 0.15 |
| A-38 | 11003 | Eucla | Australia | 1.68 | 2.39 | 0.0029 | 0.0041 | 0.17 |
| A-39 | 12071 | Salmon Gums | Australia | 1.65 | 2.28 | 0.0027 | 0.0038 | 0.16 |
| A-40 | 7045 | Meekatharra Airp. | Australia | 1.98 | 2.84 | 0.0031 | 0.0044 | 0.18 |
| A-41 | 1025 | Doongan | Australia | 1.38 | 1.82 | 0.0027 | 0.0035 | 0.19 |
| A-42 | 2012 | Halls Creek Airp. | Australia | 1.72 | 2.39 | 0.0025 | 0.0034 | 0.17 |
| A-43 | 13015 | Carnegie | Australia | 2.12 | 3.09 | 0.0030 | 0.0044 | 0.17 |
| A-44 | 3080 | Curtin Aero | Australia | 1.59 | 2.17 | 0.0026 | 0.0036 | 0.18 |
| A-45 | 6022 | Gascoyne Junction | Australia | 1.97 | 2.83 | 0.0029 | 0.0041 | 0.17 |
| A-46 | 9789 | Esperance | Australia | 1.53 | 2.12 | 0.0027 | 0.0038 | 0.17 |
| A-47 | 91223 | Marrawah | Australia | 1.10 | 1.47 | 0.0023 | 0.0030 | 0.19 |
| A-48 | 18106 | Nullarbor | Australia | 1.77 | 2.52 | 0.0027 | 0.0039 | 0.16 |
| A-49 | 16090 | Coober Pedy Airp. | Australia | 2.05 | 2.98 | 0.0030 | 0.0044 | 0.17 |
| A-50 | 16085 | Marla Police St. | Australia | 2.05 | 2.98 | 0.0030 | 0.0044 | 0.17 |
| A-51 | 13011 | Warburton Airfield | Australia | 2.19 | 3.22 | 0.0031 | 0.0046 | 0.17 |
| A-52 | 15528 | Yuendumu | Australia | 2.14 | 3.13 | 0.0032 | 0.0046 | 0.17 |
| A-53 | 15666 | Rabbit Flat | Australia | 2.15 | 3.14 | 0.0029 | 0.0042 | 0.16 |
| A-54 | 14829 | Lajamanu Airp. | Australia | 1.85 | 2.63 | 0.0026 | 0.0036 | 0.17 |
| A-55 | 15135 | Tennant Creek Airp. | Australia | 2.05 | 2.98 | 0.0031 | 0.0045 | 0.18 |
| A-56 | 37010 | Camooweal Township | Australia | 1.93 | 2.78 | 0.0027 | 0.0038 | 0.16 |
| A-57 | 14707 | Wollogorang | Australia | 1.56 | 2.12 | 0.0028 | 0.0037 | 0.19 |
| A-58 | 14938 | Mango Farm | Australia | 1.37 | 1.79 | 0.0023 | 0.0030 | 0.17 |
| A-59 | 69134 | Batemans Bay | Australia | 1.19 | 1.51 | 0.0021 | 0.0027 | 0.16 |
| A-60 | 14198 | Jabiru Airp. | Australia | 1.28 | 1.60 | 0.0023 | 0.0028 | 0.18 |
| A-61 | 28008 | Lockhart River Airp. | Australia | 1.27 | 1.63 | 0.0026 | 0.0033 | 0.19 |
| A-62 | 34084 | Charters Towers Airp. | Australia | 1.27 | 1.60 | 0.0022 | 0.0028 | 0.17 |
| A-63 | 29038 | Kowanyama Airp. | Australia | 1.29 | 1.65 | 0.0024 | 0.0030 | 0.19 |
| A-64 | 32078 | Ingham Composite | Australia | 1.34 | 1.76 | 0.0025 | 0.0032 | 0.18 |
| A-65 | 40854 | Logan City W.T.P. | Australia | 1.33 | 1.79 | 0.0023 | 0.0031 | 0.17 |
| A-66 | 8095 | Mullewa | Australia | 1.78 | 2.51 | 0.0027 | 0.0038 | 0.16 |
| A-67 | 8251 | Kalbarri | Australia | 1.58 | 2.18 | 0.0028 | 0.0038 | 0.18 |
| A-68 | 8225 | Eneabba | Australia | 1.82 | 2.60 | 0.0029 | 0.0041 | 0.17 |
| A-69 | 7139 | Paynes Find | Australia | 1.81 | 2.54 | 0.0027 | 0.0038 | 0.17 |
| A-70 | 10007 | Bencubbin | Australia | 1.61 | 2.20 | 0.0025 | 0.0034 | 0.16 |
| A-71 | 10092 | Merredin | Australia | 1.62 | 2.21 | 0.0025 | 0.0035 | 0.16 |
| A-72 | 12038 | Kalgoorlie-Boulder Airp. | Australia | 1.79 | 2.52 | 0.0028 | 0.0040 | 0.17 |
| A-73 | 16098 | Tarcoola Aero | Australia | 1.95 | 2.80 | 0.0028 | 0.0041 | 0.16 |
| A-74 | 18195 | Minnipa Pirsa | Australia | 1.73 | 2.44 | 0.0027 | 0.0038 | 0.16 |
| A-75 | 46126 | Tibooburra Airp. | Australia | 2.02 | 2.92 | 0.0029 | 0.0042 | 0.17 |
| A-76 | 48245 | Boorke Airp. AWS | Australia | 1.68 | 2.30 | 0.0025 | 0.0034 | 0.16 |
| A-77 | 55325 | Tamworth Airp. AWS | Australia | 1.21 | 1.48 | 0.0020 | 0.0024 | 0.15 |
| A-78 | 38026 | Birdsville Airp. | Australia | 2.36 | 3.52 | 0.0032 | 0.0047 | 0.16 |
| A-79 | 30161 | Richmond Airp. | Australia | 1.64 | 2.25 | 0.0024 | 0.0033 | 0.16 |
| A-80 | 33013 | Collinsville Airp. | Australia | 1.38 | 1.81 | 0.0024 | 0.0031 | 0.17 |

**Table S3.** Ranking of models for each criterion (1 is the best, 12 is the worst).

| Model | MAE | RMSE | NRMSE% | PBIAS% | $R^2$ | $bR^2$ | NSE | d | KGE |
|---|---|---|---|---|---|---|---|---|---|
| P-T (Eq.2) with $a_{pt}$=1.26 | 12 | 12 | 12 | 12 | 12 | 12 | 12 | 12 | 12 |
| P-T (Eq.2) with $a_{pt}$=p.w.a.s. | 9 | 7 | 5 | 7 | 10 | 6 | 5 | 6 | 4 |
| H-S (Eq.4b) with $c_{hs2}$=0.0023 | 7 | 9 | 9 | 9 | 9 | 10 | 9 | 9 | 9 |
| H-S (Eq.4b) with $c_{hs2}$=p.w.a.s. | 4 | 4 | 4 | 2 | 6 | 4 | 4 | 4 | 2 |
| DRAL1 (Eq.8) | 5 | 6 | 7 | 5 | 8 | 8 | 7 | 7 | 6 |
| DRAL2 (Eq.9) | 10 | 8 | 8 | 3 | 11 | 9 | 8 | 8 | 3 |
| Copais (Eq.10) | 3 | 3 | 3 | 6 | 4 | 5 | 3 | 3 | 7 |
| VAL1 (Eq.11) | 8 | 10 | 11 | 10 | 7 | 11 | 11 | 10 | 11 |
| VAL2 (Eq.12) | 2 | 2 | 2 | 4 | 3 | 2 | 2 | 2 | 5 |
| VAL3 (Eq.13) | 1 | 1 | 1 | 1 | 2 | 1 | 1 | 1 | 1 |
| AKJ1 (Eq.14) | 6 | 5 | 6 | 8 | 1 | 3 | 6 | 5 | 8 |
| AKJ2 (Eq.15) | 11 | 11 | 10 | 11 | 5 | 7 | 10 | 11 | 10 |

*Analysis of Dc (distance from the coastline) and DT (difference between max and min monthly temperature) effects on $K_{RS}$ coefficient.*

[Figure]

**Fig.S4** Correlation between **(a)** p.w.a. $K_{RS}$ and *Dc* (59031 observations derived by 0.5 degree resolution maps, all regions included except Greenland that showed extremely high $K_{RS}$ values in inland areas, see Fig.7 in the manuscript) and (b) monthly $K_{RS}$ and monthly *TD* values (1680 mean monthly observations derived by the 140 stations of Table 1 in the manuscript).

*Example case using the Hargreaves-Samani method of evapotranspiration for the stations of California with revised coefficients.*

[Figure]

**Fig.S5** Comparative 1:1 plots between the results of ASCE-short versus **(a)** the H-S method with $c_{hs2}$=0.0024 (mean value of p.w.a.s. $c_{hs2}$ coefficients of all California stations obtained from Table S.2), **(b)** the H-S method using the individual values of $c_{hs2}$=p.w.a.s. for each station of California stations (Table S.2).

**Table S4.** Statistical criteria from the respective comparisons given in Fig.S5.

| | | H-S vs. ASCE-short | |
|---|---|---|---|
| Criterion | Optimum value | H-S (Eq.4b) with $c_{hs2}$=0.0024 | H-S (Eq.4b) with $c_{hs2}$=p.w.a.s. |
| MAE | 0 | 13.237* | 14.297 |
| RMSE | 0 | 16.693* | 19.119 |
| NRMSE% | 0 | 26.900* | 30.500 |
| PBIAS% | 0 | -7.100* | -7.200 |
| $R^2$ | 1 | 0.947* | 0.927 |
| $bR^2$ | 1 | 0.887* | 0.863 |
| NSE | 1 | 0.928* | 0.907 |
| d | 1 | 0.982* | 0.976 |
| KGE | 1 | 0.924* | 0.916 |

*The asterisk is used to indicate the best value of each criterion.

**Attributes of the datasets provided in the context of this study**

**Table S5.** Contents of the database produced in this study (all five resolutions are included: 30 arc-sec, 2.5 arc-min, 5 arc-min, 10 arc-min, 0.5 deg.). The order of contents follows the alphabetical order of file names as they are stored in PANGAEA (https://doi.pangaea.de/10.1594/PANGAEA.868808)

| No. | Content/resolution | File name | Method | Comment |
|---|---|---|---|---|
| 1 | Re-adjusted Priestley-Taylor coefficient for short ref.crop ETo (rescaled ×100) (unitless)/(30 arc-sec) | apts1_30s.zip | Re-calibration of Priestley-Taylor coefficient apt=1.26 for ETo method (Priestley and Taylor, 1972) using ASCE-EWRI method (Allen et al., 2005) for short ref.crop | the zip contains 1 raster (ESRI-grid) (partial weighted average of mean monthly values). For zero values use the closest non-zero value. |
| 2 | Re-adjusted Priestley-Taylor coefficient for tall ref.crop ETo (rescaled ×100) (unitless)/(30 arc-sec) | aptt1_30s.zip | Re-calibration of Priestley-Taylor coefficient apt=1.26 for ETo method (Priestley and Taylor, 1972) using ASCE-EWRI method (Allen et al., 2005) for tall ref.crop | the zip contains 1 raster (ESRI-grid) (partial weighted average of mean monthly values). For zero values use the closest non-zero value. |
| 3 | Re-adjusted Hargreaves-Samani coefficient for short ref.crop ETo (rescaled ×100,000) (unitless)/(30 arc-sec) | chs2s1_30s.zip | Re-calibration of Hargreaves-Samani coefficient chs2=0.0023 for ETo method (Hargreaves and Samani, 1982, 1985) using ASCE-EWRI method (Allen et al., 2005) for short ref.crop | the zip contains 1 raster (ESRI-grid) (partial weighted average of mean monthly values). For zero values use the closest non-zero value. |
| 4 | Re-adjusted Hargreaves-Samani coefficient for tall ref.crop ETo (rescaled ×100,000) (unitless)/(30 arc-sec) | chs2t1_30s.zip | Re-calibration of Hargreaves-Samani coefficient chs2=0.0023 for ETo method (Hargreaves and Samani, 1982, 1985) using ASCE-EWRI method (Allen et al., 2005) for tall ref.crop | the zip contains 1 raster (ESRI-grid) (partial weighted average of mean monthly values). For zero values use the closest non-zero value. |
| 5 | Hargeaves-Samani versus Priestley-Taylor (comparison between original methods versus ASCE-short) (DMADhp) (%)/(30 arc-sec) | dmadhp1_30s.zip | abs(madhs)-abs(madpt), higher negative values suggest better performance of original Hargreaves-Samani ETo method while higher positive values suggest better performance of original Priestley-Taylor ETo method using as reference the ASCE-short | the zip contains 1 raster (ESRI-grid) |
| 6 | Mean monthly ASCE-ETo for short reference crop (clipped grass) (mm/month)/(30 arc-sec) | etos1_30s.zip | ASCE-EWRI method (Allen et al., 2005) using climatic data from Hijmans et al. (2005) and Sheffield et al. (2006) | the zip contains 12 rasters (ESRI-grids) for each month (January is the first month) |
| 7 | Mean monthly ASCE-ETo for tall reference crop (alfalfa) (mm/month)/(30 arc-sec) | etot1_30s.zip | ASCE-EWRI method (Allen et al., 2005) using climatic data from Hijmans et al. (2005) and Sheffield et al. (2006) | the zip contains 12 rasters (ESRI-grids) for each month (January is the first month) |
| 8 | Re-adjusted coefficient for solar radiation formula of Hargreaves-Samani (rescaled ×1000) (unitless)/(30 arc-sec) | krs1_30s.zip | Re-calibration of Hargreaves-Samani coefficient krs=0.16-0.19 for solar radiation formula (Hargreaves and Samani, 1982, 1985) using solar radiation data (from Sheffield et al., 2006) | the zip contains 1 raster (ESRI-grid) (partial weighted average of mean monthly values) |

| | | | |
|---|---|---|---|
| 9 | Expected Mean Annual Difference/Error (MAD%) between original Hargreaves-Samani ETo and ASCE-ETo for short ref.crop (%)/(30 arc-sec) | madhs1_30s.zip | 100*[(Annual ETo H-S)-(Annual ETo ASCE-short)]/(Annual ETo ASCE-short), Annual ETo H-S is estimated with the typical value chs2=0.0023 | the zip contains 1 raster (ESRI-grid) |
| 10 | Expected Mean Annual Difference/Error (MAD%) between original Priestley-Taylor ETo and ASCE-ETo for short ref.crop (%)/(30 arc-sec) | madpt1_30s.zip | 100*[(Annual ETo P-T)-(Annual ETo ASCE-short)]/(Annual ETo ASCE-short), Annual ETo P-T is estimated with the typical value apt=1.26 | the zip contains 1 raster (ESRI-grid) |
| 11 | Expected Mean Annual Difference/Error (MAD%) between original Hargreaves-Samani radiation formula versus solar radiation data (%)/(30 arc-sec) | madrs1_30s.zip | 100*[(Annual RS of H-S)-(Annual RS data)]/(Annual RS data), Annual RS H-S is estimated with the typical value krs=0.17 and RS obtained from Sheffield et al. (2006) | the zip contains 1 raster (ESRI-grid) |
| 12 | Re-adjusted Priestley-Taylor coefficient for short ref.crop ETo (rescaled ×100) (unitless)/(2.5 arc-min) | apts2_2-5m.zip | Re-calibration of Priestley-Taylor coefficient apt=1.26 for ETo method (Priestley and Taylor, 1972) using ASCE-EWRI method (Allen et al., 2005) for short ref.crop | the zip contains 1 raster (ESRI-grid) (partial weighted average of mean monthly values) |
| 13 | Re-adjusted Priestley-Taylor coefficient for tall ref.crop ETo (rescaled ×100) (unitless)/(2.5 arc-min) | aptt2_2-5m.zip | Re-calibration of Priestley-Taylor coefficient apt=1.26 for ETo method (Priestley and Taylor, 1972) using ASCE-EWRI method (Allen et al., 2005) for tall ref.crop | the zip contains 1 raster (ESRI-grid) (partial weighted average of mean monthly values) |
| 14 | Re-adjusted Hargreaves-Samani coefficient for short ref.crop ETo (rescaled ×100,000) (unitless)/(2.5 arc-min) | chs2s2_2-5m.zip | Re-calibration of Hargreaves-Samani coefficient chs2=0.0023 for ETo method (Hargreaves and Samani, 1982, 1985) using ASCE-EWRI method (Allen et al., 2005) for short ref.crop | the zip contains 1 raster (ESRI-grid) (partial weighted average of mean monthly values) |
| 15 | Re-adjusted Hargreaves-Samani coefficient for tall ref.crop ETo (rescaled ×100,000) (unitless)/(2.5 arc-min) | chs2t2_2-5m.zip | Re-calibration of Hargreaves-Samani coefficient chs2=0.0023 for ETo method (Hargreaves and Samani, 1982, 1985) using ASCE-EWRI method (Allen et al., 2005) for tall ref.crop | the zip contains 1 raster (ESRI-grid) (partial weighted average of mean monthly values) |
| 16 | Hargeaves-Samani versus Priestley-Taylor (comparison between original methods versus ASCE-short) (DMADhp) (%)/(2.5 arc-min) | dmadhp2_2-5m.zip | abs(madhs)-abs(madpt), higher negative values suggest better performance of original Hargreaves-Samani ETo method while higher positive values suggest better performance of original Priestley-Taylor ETo method using as reference the ASCE-short | the zip contains 1 raster (ESRI-grid) |
| 17 | Mean monthly ASCE-ETo for short reference crop (clipped grass) (mm/month)/(2.5 arc-min) | etos2_2-5m.zip | ASCE-EWRI method (Allen et al., 2005) using climatic data from Hijmans et al. (2005) and Sheffield et al. (2006) | the zip contains 12 rasters (ESRI-grids) for each month (January is the first month) |

| 18 | Mean monthly ASCE-ETo for tall reference crop (alfalfa) (mm/month)/(2.5 arc-min) | etot2_2-5m.zip | ASCE-EWRI method (Allen et al., 2005) using climatic data from Hijmans et al. (2005) and Sheffield et al. (2006) | the zip contains 12 rasters (ESRI-grids) for each month (January is the first month) |
|---|---|---|---|---|
| 19 | Re-adjusted coefficient for solar radiation formula of Hargreaves-Samani (rescaled ×1000) (unitless)/(2.5 arc-min) | krs2_2-5m.zip | Re-calibration of Hargreaves-Samani coefficient krs=0.16-0.19 for solar radiation formula (Hargreaves and Samani, 1982, 1985) using solar radiation data (from Sheffield et al., 2006) | the zip contains 1 raster (ESRI-grid) (partial weighted average of mean monthly values) |
| 20 | Expected Mean Annual Difference/Error (MAD%) between original Hargreaves-Samani ETo and ASCE-ETo for short ref.crop (%)/(2.5 arc-min) | madhs2_2-5m.zip | 100*[(Annual ETo H-S)-(Annual ETo ASCE-short)]/(Annual ETo ASCE-short), Annual ETo H-S is estimated with the typical value chs2=0.0023 | the zip contains 1 raster (ESRI-grid) |
| 21 | Expected Mean Annual Difference/Error (MAD%) between original Priestley-Taylor ETo and ASCE-ETo for short ref.crop (%)/(2.5 arc-min) | madpt2_2-5m.zip | 100*[(Annual ETo P-T)-(Annual ETo ASCE-short)]/(Annual ETo ASCE-short), Annual ETo P-T is estimated with the typical value apt=1.26 | the zip contains 1 raster (ESRI-grid) |
| 22 | Expected Mean Annual Difference/Error (MAD%) between original Hargreaves-Samani radiation formula versus solar radiation data (%)/(2.5 arc-min) | madrs2_2-5m.zip | 100*[(Annual RS of H-S)-(Annual RS data)]/(Annual RS data), Annual RS H-S is estimated with the typical value krs=0.17 and RS obtained from Sheffield et al. (2006) | the zip contains 1 raster (ESRI-grid) |
| 23 | Re-adjusted Priestley-Taylor coefficient for short ref.crop ETo (rescaled ×100) (unitless)/(5 arc-min) | apts3_5m.zip | Re-calibration of Priestley-Taylor coefficient apt=1.26 for ETo method (Priestley and Taylor, 1972) using ASCE-EWRI method (Allen et al., 2005) for short ref.crop | the zip contains 1 raster (ESRI-grid) (partial weighted average of mean monthly values) |
| 24 | Re-adjusted Priestley-Taylor coefficient for tall ref.crop ETo (rescaled ×100) (unitless)/(5 arc-min) | aptt3_5m.zip | Re-calibration of Priestley-Taylor coefficient apt=1.26 for ETo method (Priestley and Taylor, 1972) using ASCE-EWRI method (Allen et al., 2005) for tall ref.crop | the zip contains 1 raster (ESRI-grid) (partial weighted average of mean monthly values) |
| 25 | Re-adjusted Hargreaves-Samani coefficient for short ref.crop ETo (rescaled ×100,000) (unitless)/(5 arc-min) | chs2s3_5m.zip | Re-calibration of Hargreaves-Samani coefficient chs2=0.0023 for ETo method (Hargreaves and Samani, 1982, 1985) using ASCE-EWRI method (Allen et al., 2005) for short ref.crop | the zip contains 1 raster (ESRI-grid) (partial weighted average of mean monthly values) |
| 26 | Re-adjusted Hargreaves-Samani coefficient for tall ref.crop ETo (rescaled ×100,000) (unitless)/(5 arc-min) | chs2t3_5m.zip | Re-calibration of Hargreaves-Samani coefficient chs2=0.0023 for ETo method (Hargreaves and Samani, 1982, 1985) using ASCE-EWRI method (Allen et al., 2005) for tall ref.crop | the zip contains 1 raster (ESRI-grid) (partial weighted average of mean monthly values) |

| 27 | Hargeaves-Samani versus Priestley-Taylor (comparison between original methods versus ASCE-short) (DMADhp) (%)/(5 arc-min) | dmadhp3_5m.zip | abs(madhs)-abs(madpt), higher negative values suggest better performance of original Hargreaves-Samani ETo method while higher positive values suggest better performance of original Priestley-Taylor ETo method using as reference the ASCE-short | the zip contains 1 raster (ESRI-grid) |
|---|---|---|---|---|
| 28 | Mean monthly ASCE-ETo for short reference crop (clipped grass) (mm/month)/(5 arc-min) | etos3_5m.zip | ASCE-EWRI method (Allen et al., 2005) using climatic data from Hijmans et al. (2005) and Sheffield et al. (2006) | the zip contains 12 rasters (ESRI-grids) for each month (January is the first month) |
| 29 | Mean monthly ASCE-ETo for tall reference crop (alfalfa) (mm/month)/(5 arc-min) | etot3_5m.zip | ASCE-EWRI method (Allen et al., 2005) using climatic data from Hijmans et al. (2005) and Sheffield et al. (2006) | the zip contains 12 rasters (ESRI-grids) for each month (January is the first month) |
| 30 | Re-adjusted coefficient for solar radiation formula of Hargreaves-Samani (rescaled ×1000) (unitless)/(5 arc-min) | krs3_5m.zip | Re-calibration of Hargreaves-Samani coefficient krs=0.16-0.19 for solar radiation formula (Hargreaves and Samani, 1982, 1985) using solar radiation data (from Sheffield et al., 2006) | the zip contains 1 raster (ESRI-grid) (partial weighted average of mean monthly values) |
| 31 | Expected Mean Annual Difference/Error (MAD%) between original Hargreaves-Samani ETo and ASCE-ETo for short ref.crop (%)/(5 arc-min) | madhs3_5m.zip | 100*[(Annual ETo H-S)-(Annual ETo ASCE-short)]/(Annual ETo ASCE-short), Annual ETo H-S is estimated with the typical value chs2=0.0023 | the zip contains 1 raster (ESRI-grid) |
| 32 | Expected Mean Annual Difference/Error (MAD%) between original Priestley-Taylor ETo and ASCE-ETo for short ref.crop (%)/(5 arc-min) | madpt3_5m.zip | 100*[(Annual ETo P-T)-(Annual ETo ASCE-short)]/(Annual ETo ASCE-short), Annual ETo P-T is estimated with the typical value apt=1.26 | the zip contains 1 raster (ESRI-grid) |
| 33 | Expected Mean Annual Difference/Error (MAD%) between original Hargreaves-Samani radiation formula versus solar radiation data (%)/(5 arc-min) | madrs3_5m.zip | 100*[(Annual RS of H-S)-(Annual RS data)]/(Annual RS data), Annual RS H-S is estimated with the typical value krs=0.17 and RS obtained from Sheffield et al. (2006) | the zip contains 1 raster (ESRI-grid) |
| 34 | Re-adjusted Priestley-Taylor coefficient for short ref.crop ETo (rescaled ×100) (unitless)/(10 arc-min) | apts4_10m.zip | Re-calibration of Priestley-Taylor coefficient apt=1.26 for ETo method (Priestley and Taylor, 1972) using ASCE-EWRI method (Allen et al., 2005) for short ref.crop | the zip contains 1 raster (ESRI-grid) (partial weighted average of mean monthly values) |
| 35 | Re-adjusted Priestley-Taylor coefficient for tall ref.crop ETo (rescaled ×100) (unitless)/(10 arc-min) | aptt4_10m.zip | Re-calibration of Priestley-Taylor coefficient apt=1.26 for ETo method (Priestley and Taylor, 1972) using ASCE-EWRI method (Allen et al., 2005) for tall ref.crop | the zip contains 1 raster (ESRI-grid) (partial weighted average of mean monthly values) |

| 36 | Re-adjusted Hargreaves-Samani coefficient for short ref.crop ETo (rescaled ×100,000) (unitless)/(10 arc-min) | chs2s4_10m.zip | Re-calibration of Hargreaves-Samani coefficient chs2=0.0023 for ETo method (Hargreaves and Samani, 1982, 1985) using ASCE-EWRI method (Allen et al., 2005) for short ref.crop | the zip contains 1 raster (ESRI-grid) (partial weighted average of mean monthly values) |
|---|---|---|---|---|
| 37 | Re-adjusted Hargreaves-Samani coefficient for tall ref.crop ETo (rescaled ×100,000) (unitless)/(10 arc-min) | chs2t4_10m.zip | Re-calibration of Hargreaves-Samani coefficient chs2=0.0023 for ETo method (Hargreaves and Samani, 1982, 1985) using ASCE-EWRI method (Allen et al., 2005) for tall ref.crop | the zip contains 1 raster (ESRI-grid) (partial weighted average of mean monthly values) |
| 38 | Hargeaves-Samani versus Priestley-Taylor (comparison between original methods versus ASCE-short) (DMADhp) (%)/(10 arc-min) | dmadhp4_10m.zip | abs(madhs)-abs(madpt), higher negative values suggest better performance of original Hargreaves-Samani ETo method while higher positive values suggest better performance of original Priestley-Taylor ETo method using as reference the ASCE-short | the zip contains 1 raster (ESRI-grid) |
| 39 | Mean monthly ASCE-ETo for short reference crop (clipped grass) (mm/month)/(10 arc-min) | etos4_10m.zip | ASCE-EWRI method (Allen et al., 2005) using climatic data from Hijmans et al. (2005) and Sheffield et al. (2006) | the zip contains 12 rasters (ESRI-grids) for each month (January is the first month) |
| 40 | Mean monthly ASCE-ETo for tall reference crop (alfalfa) (mm/month)/(10 arc-min) | etot4_10m.zip | ASCE-EWRI method (Allen et al., 2005) using climatic data from Hijmans et al. (2005) and Sheffield et al. (2006) | the zip contains 12 rasters (ESRI-grids) for each month (January is the first month) |
| 41 | Re-adjusted coefficient for solar radiation formula of Hargreaves-Samani (rescaled ×1000) (unitless)/(10 arc-min) | krs4_10m.zip | Re-calibration of Hargreaves-Samani coefficient krs=0.16-0.19 for solar radiation formula (Hargreaves and Samani, 1982, 1985) using solar radiation data (from Sheffield et al., 2006) | the zip contains 1 raster (ESRI-grid) (partial weighted average of mean monthly values) |
| 42 | Expected Mean Annual Difference/Error (MAD%) between original Hargreaves-Samani ETo and ASCE-ETo for short ref.crop (%)/(10 arc-min) | madhs4_10m.zip | 100*[(Annual ETo H-S)-(Annual ETo ASCE-short)]/(Annual ETo ASCE-short), Annual ETo H-S is estimated with the typical value chs2=0.0023 | the zip contains 1 raster (ESRI-grid) |
| 43 | Expected Mean Annual Difference/Error (MAD%) between original Priestley-Taylor ETo and ASCE-ETo for short ref.crop (%)/(10 arc-min) | madpt4_10m.zip | 100*[(Annual ETo P-T)-(Annual ETo ASCE-short)]/(Annual ETo ASCE-short), Annual ETo P-T is estimated with the typical value apt=1.26 | the zip contains 1 raster (ESRI-grid) |
| 44 | Expected Mean Annual Difference/Error (MAD%) between original Hargreaves-Samani radiation formula versus solar radiation data (%)/(10 arc-min) | madrs4_10m.zip | 100*[(Annual RS of H-S)-(Annual RS data)]/(Annual RS data), Annual RS H-S is estimated with the typical value krs=0.17 and RS obtained from Sheffield et al. (2006) | the zip contains 1 raster (ESRI-grid) |

| 45 | Re-adjusted Priestley-Taylor coefficient for short ref.crop ETo (rescaled ×100) (unitless)/(0.5 deg) | apts5_0-5d.zip | Re-calibration of Priestley-Taylor coefficient apt=1.26 for ETo method (Priestley and Taylor, 1972) using ASCE-EWRI method (Allen et al., 2005) for short ref.crop | the zip contains 1 raster (ESRI-grid) (partial weighted average of mean monthly values) |
|---|---|---|---|---|
| 46 | Re-adjusted Priestley-Taylor coefficient for tall ref.crop ETo (rescaled ×100) (unitless)/(0.5 deg) | aptt5_0-5d.zip | Re-calibration of Priestley-Taylor coefficient apt=1.26 for ETo method (Priestley and Taylor, 1972) using ASCE-EWRI method (Allen et al., 2005) for tall ref.crop | the zip contains 1 raster (ESRI-grid) (partial weighted average of mean monthly values) |
| 47 | Re-adjusted Hargreaves-Samani coefficient for short ref.crop ETo (rescaled ×100,000) (unitless)/(0.5 deg) | chs2s5_0-5d.zip | Re-calibration of Hargreaves-Samani coefficient chs2=0.0023 for ETo method (Hargreaves and Samani, 1982, 1985) using ASCE-EWRI method (Allen et al., 2005) for short ref.crop | the zip contains 1 raster (ESRI-grid) (partial weighted average of mean monthly values) |
| 48 | Re-adjusted Hargreaves-Samani coefficient for tall ref.crop ETo (rescaled ×100,000) (unitless)/(0.5 deg) | chs2t5_0-5d.zip | Re-calibration of Hargreaves-Samani coefficient chs2=0.0023 for ETo method (Hargreaves and Samani, 1982, 1985) using ASCE-EWRI method (Allen et al., 2005) for tall ref.crop | the zip contains 1 raster (ESRI-grid) (partial weighted average of mean monthly values) |
| 49 | Hargeaves-Samani versus Priestley-Taylor (comparison between original methods versus ASCE-short) (DMADhp) (%)/(0.5 deg) | dmadhp5_0-5d.zip | abs(madhs)-abs(madpt), higher negative values suggest better performance of original Hargreaves-Samani ETo method while higher positive values suggest better performance of original Priestley-Taylor ETo method using as reference the ASCE-short | the zip contains 1 raster (ESRI-grid) |
| 50 | Mean monthly ASCE-ETo for short reference crop (clipped grass) (mm/month)/(0.5 deg) | etos5_0-5d.zip | ASCE-EWRI method (Allen et al., 2005) using climatic data from Hijmans et al. (2005) and Sheffield et al. (2006) | the zip contains 12 rasters (ESRI-grids) for each month (January is the first month) |
| 51 | Mean monthly ASCE-ETo for tall reference crop (alfalfa) (mm/month)/(0.5 deg) | etot5_0-5d.zip | ASCE-EWRI method (Allen et al., 2005) using climatic data from Hijmans et al. (2005) and Sheffield et al. (2006) | the zip contains 12 rasters (ESRI-grids) for each month (January is the first month) |
| 52 | Re-adjusted coefficient for solar radiation formula of Hargreaves-Samani (rescaled ×1000) (unitless)/(0.5 deg) | krs5_0-5d.zip | Re-calibration of Hargreaves-Samani coefficient krs=0.16-0.19 for solar radiation formula (Hargreaves and Samani, 1982, 1985) using solar radiation data (from Sheffield et al., 2006) | the zip contains 1 raster (ESRI-grid) (partial weighted average of mean monthly values) |
| 53 | Expected Mean Annual Difference/Error (MAD%) between original Hargreaves-Samani ETo and ASCE-ETo for short ref.crop (%)/(0.5 deg) | madhs5_0-5d.zip | 100*[(Annual ETo H-S)-(Annual ETo ASCE-short)]/(Annual ETo ASCE-short), Annual ETo H-S is estimated with the typical value chs2=0.0023 | the zip contains 1 raster (ESRI-grid) |

| 54 | Expected Mean Annual Difference/Error (MAD%) between original Priestley-Taylor ETo and ASCE-ETo for short ref.crop (%)/(0.5 deg) | madpt5_0-5d.zip | 100*[(Annual ETo P-T)-(Annual ETo ASCE-short)]/(Annual ETo ASCE-short), Annual ETo P-T is estimated with the typical value apt=1.26 | the zip contains 1 raster (ESRI-grid) |
|---|---|---|---|---|
| 55 | Expected Mean Annual Difference/Error (MAD%) between original Hargreaves-Samani radiation formula versus solar radiation data (%)/(0.5 deg) | madrs5_0-5d.zip | 100*[(Annual RS of H-S)-(Annual RS data)]/(Annual RS data), Annual RS H-S is estimated with the typical value krs=0.17 and RS obtained from Sheffield et al. (2006) | the zip contains 1 raster (ESRI-grid) |

---

## Author Comment (AC2) · 21 Jun 2017

Comment R2.1: This study analyses the global relationship between ETo annual maps obtained by means of three different methods (ASCE PM, P-T and H-S), with the purpose of determining the accuracy of the methods based on poor data availability (P-T and H-S). In addition, the manuscript includes a calibration exercise to obtain revised coefficients at the global scale for P-T and H-S equations based on the obtained ASCE

PM data. The research topic is highly relevant given the relevance of estimating the atmospheric evaporative demand (AED) with accuracy since AED is an important hydroclimatic variable with strong implications in aridity conditions and climate change processes. The manuscript is in general well written, the figures show high quality and it has a good structure. The authors use a high amount of data for analysis and validation, including gridded datasets and meteorological networks in California and Australia. The manuscript is a bit long and sometimes it is different to follow but independently of formal issues I find major methodological problems in the manuscript, which are related to the treatment of the data used, the spatial resolution of the gridded products and the assessment of the uncertainty in the ETo estimations. I am including some detailed issues below about these issues.

Response: We would like to thank the reviewer for the precise comments related to methodological problems since he gave us the opportunity to provide more justifications, clarifications and details about the methods and data used in this work. We carefully considered all the comments and we followed all his recommendations in order to improve the manuscript and reduce any uncertainties related to methodological issues about the spatial resolutions. More details are given to the responses of the following specific comments. We also suggest the reviewer to check carefully the responses to the comments of Reviewer 1, since his suggestions led to substantial changes in the manuscript.

Comment R2.2: I would recommend the authors to work at coarser spatial resolution to reduce the strong uncertainty associated to the selected high resolution (1 km) of final products. Page 4: I find highly problematic to interpolate the low resolution 0.5âUe data for wind speed, humidity and solar radiation to 1 km. The results of the bilinear interpolation of the 0.5âUe data does not really increase the necessary spatial resolution of these variables to be compared with the high resolution of tmax and tmin data (in any case high resolution temperature data from the global dataset used is also affected by spatial errors and uncertainties, which should be also taken into account). The 1 km in-
terpolated wind speed, humidity and solar radiation has a spatial resolution completely unreal. These variables are essential to be taken into account to estimate ETo spatial patterns since ETo is usually more sensitive to these variables than to temperature (McVicar et al., 2012a and b). For this reason, I consider that 1 km gridded maps generated in this study show high uncertainty, which is not quantified/provided in this study. The authors are computing Eto by PM equation as reference to be compared with H-S and P-T methods, but there is not any assessment of the error in the PM estimations related to the data inaccuracies and the poor resolution of the input climate data. I think these problems would be solved (not completely since an assessment of uncertainty should be taken into account) if authors consider to focus at coarse (0.5âŮę) spatial resolution, which avoids unnecessary interpolation of wind speed, radiation and humidity variables and the outputs would be useful for continental to global assessments. Thus, the results of figures 8-10 confirms that interpolation of low resolution variables have strong influence on the comparability of different ETo estimations, which can be associated to the poor interpolation approach applied to the coarse climate variables.

Response: We agree with the reviewer that the bilinear interpolation method may not be the most appropriate method to increase the resolution of wind speed, solar radiation and humidity data and we also agree that 1 km raster resolutions are in general an exaggeration for describing climatic variables. The basic reason that led us to show the results of both ~1 km (30 arc-sec) and 0.5 deg was to cover the complete range of resolutions observed in the initial data. In addition, the aim was to provide ETo rasters of 1 km for comparative purposes with other studies, which have also provided 1 km resolutions of global ETo for the same period using other methods and the same sources of data. For example, Zomer et al. (2008) provided 1 km resolution maps using the Hargreaves-Samani method based on the temperature data of Hijmans et al. (2005). The bilinear interpolation used for global solar radiation, specific humidity and wind speed data of Sheffield et al. (2006) provided insignificant improvement but allowed to develop 1 km rasters of exact spatial arrangement with the 1 km rasters of temperature, especially in the coastlines and small islands. This has provided an
improvement of ~4% in the RMSE of ASCE-ETo estimations obtained from the map, when compared with the respective values of stations (Fig.R1a,b). This improvement seems negligible for the total validation dataset, but it was significant when it was examined for some individual stations located in regions of 0.5 degree pixels with high internal topographic-temperature variability. In order to avoid any criticism about the interpolation method used for increasing the resolution of solar radiation, humidity and wind speed data, we decided to remove any results and discussion about the finer resolutions keeping only the results for 0.5 degree resolution. For this reason all the results and all the maps and tables presented in the revised version correspond only to the 0.5 deg resolution. The comparisons between the ETo values of rasters (0.5 degree) and stations for both reference crops were added in the supplementary material (see Fig.S2g,h) and their reference in the text can be found in Page 10, lines 25-26.

[FIGURE R1, PLACE HERE]

Fig.R1 Comparison of ETo ASCE-short values (mm month-1) between the 140 stations (both CA-USA and Australia stations) and (a) the produced rasters of 30 arc-sec resolution and (b) the produced rasters of 0.5 degree resolution.

The only reference about the finer resolutions is given in section 5. Data availability, where we added the following text "Apart from the 0.5 degree resolution raster datasets, the database contains the same datasets at finer resolution (30 arc-sec, 2.5 arc-min, 5 arc-min and 10 arc-min). These finer datasets are provided in order to cover the observed resolution range in the initial climatic data (e.g. the temperature data of Hijmans et al. (2005) are provided at 30 arc-sec resolution). The finer resolutions were produced using bilinear interpolation on solar radiation, humidity and wind speed data of Sheffield et al. (2006). This interpolation method is not the most appropriate for such purposes. The data of finer resolutions can only be used as a tool to assess uncertainties associated to temperature variation effects within a 0.5 degree pixel or to estimate average values of the coefficients for larger territories in order to capture a better representation of the coastlines or islands that do not exist in 0.5 degree resoluESSDD
tion (use of values from individual pixels is not recommended). A complete list of the datasets is provided in the Table S5."

Comment R2.3: I have also doubts on the use of the coefficients calculated in this study to calculate ETo using H-S and P-T equations. The authors obtain the calibration coefficients for the period 1950-2000 and assume stationary climate conditions. Nevertheless, under climate scenarios in which input climate variables change (I refer to wind speed, relative humidity and incoming solar radiation) under a non-stationary scenario, the obtained coefficients would not be useful to calculate ETo based on scarce climate data. Different studies have showed recent changes in solar radiation (Wild et al., 2013), wind speed (McVicar et al., 2012b) and atmospheric humidity (Willet et al. 2014). Given that the main objective of this study is the re-calibration of the H-S and P-T equations, it would be necessary that authors provide not only the recalibrated coefficients but also a measure of the accuracy considering the errors in the interpolated variables used in P-M calculations.

Response: We agree with the reviewer that climate change effects can significantly affect the prediction accuracy of the coefficients. This was the reason why we included data from 2000-2016 for all stations in the validation procedure (see periods of observations for each station in Table 1 of the manuscript). We also have to mention that more than 50% of stations in the total validation dataset have more data from years after 2000, while there are 4 stations with data only for the period ~2000-2016. Taking into account these specific features of the validation dataset, we cannot reject the hypothesis that the revised coefficients have a good explanatory power even for the years 2000-2016, since they improved significantly the ETo predictions in comparison to the standard coefficients and gave better results from other models that use additional parameters (see new additional models in Table 2 and comparative results in Table 5 of the manuscript; the additional models were added after the request of Revier 1). We also thought to break the validation datasets into two periods (before and after 2000), since the produced ETo rasters and the revised coefficients correspond to
1950-2000, but this idea could not be implemented for two reasons: aAć The database of Australian stations provides freely available online only the mean monthly values of the parameters for the total periods of observations and not the complete records of monthly values for each specific year. ć The available data from CIMIS database for the stations of California-USA start after 1982 (Table 1 in the manuscript), and cover less than 36% of the total period 1950-2000, while they correspond to the late years of the specific period. It is well documented that climate differences were also observed even during the 1950-2000 (through comparisons between 1950-1975 and 1975-2000) in many parts of the world (Hang et al., 2000; Norrant and Douguédroit, 2006; Sheffield and Wood, 2008; You et al., 2011). Thus, we also did not divide the California-USA data into two periods in order to avoid any arguments that would probably occur based on the observation periods. Since we followed the recommendation of the reviewer to remove the finer resolution results of 30 arc-sec (1 km) and present only the 0.5 degree results, we also performed an accuracy analysis for the internal parameters of ASCE-ETo between the values provided by the 0.5 degree rasters data (Hijmans et al. 2005; Sheffield et al. 2006) and the respective data of stations (Fig.R2 below). The temperature data of 30 arc-sec resolution were also converted to 0.5 deg for this analysis. Fig.R2a,b,c,d,e,f provides the respective comparisons for the mean monthly values of Tmax, Tmin, Rs, Rn, DE (vapour pressure deficit es-ea), and u2 between stations data and rasters of 0.5 degree resolution. The Rs values of both rasters and stations given in Fig.R2c are those after correcting the ones exceeding the clear sky solar radiation Rso (i.e. when Rs/Rso>1, Rs=Rso), as it is required before ASCE-ETo estimations (Allen et al., 1998; 2005). Additionally, the values of u2 given in Fig.R2f are those after adjusting the raster values of Sheffield et al. (2006) and Australia stations data from z=10 m to 2 m height using the formula (Allen et al., 1998; 2005):

u2=4.87\*uz/(ln(67.8z-5.42)) (Eq.R1)

The original wind data of Sheffield et al. (2006) and Australia stations are given for z=10 m, while the data of California stations were already given at 2 m height. The
comparisons for the Tmax, Tmin, Rs, Rn, DE (vapour pressure deficit es-ea), and u2 between stations data and rasters of 0.5 degree resolution were added in the supplementary material (see Fig.S2a,b,c,d,e,f) and their reference in the text can be found in Page 10, lines 25-26.

[FIGURE R2, PLACE HERE]

Fig.R2 Comparison of mean monthly values between rasters data (0.5 degree resolution) and stations data for (a) the maximum monthly temperature Tmax, (b) the minimum monthly temperature Tmin, (c) the solar radiation Rs, (d) the net solar radiation Rn, (e) the vapour pressure deficit DE=es-ea, and (f) the wind speed at 2 m height u2.

For the cases of Tmax, Tmin, Rs, Rn and DE (Fig.R2a,b,c,d,e), the comparisons between rasters and stations are satisfactory, if we consider that rasters provide values of 0.5 degree ( $\sim$ 50 km) pixels of the period 1950-2000 while stations data cover also the period from 2000-2016. In the case of u2, the correlation between rasters and stations data was not good. We examined with various ways the wind data in order to explain the possible sources of this problem. We derived some findings when comparing the mean monthly u2 values of all California-USA (Fig.R3a) and Australia (Fig.R3b) stations, separately. Fig.R3a shows that the total average raster values of mean monthly u2 from the pixel positions of CA-USA stations are higher than the respective measured u2 values, while in Fig.R3b for Australia stations is observed the opposite trend. These differences are the main reason why the regression line in Fig.R2f is above the 45 degrees line for the values <2.5 m s-1 (the majority of points belong to CA-USA stations) and below the 45 degrees line for the values >2.5 m s-1 (the majority of points belong to Ausralia stations). This opposite trend between the two validation datasets was also the reason of the high RMSE in Fig.R2f. To avoid any possible misunderstanding that could arise from the merged California and Australia datasets, we also give the results of Fig.R1b separately for CA-USA and Australia stations in Fig.R4a,b, respectively. Despite the difference in u2 values between CA-USA stations and rasters (Figs.R3a), the regression line in Fig.R4a of ETo presents a good slope and intercept Interactive comment

probably because the wind differences are counterbalanced with differences in other parameters of ETo. In the case of Australia stations (Fig.R4b), the higher observed u2 values from stations in comparison to rasters are probably the reason for the observed downward deviation of regression line from the 450 degrees line.

[FIGURE R3, PLACE HERE]

Fig.R3 Comparison of mean monthly u2 values through Box-Whisker plots: (a) between 0.5 degree rasters (Sheffield et al., 2006) and California-USA stations, (b) between 0.5 degree rasters (Sheffield et al., 2006) and Australia stations.

[FIGURE R4, PLACE HERE]

Fig.R4 Comparison of ETo ASCE-short values (mm month-1) between (a) the produced rasters of 0.5 degree and the 60 stations of CA-USA and (b) the produced rasters of 0.5 degree and the 80 stations of Australia.

Some justifications about the low correlation between wind data of rasters and stations (Fig.R2f) and the observed differences in Figs.R3a,b are the following:  $\hat{a}A\dot{c}$  Part of this difference may be associated to climate change effects since the larger part of wind data from stations, especially for Australia stations, represent the period after 2000 while the rasters correspond to the period 1950-2000.  $\hat{a}A\dot{c}$  The representativity of wind speed rasters of 1950-2000 produced by the model of Sheffield et al. (2006) may be low at 0.5 degree resolution due to the scarce existing wind data at global scale during the total period of simulation and especially for the years belonging to the first half of the total period.  $\hat{a}A\dot{c}$  An additional factor responsible for the differences in Figs.3a,b may be the conversion of wind raster data of Sheffield et al. (2006) from z=10 m to 2 m using Eq.R1. The degree of accuracy of this equation is unknown when is applied at global scale and for a pixel of 0.5 degree resolution, which may contain high topographical variability. The error, which may be introduced by the use of Eq.R1 is impossible to be assessed.  $\hat{a}A\dot{c}$  In the case of CA-USA stations, the mean monthly u2 values (measured directly at 2 m height) were estimated after removing extremely high observed values,

ESSDD
which were flagged by the CIMIS database as unreasonable extremes. Additionally, in some months of some stations, u2 values were missing. We also observed that many of these extreme and missing values were during months of extreme rainfall events. Many of these months were associated to extreme hurricane events, which are very common in California (at least 54 catastrophic events for the period 1950-2015, with extremely high wind speeds). For example, the Guillermo hurricane of 1997 led to wind speeds of ~70 m s-1 (https://en.wikipedia.org/wiki/List of California hurricanes). We had already mentioned in the initial version of the manuscript that we removed flagged values from CA-USA data in order to make comparisons with the given ETo values provided by the CIMIS database (Page 10, lines 13-21 and Fig.S1 in the supplementary material). On the other hand, we believe that in the climatic model of Sheffield et al. (2006), which is expanded also in the oceans, such events were included (the degree of inclusion is unknown) and this may be probably an additional reason of the larger pixel values observed in the wind rasters at the positions of CA-USA stations (Fig.R3a). âÅć In the case of Australian stations, the AGBM database (Australian Government - Bureau of Meteorology) provides 12 values of mean monthly wind speeds of the total observation periods for 9am and another 12 values for 3pm local time (the website mentions that wind speeds are generally measured at 10 m height). Thus, we estimated the average value of 9am and 3pm conditions in order to get the mean monthly wind speeds and then we used Eq.R1 to adjust them at 2 m height. Thus, it is unknown the degree of error by averaging the 9am and 3pm conditions in order to get the mean monthly wind speeds and also unknown the possible error by the use of Eq.R1 locally at the position of stations. This equation is usually not calibrated for meteorological stations with anemometers positioned above 2 m height.

Such uncertainties may also exist in the case of DE=es-ea (Fig.R2e), since Sheffield et al. (2006) provides data of specific humidity that were directly converted to actual vapour pressure ea using the equation of Peixoto and Oort (1996). This equation uses the additional parameter of atmospheric pressure as internal parameter. The atmospheric pressure in the case of rasters was estimated based on elevation data
of 1 km resolution that were further converted to 0.5 degree resolution. The use of ea data from 0.5 degree resolution pixels may also added additional error, especially when there is large topographic variability within the 0.5 degree pixel. On the other hand, the ea of stations was estimated by relative humidity and temperature data.

Taking into account all the aforementioned observations, we would like to summarize our conclusions related to the specific comment:  $\hat{a}Ac$  Apart from the wind speed data, it was found an adequate correspondence between the 0.5 degree raster data of Tmax, Tmin, Rs, Rn and de of 1950-2000 with the respective values of stations, which are expanded until 2016. aAc As regards the wind speed data, the discrepancy between rasters and stations can be justified: a) either by possible wind differences before and after 2000, b) or by the effect of Eq.R1, which is used to adjust the wind rasters and the wind data of Australia stations from 10 to 2 m height, c) or by uncertainties in the Sheffield et al. (2006) wind data due to the scarce existing wind data for calibrating their model at global scale during the period of 1950-2000 (especially for years before 1975). d) or by uncertainties introduced after eliminating extreme wind values in the data of CA-USA stations, e) or by uncertainties introduced after averaging the 9am and 3pm wind conditions in the data of Australia stations, f) or by combinations of all the aforementioned cases. Thus, uncertainties exist in both rasters and stations wind data, which can not be solved. These specific problems were included in the discussion section. Despite the differences in the wind speed data between rasters and stations, the observed correlation between ETo ASCE-short of 1950-2000 (0.5 degree resolution) and the respective values of California-USA and Australia stations (which are expanded until 2016) is adequate for a global scale application (Fig.R1b), if we consider a) that the ETo values of rasters were obtained from large pixels ( $\sim$ 50 km) and b) that uncertainties, especially in the wind datasets, exist not only in the raster datasets but also in the stations datasets. In order to prove that the re-adjusted coefficients of P-T and H-S methods are valuable, we included other models of reduced parameters from the literature in order to perform comparisons (see new additional models in Table 2 and comparative results in Table 5 of the manuscript). In order to provide information

ESSDD
about the aforementioned uncertainties related to the data that may affect the validity of the revised coefficients, we added a new section in the Discussion with title "Uncertainties in the data used for calibrating and validating the revised coefficients of P-T and H-S methods"

Finally, we would like to stress that this study used the published data of Hijmans et al. (2005) and Sheffield et al. (2006) that have been used by too many other studies (7129 and 186 citations, respectively, source: SCOPUS, last accessed 12/6/2017). We believe that we used the best available global information for developing the rasters. Additionally, we clearly state that our products of reference evapotranspiration and revised coefficients correspond to 1950-2000 and thus we leave the choice to the readers/users for using them for more recent periods. Finally, we observed that Fick and Hijmans (2017) just published a new version of their database for the period before 2000 including solar radiation, humidity and wind speed at 1 km resolution. Thus, we believe that there may be not problems related to the fact that our raster products do not include information after 2000, or because the wind rasters showed discrepancies with observed data, which mainly cover periods after 2000.

Comment R2.4: - Page 8. Really I do not find useful the annual coefficients in areas that show strong climate seasonality (as in the majority of world regions). - In addition, there are not seasonal accuracy statistics, which can be much more relevant than annual ones.

Response: Before we proceed to any justifications about the use of annual coefficients, we would like to mention that the stations that we used in this study, present adequate seasonal variability, which can be visualized in the graphs of Fig.R5a,b,c,d. Figs.R5a,b show the box-whisker plots of mean monthly ETo ASCE-short values for the California-USA and Australia stations, respectively, while Figs.R5c,d provide the respective frequency (number of stations) for classes that describe the maximum difference ( $\Delta$ max) between maximum and minimum values of mean monthly ETo ASCE-short of the respective stations. Taking into account Fig.R5a,b,c,d, and especially R5c,d, we believe
that the validation dataset includes stations of high seasonality. Based on Figs.R5c,d, more than 50% of the stations present  $\Delta$ max of ETo-short greater than 150 mm month-1.

**[FIGURE R5, PLACE HERE]**

Fig.R5 (a) Box-Whisker plot of mean monthly ETo ASCE-short (mm month-1) for California-USA stations, (b) Box-Whisker plot of mean monthly ETo ASCE-short (mm month-1) for Australia stations, (c) frequency (number of stations) for each class that describes the difference between maximum and minimum values of mean monthly ETo ASCE-short for California-USA stations and (d) frequency (number of stations) for each class that describes the difference between maximum and minimum values of mean monthly ETo ASCE-short for California-USA stations and (d) frequency (number of stations) for each class that describes the difference between maximum and minimum values of mean monthly ETo ASCE-short for Australia stations.

We also have to stress that the revised coefficients of H-S and P-T methods are not just annual averages of the mean monthly coefficients but partial weighted averages (p.w.a.), which give more weight to the monthly coefficients of the months with higher ETo during the year excluding the coefficients of colder months that present unreasonably high or low coefficients (see procedure of Eq.7 in the manuscript for estimating the weighted averages). Thus, based on our experience and after handling with the stations and the raster data, we believe that the p.w.a. annual coefficients are very useful in areas of strong seasonality. The detailed reasons for selecting the annual p.w.a. coefficients were incorporated in the new subsection of the Discussion with title: "Reasons for using annual p.w.a. coefficients instead of monthly or seasonal ones in the case of H-S and P-T methods" It is also important to note that the derivation of annual coefficients is a pure optimization problem when stations data are used. For example, Cristea et al. (2013) derived coefficients of the P-T method for 106 stations that represent a range of climates across the contiguous USA. The coefficients were estimated by minimizing the sum of the squared residuals between the benchmark FAO-56 and P-T (optimization method) using data only for the period April-September. The obtained optimized values of the coefficients were interpolated in order to make
a map of the apt coefficient (the map is not available for comparisons). In this study, the maps of the coefficients are produced based on raster data and not stations data, which means that optimization should be performed pixel by pixel (~62000 pixels globally for the 0.5 degree resolution excluding Antarctica). This procedure would require special programming since readily available tool to perform this procedure does not exist in commercial or free GIS software packages. This is the main reason for using as an alternative method the Eqs.7 in GIS environment, since it can be calculated easily in raster calculators incorporated in the GIS packages while approximates to the optimized values because it gives more weight to the monthly coefficients of the warmer months. A solution could be the development of a tool for GIS purposes using rasters data that could be able to run using 24 rasters; 12 for the benchmark ETo and another 12 for the P-T or H-S ETo formula without the 1.26 and 0.0023 factors, respectively, in order to provide optimized annual values of their coefficients (for a global application filters to remove unreasonable values are also required). Finally, we have to clarify that Figs.8-10 in the manuscript include results of mean monthly values of each month of each station and not one value per station. We mention this because in the second part of the comment the reviewer notes that "there are not seasonal accuracy statistics, which can be much more relevant than annual ones". All the statistics that we provided in this study concern comparisons between observed and predicted mean monthly values by the models (160 stations  $\times$  12 mean monthly values = 1680 observations were tested for each parameter). We believe that the monthly comparison includes also the seasonal one. Seasonal separation would create a problem due to the different seasons between northern and southern hemisphere. Comparative statistics per season would also create a great expansion of the article in the results but also in the discussion section, which are already large after the addition of the additional models (request of the reviewer 1). Additionally, we would like to present the seasonal statistics in new studies where we will present further analysis related to optimization methods and other new models separately for California and Australia stations. In order to provide something relevant to seasonal variations to the reviewer, we prepared

**ESSDD**
the Figs.R6 and R7, which will not be included in the manuscript. Fig.R6a and R6b give the average monthly ETo based on the mean monthly estimations from California and Australia stations, respectively, using the ASCE-short method, the P-T(p.w.a.s.), the H-S(p.w.a.s.) and VAL3 model (best model according to Table 5a in the manuscript). Similarly, Fig.R7a and R7b give the average monthly Rs based on the mean monthly observations from California and Australia stations, respectively, and based on the Rs estimations using the radiation formula of H-S with revised coefficients.

**[FIGURE R6, PLACE HERE]**

Fig.R6 Monthly average ETo based on mean monthly estimations using the ASCEshort method, the P-T(p.w.a.s.), the H-S(p.w.a.s.) and VAL3 model (best model according to Table 5a in the manuscript) for (a) the 60 stations of California and (b) the 80 stations of Australia (For Australia the graph starts from July).

**[FIGURE R7, PLACE HERE]**

Fig.R7 Monthly average Rs based on mean monthly observations and based on the radiation formula of H-S with revised coefficients for (a) the 60 stations of California and (b) the 80 stations of Australia (For Australia the graph starts from July).

Figs.R6,7 give a general indication about the seasonal variations in the ETo and Rs estimations by the models separately for California and Australia datasets, while they also provide a general overview about the underestimation/overestimation of each model per month in comparison to the benchmark values (ASCE-short or observed Rs). We believe that the general variation that was succeeded by the models is satisfactory in the context of a global application and any observed deviations are adequately justified by the uncertainties related to the data. The only thing that we have to address is the response of P-T(p.w.a.s.) model. The P-T(p.w.a.s.) was not as good as the H-S(p.w.a.s.) (the same thing was also observed between the standard H-S and P-T methods). The prevalence of H-S can be attributed to the fact that the majority of stations from Table 1 are located in territories with negative DMAD values (Fig.5a) giving a ESSDD
general advantage to H-S method for more robust estimations (this explanation is also mentioned in the manuscript, for more details see Page 20, lines 1-10). In the case of Australia the bad performance is evident during the cold months, but it presented better performance on the warmer months (DJF) in comparison to H-S(p.w.a.s.) and VAL3.

Other major corrections made in the text: 1. Some affiliations changed because some authors were transferred to other institutions or because one of the Institutions changed name. 2. The abstract reformed in order to be more descriptive. 3. Any analysis related to finer resolutions below 0.5 degrees was removed from the text following the comments of reviewer 2. For this reason, the 30 arc-sec resolution maps given in Figs.2,3,4,5,6,7 were substituted with the ones of 0.5 degree resolution with respective changes in the range of values in their legends. Any discussion about the comparison of different resolutions was also removed from the discussion section. Additionally, all the results and tables changed based on 0.5 degree resolution. Similar changes were also made in the supplementary material. The only reference about the finer resolutions is given in section 5. Data availability, where we added the following text: "Apart from the 0.5 degree resolution raster datasets, the database contains the same datasets at finer resolution (30 arc-sec, 2.5 arc-min, 5 arc-min and 10 arc-min). These finer datasets are provided in order to cover the observed resolution range in the initial climatic data (e.g. the temperature data of Hijmans et al. (2005) are provided at 30 arc-sec resolution). The finer resolutions were produced using bilinear interpolation on solar radiation, humidity and wind speed data of Sheffield et al. (2006). This interpolation method is not the most appropriate for such purposes. The data of finer resolutions can only be used as a tool to assess uncertainties associated to temperature variation effects within a 0.5 degree pixel or to estimate average values of the coefficients for larger territories in order to capture a better representation of the coastlines or islands that do not exist in 0.5 degree resolution (use of values from individual pixels is not recommended). A complete list of the datasets is provided in the Table S5." 4. The reviewer also commented that the manuscript is guite long (Comment R2.1). For this
reason, we removed the accuracy analysis by splitting the stations based on their elevation, and we also removed the Taylor diagrams analysis since the criteria that we give in Table 5 are more than enough. 5. We added another 8 models of short reference crop evapotranspiration for comparative purposes after the request of Reviewer 1. 6. The Discussion section was completely reformed in order to create subsections (request of reviewer 1). 7. An error was found in the coordinates of Australian station Paynes Find station (A-69) of the validation dataset and the associated coefficients extracted from the specific coordinates. The position of the station was corrected in Fig.1 and any information related to the station was corrected. An additional arithmetic error was found and corrected in the ETo ASCE estimations of Australian stations. We performed a detailed check for all stations data, all the calculations/equations used for rasters development, all the calculations/equations used for analyzing stations data.

References Allen, R. G., Pereira, L. S., Raes, D., and Smith, M.: Crop Evapotranspiration: Guidelines for computing crop water requirements. Irrigation and Drainage Paper 56, Food and Agriculture Organization of the United Nations, Rome, 1998. Allen, R. G., Walter, I. A., Elliott, R., Howell, T., Itenfisu, D., and Jensen M.: The ASCE standardized reference evapotranspiration equation. Final Report (ASCE-EWRI). Pr. In: Allen RG, Walter IA, Elliott R, Howell T, Itenfisu D, Jensen M (Eds.) Environmental and Water Resources Institute, 2005. Task Committee on Standardization of Reference Evapotranspiration of the Environmental and Water Resources Institute, 2005. Fick, S. E., and Hijmans, R. J.: WorldClim 2: New 1-km spatial resolution climate surfaces for global land areas. International Journal of Climatology, Article in Press, 2017, DOI: 10.1002/joc.5086 Hang, X., Vincent, L. A., Hogg, W. D., and Niitsoo, A.: Temperature and precipitation trends in Canada during the 20th century. Atmosphere - Ocean, 38 (3), 395-429, 2000. Hijmans, R. J., Cameron, S. E., Parra, J. L., Jones, P. G., and Jarvis, ÎS.: Very high resolution interpolated climate surfaces for global land areas. Int. J. Climatol., 25, 1965-1978, 2005. McVicar, T. R., Roderick, M. L., Donohue, R. J., and Van Niel, T. G.: Less bluster ahead? ecohydrological implications of global trends of terrestrial near-surface wind speeds. Ecohydrology, 5, 381-388,
2012a. McVicar, T. R., Roderick, M. L., Donohue, R. J., Li, L. T., Van Niel, T. G., Thomas, A., Grieser, J., Jhajharia, D., Himri, Y., Mahowald, N. M., Mescherskaya, A. V., Kruger, A. C., Rehman, S., and Dinpashoh, Y.: Global review and synthesis of trends in observed terrestrial near-surface wind speeds: implications for evaporation. J. Hydrol., 416-417, 182-205, 2012b. Norrant, C., and Douguédroit, A.: Monthly and daily precipitation trends in the Mediterranean (1950-2000). Theor. Appl. Climatol., 83, 89-106, 2006. Peixoto, J. P., and Oort, A. H.: The climatology of relative humidity in the atmosphere. J. Climate, 9, 3443-3463, 1996. Sheffield, J., Goteti, G., and Wood, E. F.: Development of a 50-yr high-resolution global dataset of meteorological forcings for land surface modeling. J. Climate, 19(13), 3088-3111, 2006. Sheffield, J., and Wood, E. F.:. Global trends and variability in soil moisture and drought characteristics, 1950-2000, from observation-driven simulations of the terrestrial hydrologic cycle. J. Climate, 21, 432-458, 2008. Wild, M., Folini, D., Schär, C., Loeb, N., Dutton, E.G., and König-Langlo, G.: The global energy balance from a surface perspective, Clim. Dyn., 40, 3107-3134, 2013. Willett, K. M., Dunn, R. J. H., Thorne, P. W., Bell, S., De Podesta, M., Parker, D. E., Jones, P. D., and Williams, C. N.: HadISDH land surface multi-variable humidity and temperature record for climate monitoring. Clima. Past, 10, 1983-2006, 2014. You, Q., Kang, S., Aguilar, E., Pepin, N., Flugel, W.-A., Yan, Y., Xu, Y., Zhang, Y., and Huang, J.: Changes in daily climate extremes in China and their connection to the large scale atmospheric circulation during 1961-2003. Clim. Dyn., 36 (11-12), 2399-2417, 2011. Zomer, R. J., Trabucco, A., Bossio, D. A., van Straaten, O., and Verchot, L. V.: Climate change mitigation: A spatial analysis of global land suitability for clean development mechanism afforestation and reforestation. Agr. Ecosyst. Environ, 126, 67-80, 2008.

Please also note the supplement to this comment: http://www.earth-syst-sci-data-discuss.net/essd-2016-59/essd-2016-59-AC2supplement.pdf